# PROVABLE SEPARATIONS BETWEEN MEMORIZATION AND GENERALIZATION IN DIFFUSION MODELS

**Zeqi Ye**[*]
Northwestern University
zeqiye2029@u.northwestern.edu

**Qijie Zhu**[*]
Northwestern University
qijiezhu2029@u.northwestern.edu

**Molei Tao**
Georgia Institute of Technology
mtao@gatech.edu

**Minshuo Chen**
Northwestern University
minshuo.chen@northwestern.edu

## ABSTRACT

Diffusion models have achieved remarkable success across diverse domains, but they remain vulnerable to memorization—reproducing training data rather than generating novel outputs. This not only limits their creative potential but also raises concerns about privacy and safety. While empirical studies have explored mitigation strategies, theoretical understanding of memorization remains limited. We address this gap through developing a dual-separation result via two complementary perspectives: statistical estimation and network approximation. From the estimation side, we show that the ground-truth score function does not minimize the empirical denoising loss, creating a separation that drives memorization. From the approximation side, we prove that implementing the empirical score function requires network size to scale with sample size, spelling a separation compared to the more compact network representation of the ground-truth score function. Guided by these insights, we develop a pruning-based method that reduces memorization while maintaining generation quality in diffusion transformers.

## 1 INTRODUCTION

Diffusion models have emerged as one of the most powerful families of generative models, achieving state-of-the-art performance across a wide range of tasks (Song & Ermon, 2019; Ho et al., 2020; Song et al., 2020a;b; Kong et al., 2020; Mittal et al., 2021; Jeong et al., 2021; Huang et al., 2022; Avrahami et al., 2022; Ulhaq & Akhtar, 2022). Applications span image synthesis (Nichol et al., 2021; Yang et al., 2024), molecular design (Weiss et al., 2023; Guo et al., 2024), and time-series modeling (Tashiro et al., 2021; Alcaraz & Strodthoff, 2022), where diffusion models consistently generate samples of high fidelity. Their remarkable empirical success has established them as a leading paradigm in modern generative modeling.

Despite these advances, diffusion models have raised critical concerns. A central one is memorization, where trained models reproduce training data instead of generating genuinely novel samples (Gu et al., 2023; Stein et al., 2023; Webster, 2023; Kadkhodaie et al., 2023; Rahman et al., 2025; Chen et al., 2024). Such behavior undermines the creative potential of generative modeling and threatens the promise of generalization (Somepalli et al., 2023; Carlini et al., 2023). Memorization also leads to serious risks for data privacy and intellectual property, as training datasets may include copyrighted works or sensitive information (Ghalebikesabi et al., 2023; Cui et al., 2023; Vyas et al., 2023).

A growing body of research has attempted to characterize and mitigate memorization in diffusion models. Empirical studies have explored its correlation with data duplication, training procedure, and model architecture and capacity (Somepalli et al., 2023; Gu et al., 2023; Stein et al., 2023), and proposed defenses such as dataset de-duplication, modified training objectives, or improved sampling strategies (Wen et al., 2024; Ross et al., 2024; Wang et al., 2024). These methods provide

---

[*]Equal contribution.

valuable heuristics yet leave principles underneath their success underexplored. In parallel, theoretical investigations have begun to analyze memorization from a statistical perspective. For instance, asymptotic analyses, where both sample size and data dimension grow proportionally, have provided insights into the interplay between data availability, model complexity, and generalization (Raya & Ambrogioni, 2023; Biroli et al., 2024; George et al., 2025). However, these analyses do not fully explain memorization in practical, finite-sample regimes, leaving open a fundamental question:

*Can we disentangle memorization from generalization in practical regimes and mitigate it?*

In this work, we take a step toward addressing this question. We develop a non-asymptotic analysis that theoretically explains the emergence of memorization through the dual lenses of statistical estimation and neural function approximation. Our analysis reveals that memorization is fundamentally tied to the statistical properties of the training objective—the denoising score matching loss, and the approximation capacity of score neural networks. More specifically, from the statistical estimation side, we show that the ground-truth score function does not minimize the empirical denoising score matching loss, leading to an inherent gap that drives memorization. From the approximation side, we establish results demonstrating that the empirical score function demands network size scaling with the sample size, whereas the ground-truth score admits a compact representation. Guided by these insights, we explore empirical consequences and mitigation strategies. Our experiments not only validate the theories but also introduce a pruning-based method that reduces memorization while maintaining generation quality for diffusion transformers.

Our contributions are summarized as follows.

• **Statistical separation theory**: We show that the denoising score matching loss admits an inherent gap between the ground-truth score function and the empirical score function (Proposition 4.1). Furthermore, for mixture models, we provide a lower bound on the gap in Theorem 4.3, which provides a formal characterization of how memorization arises from a statistical perspective.

• **Neural architectural separation theory**: We establish bounds on neural networks approximating both ground-truth and empirical score functions in Theorem 5.1. Our results reveal that the ground-truth score function admits a compact neural representation, whereas approximating the empirical score function requires the network size to grow with the sample size.

Guided by our theory, we conduct experiments in Section 6 that (a) validate our insights regarding memorization and generalization in diffusion models, and (b) propose mitigation strategies that reduce memorization while preserving generation quality.

**Notations**: For a vector $x$, we use $\|x\|_2$ to denote its Euclidean norm, $\|x\|_1$ to denote its $\ell_1$-norm, and $\|x\|_\infty$ to denote its $\ell_\infty$-norm. For a matrix $A$, $\|A\|_2$ and $\|A\|_F$ denote its spectral norm and Frobenius norm, respectively, and $\|A\|_\infty = \max_{i,j} |A_{ij}|$. We use $\mathcal{O}(\cdot)$ to suppress multiplicative constants in upper bounds, while $\widetilde{\mathcal{O}}(\cdot)$ further suppresses logarithmic factors. Similarly, $\Omega(\cdot)$ suppresses multiplicative constants in lower bounds, and $\Theta(\cdot)$ suppresses constants in both upper and lower bounds.

## 2 RELATED WORK

Memorization and generalization in diffusion models have drawn increasing attention in recent years. In this section we provide an overview of progress on both empirical and theoretical sides.

From an empirical perspective, memorization is a significant issue observed across various settings, raising practical concerns about privacy, copyright, and model generalization (Ghalebikesabi et al., 2023; Cui et al., 2023; Vyas et al., 2023). This phenomenon is widely identified in different domains, and researchers have revealed several contributing factors, such as training dataset size and score network size, and have proposed corresponding general mitigation methods like data augmentation and data de-duplication (Somepalli et al., 2023; Gu et al., 2023; Stein et al., 2023; Webster, 2023; Kadkhodaie et al., 2023; Rahman et al., 2025; Chen et al., 2024). More targeted mitigation methods have also been developed recently, including tracing memorized samples to network architectural activations for pruning-based remedies (Chavhan et al., 2024; Hintersdorf et al., 2024), excluding trigger tokens (Wen et al., 2024), and penalizing manifold memorization (Ross et al.,

2024). Interested readers may refer to a recent survey (Wang et al., 2024) for a more comprehensive exposure of contributing factors and mitigation methods for memorization.

From a theoretical perspective, memorization in diffusion models has been analyzed from a statistical physics perspective, with a focus on phase transition phenomena (Biroli et al., 2024; Li et al., 2023; Ambrogioni, 2023; Ventura et al., 2024; Raya & Ambrogioni, 2023; Sakamoto et al., 2024; Pavasovic et al., 2025). For example, Biroli et al. (2024) relate the sample generation process to memorization and generalization of diffusion models by identifying critical transitions in generation trajectories. George et al. (2025) use asymptotic analysis of random-feature denoisers, which are functionally equivalent to score networks, to characterize learning curves and reveal the inherent trade-offs between generalization and memorization. Bonnaire et al. (2025) provide an asymptotic analysis of the training dynamics of random-feature denoisers, identifying a generalization–memorization phase transition and examining how network architectural regularization mitigates memorization, with their theoretical findings supported by extensive numerical experiments. Baptista et al. (2025) also investigate the dynamics of empirical score matching through a dynamical-systems lens, identifying a training-time generalization–memorization transition and demonstrating how various forms of regularization help prevent memorization. Other lines of work emphasize the role of implicit bias in underparameterized denoisers (Kamb & Ganguli, 2024; Niedoba et al., 2024; Vastola, 2025) and how dataset statistics shape a model's generalization behavior (Lukoianov et al., 2025).

During the preparation of this manuscript, we are aware of a closely related work (Buchanan et al., 2025), where memorization and generalization properties in well-separated Gaussian mixture distributions are studied. By considering a specific type of denoiser parameterized by Gaussian mixture, they demonstrate a sharp transition from generalization to memorization as the capacity of the network increases. Different from their study, our analysis holds for generic sub-Gaussian distributions and establishes a statistical separation theory. In addition, we analyze the representation power of general score neural networks and show another separation for approximating empirical and ground-truth score functions. Based on our theoretical insights, we further develop mitigation methods to improve generalization.

## 3 DIFFUSION MODEL AND DATA DISTRIBUTION REGULARITY

In this section, we briefly review the continuous-time formulation of diffusion models and introduce the structural assumptions on the data distribution that will be used throughout our analysis.

**Score-based diffusion model** A score-based diffusion model aims to learn and sample from an unknown data distribution $P_{\text{data}}$ by estimating the score function (Song & Ermon, 2019; Ho et al., 2020; Song et al., 2020a;b). It consists of coupled forward and backward processes. We adopt a continuous-time description, where the forward process is

$$\mathrm{d}X_t = -\frac{1}{2}X_t\mathrm{d}t + \mathrm{d}B_t \quad \text{for} \quad X_0 \sim P_{\text{data}} \text{ and } B_t \text{ is a standard Brownian motion.}$$

The forward process gradually corrupts the data distribution by Gaussian noise injection. Here $P_{\text{data}}$ represents the ground-truth data distribution. We denote $P_t$ as the marginal distribution of $X_t$ at time $t$ and $p_t$ the corresponding density function. In practice, the forward process terminates at a sufficiently large time $T$.

The backward process reverses the noise corruption in the forward process—often referred to as denoising for new sample generation. Mathematically, the backward process is

$$\mathrm{d}\widetilde{X}_t = \left[\frac{1}{2}\widetilde{X}_t + \nabla\log p_{T-t}(\widetilde{X}_t)\right]\mathrm{d}t + \mathrm{d}\widetilde{B}_t \quad \text{for} \quad \widetilde{X}_0 \sim P_T,$$

where $\widetilde{B}_t$ is another Brownian motion and $\nabla\log p_t$ is the score function. To simulate the backward process, one needs to estimate the score function using samples from the data distribution.

● Score estimation. We collect i.i.d samples $\mathcal{D} = \{x_1, x_2, ..., x_n\}$ from the data distribution $P_{\text{data}}$, we estimate the score function by minimizing the following denoising score matching loss:

$$\widehat{\mathcal{L}}(s) = \int_{t_0}^T \frac{1}{n}\sum_{i=1}^n \ell_t(x_i, s)\mathrm{d}t \quad \text{with} \quad \ell_t(x_i, s) = \mathbb{E}_{X_t|X_0=x_i}\left[\left\|-\frac{X_t - \alpha_t x_i}{\sigma_t^2} - s(X_t, t)\right\|_2^2\right], \quad (3.1)$$

where $\alpha_t = e^{-t/2}$ and $\sigma_t^2 = 1 - e^{-t}$. Note that $t_0$ is an early-stopping time to prevent score blow-up and secure numerical stability (Song et al., 2020b; Ho et al., 2020). The estimator $s$ is parameterized by a large-scale neural network such as a UNet (Ronneberger et al., 2015) or a transformer (Peebles & Xie, 2023).

● Empirical and ground-truth score function. Although the primary focus of optimizing (3.1) is to estimate the ground-truth score function $\nabla \log p_t$, the use of finite collected samples introduces a bias towards the so-called "empirical score function". More specifically, we denote $\widehat{P}_{\text{data}} = \frac{1}{n} \sum_{i=1}^{n} \mathbb{1}_{x_i}$ as the empirical data distribution. Let $\widehat{P}_t$ be the marginal distribution of the forward process if the initial state $X_0$ follows $\widehat{P}_{\text{data}}$. In fact, $\frac{1}{n} \sum_{i=1}^{n} \mathsf{N}(\alpha_t x_i, \sigma_t^2 I)$ is a Gaussian mixture with mean and variance dependent on time $t$. Consequently, $\widehat{P}_t$ induces the empirical score function defined as

$$\nabla \log \widehat{p}_t(x_t) = -\frac{1}{\sigma_t^2} \sum_{i=1}^{n} w_i(x_t)(x_t - \alpha_t x_i),$$

where $w_i(x_t)$ is a weight function; see detailed derivations in Appendix A.2.

An important property of the empirical score function is that it is the global minimizer of (3.1). Moreover, using the empirical score function, diffusion models only reproduce training data points instead of generating novel samples—known as memorization. Our theory in the sequel focuses on distinguishing the statistical behavior and representation requirement of empirical and ground-truth score functions, providing insights on the emergence of memorization.

**Data distribution regularity** To study different properties of empirical and ground-truth score functions, we consider sub-Gaussian data distributions with Hölder smoothness. These are commonly adopted regularity conditions in statistical literature and recent advances in the theory of diffusion models (Wasserman, 2006; Fu et al., 2024). We introduce Hölder regularity first.

**Definition 3.1** (Hölder norm). Let $\beta = s + \gamma > 0$ be a smoothness parameter, with $s = \lfloor \beta \rfloor$ an integer and $\gamma \in [0, 1)$. For a function $f : \mathbb{R}^d \to \mathbb{R}$, its Hölder norm is defined as

$$\|f\|_{\mathcal{H}^\beta(\mathbb{R}^d)} = \max_{\boldsymbol{s}:\|\boldsymbol{s}\|_1 < s} \sup_x |\partial^{\boldsymbol{s}} f(x)| + \max_{\boldsymbol{s}:\|\boldsymbol{s}\|_1 = s} \sup_{x \neq y} \frac{|\partial^{\boldsymbol{s}} f(x) - \partial^{\boldsymbol{s}} f(y)|}{\|x - y\|_2^\gamma},$$

where $\boldsymbol{s}$ is a multi-index. We say $f$ is $\beta$-Hölder if $\|f\|_{\mathcal{H}^\beta(\mathbb{R}^d)} < \infty$.

The Hölder ball of radius $B > 0$ is defined as

$$\mathcal{H}^\beta(\mathbb{R}^d, B) = \left\{ f : \mathbb{R}^d \to \mathbb{R} \,\big|\, \|f\|_{\mathcal{H}^\beta(\mathbb{R}^d)} < B \right\}.$$

We now specify a class of Hölder density functions that exhibit sub-Gaussian tail behavior.

**Definition 3.2** (Sub-Gaussian Hölder density). Let $C > 0$ and $c_f > 0$ be two positive constants. For any Hölder index $\beta > 0$, let $f \in \mathcal{H}^\beta(\mathbb{R}^d, B)$ for a constant radius $B > 0$ with $\inf_x f(x) \geq c_f$. A density function $p$ is *sub-Gaussian Hölder* if

$$p(x) = \exp(-C\|x\|_2^2/2) \cdot f(x).$$

Since $f$ is uniformly upper bounded, it holds that $p(x) \leq B \exp(-C\|x\|_2^2/2)$, which encapsulates sub-Gaussian densities widely studied in classical statistical literature (Wasserman, 2006). The lower bound on $f$ ensures the regularity of the ground-truth score function, as it is well-known that the regularity of the score function can be arbitrarily bad near low-density regions (Vahdat et al., 2021; Song & Ermon, 2020). Definition 3.1 is adopted in Fu et al. (2024) for establishing minimax optimal rate of conditional diffusion models. Yet our analysis tackles a more fine-grained understanding of the generalization capability of diffusion models.

# 4 STATISTICAL SEPARATION: GROUND-TRUTH SCORE DOES NOT MINIMIZE DENOISING SCORE MATCHING

In this section, we systematically show that the ground-truth score function does not minimize the denoising score matching loss (3.1). In particular, there exists a gap in the loss evaluated at the

empirical score function and at the ground-truth score function. The gap, perhaps surprisingly, may not vanish with polynomially many training samples. To begin with, we define

$$\texttt{Loss-Gap}_t = \frac{1}{n} \sum_{i=1}^{n} \left( \ell_t \left( x_i, \nabla \log p_t \right) - \ell_t \left( x_i, \nabla \log \widehat{p}_t \right) \right),$$

as the gap between the score matching loss at time $t$.

## 4.1 $\texttt{Loss-Gap}_t$ IS FISHER DIVERGENCE

We relate $\texttt{Loss-Gap}_t$ to the well-known *Fisher divergence* (Johnson & Barron, 2004; Holmes & Walker, 2017; Yang et al., 2019; Yamano, 2021). Fisher divergence has a fundamental connection to classical central limit theorems (Johnson & Barron, 2004) and has been widely adopted in machine learning and Bayesian inference (Hyvärinen & Dayan, 2005; Hyvärinen, 2007; Yang et al., 2019), change detection (Moushegian et al., 2025), and hypothesis testing (Wu et al., 2022). We state the formal result in the following proposition.

**Proposition 4.1.** For any time $t \leq T$, it holds that

$$\texttt{Loss-Gap}_t = \texttt{Fisher}(\widehat{P}_t, P_t),$$

where the divergence $\texttt{Fisher}(\widehat{P}_t, P_t) = \mathbb{E}_{X \sim \widehat{P}_t}[\|\nabla \log \widehat{p}_t(X) - \nabla \log p_t(X)\|_2^2]$.

The proof is provided in Appendix A.1. $\texttt{Loss-Gap}_t$ is analogous to the generalization bound of the empirical score function $\nabla \log \widehat{p}_t$, but fundamentally different. A generalization bound evaluates the deviation of $\nabla \log \widehat{p}_t$ from $\nabla \log p_t$ under the ground-truth data distribution $P_t$. Here, $\texttt{Loss-Gap}_t$ is evaluated under the empirical distribution $\widehat{P}_t$. Interestingly, Fisher divergence is not symmetric and $\texttt{Fisher}(P_t, \widehat{P}_t)$ coincides with the generalization bound of $\nabla \log \widehat{p}_t$. Existing literature presents fruitful studies on the generalization properties of diffusion models (Oko et al., 2023; Chen et al., 2023; Wibisono et al., 2024). Yet, the established analyses cannot be directly applied to our setting. Indeed, bounding $\texttt{Loss-Gap}_t$ can be much more involved due to its intricate dependence on the empirical score function and the loss evaluation over the same empirical data points. In the following section, we show a lower bound on $\texttt{Loss-Gap}_t$ under mixture models.

## 4.2 QUANTIFYING THE LOSS GAP IN MIXTURE OF DISTRIBUTIONS

We instantiate $P_{\text{data}}$ to a mixture of $K$ components with an equal prior, namely

$$P_{\text{data}} = \frac{1}{K} \sum_{k=1}^{K} P^{(k)}, \qquad \text{(Mixture Model)}$$

where each component $P^{(k)}$ admits a density $p^{(k)}$, and we denote by $X^{(k)} \sim P^{(k)}$ a random variable drawn from the $k$-th component with mean $\mathbb{E}[X^{(k)}] = \mu^{(k)}$ and covariance $\text{Cov}[X^{(k)}] = \Sigma$. Mixture Distributions align well with real-world datasets, which often exhibit multi-modality. For example, image datasets may contain distinct categories, such as cats and dogs in CIFAR-10 (Krizhevsky et al., 2009), that correspond to different components. For each component in the mixture model, we impose the following assumption.

**Assumption 4.2**. We represent $X^{(k)}$ as $X^{(k)} = \mu^{(k)} + \Sigma^{1/2}\xi$ and assume $\xi$ is a unit variance, entry-wise independent sub-Gaussian vector with $\|\xi\|_{\psi_2} = \mathcal{O}(1)$, where $\|\cdot\|_{\psi_2}$ denotes the sub-Gaussian norm (see Definition 3.4.1 in Vershynin (2018)). We also assume that $\|\Sigma\|_2 = \mathcal{O}(1), \|\Sigma\|_F = \mathcal{O}(\sqrt{d})$, and $\Sigma^{1/2}\xi$ admits the sub-Gaussian Hölder density defined in Definition 3.2. Additionally, we assume $\|\mu^{(k)}\|_2 = \mathcal{O}(\sqrt{d})$.

Assumption 4.2 ensures samples generated from the mixture are well separated with high probability when $\log(n) = \mathcal{O}(d)$. We define the minimum component separation distance as $\Delta_{\min} = \min_{j \neq k} \|\mu^{(j)} - \mu^{(k)}\|_2$. Equipped with these, we are ready to state a lower bound on $\texttt{Loss-Gap}_t$.

**Theorem 4.3** (Lower bound on $\texttt{Loss-Gap}_t$). Suppose $P_{\text{data}}$ takes the form (Mixture Model) with each component satisfying Assumption 4.2. Further assume the separation distance $\Delta_{\min} = \Theta(\sqrt{d})$. For $t_0$ and $t_1$ verifying $\log(\sigma_{t_0}) = \Omega(-d)$ and $\log(\sigma_{t_1}) = \mathcal{O}(-\log d)$ and sample size $\log n = \mathcal{O}(d)$, it holds that

$$\mathbb{E}_{\mathcal{D}}[\texttt{Loss-Gap}_t] = \Omega\left( d\sigma_t^{-2} + \text{tr}(\Sigma) \right) \quad \text{for all } t \in [t_0, t_1],$$

where $\mathbb{E}_{\mathcal{D}}$ denotes expectation with respect to the dataset $\mathcal{D}$. The proof of Theorem 4.3 is provided in Appendix A.2. We present several discussions.

**Small $t$ and large variance amplify the gap**
Theorem 4.3 says that for polynomially many training samples, `Loss-Gap`$_t$ is not negligible in the small-$t$ regime. We visualize `Loss-Gap`$_t$ in a Gaussian mixture setting in Figure 1. The $d\sigma_t^{-2}$ term arises from the Gaussian noise injected during data corruption, while the $\mathrm{tr}(\Sigma)$ term originates from the within-component variance. The effect of larger variance on increasing the loss gap can be understood through the Fisher divergence between $\widehat{P}_t$ and $P_t$. For the same number of samples, larger within-component variance makes the samples sparser in space, leading to a larger Fisher divergence between the Gaussian mixture $\widehat{P}_t$ formed by the samples and the true dis-

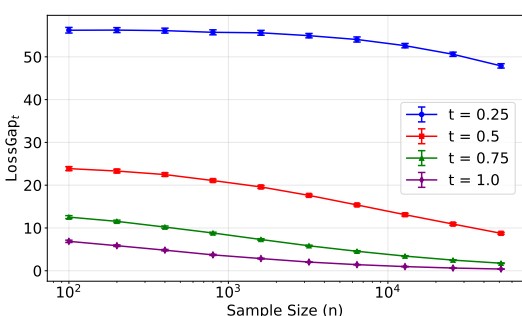

Figure 1: Smaller $t$ leads to larger `Loss-Gap`$_t$. When sample size $n$ is not sufficiently large, the gap is non-negligible.

tribution $P_t$. Although the divergence vanishes as $n \to \infty$, the convergence rate $n^{-1/d}$ is subject to the curse of dimensionality as shown in Weed & Bach (2019).

**Gap leads to memorization** Using Theorem 4.3 and revisiting (3.1), we can derive

$$\mathbb{E}_{\mathcal{D}}[\widehat{\mathcal{L}}(\nabla \log p_t) - \widehat{\mathcal{L}}(\nabla \log \widehat{p}_t)] = \int_{t_0}^{T} \mathbb{E}_{\mathcal{D}}[\text{Loss-Gap}_t] \mathrm{d}t \gtrsim \log(1/t_0) \cdot d + (t_1 - t_0)\mathrm{tr}(\Sigma).$$

This highlights an important mechanism of memorization: the training loss gap between the ground-truth score and the empirical score is non-negligible. Therefore, strong optimizers, e.g., Adam and AdamW, tend to drive a sufficiently expressive score network to learn the empirical score rather than the ground-truth score during training. This effect is more pronounced in higher dimensions.

**Extension to bounded support** Our analysis also applies to mixtures of well-separated components with bounded support. The key step in establishing Theorem 4.3 is to prove a reduced-form approximation to the empirical and ground-truth score functions, respectively. More specifically, for a given noisy state $X \sim \widehat{P}_t$ generated by injecting Gaussian noise into the empirical data points $x_i$, we argue that $\nabla \log \widehat{p}_t(X) \approx -\sigma_t^{-2}(X - \alpha_t x_i)$. Similarly, the ground-truth score function is dominated by $\nabla \log p_t(X) \approx \nabla \log p_t^{(k)}(X)$, where $x_i$ is sampled from the $k$-th component and $p_t^{(k)}$ is the density of the marginal distribution via applying diffusion process to the $P^{(k)}$. These approximations are valid thanks to the separation among the components. Bounded support naturally ensures this separation and hence the result follows.

## 5 ARCHITECTURAL SEPARATION: GROUND-TRUTH SCORE ALLOWS COMPACT REPRESENTATION

Section 4 establishes that `Loss-Gap`$_t$ does not vanish in the small-$t$ regime, implying that training a sufficiently expressive neural network with a strong optimizer can bias the training towards the empirical score function. Yet, it remains unknown whether a network is expressive enough. In this section, we investigate the representation requirements for the ground-truth and empirical score functions using ReLU networks and identify another gap in the complexity of the network architecture.

For simplicity, we focus on feedforward ReLU networks, while extending to other network architectures does not impose substantial challenges. We define a ReLU network architecture as $\mathcal{F}(W, L, N)$, where $W, L$ and $N$ are the width, depth, and non-zero parameters of the network. More specifically, we have

$$\mathcal{F}(W, L, N) = \big\{ f : f(x) = A_L \cdot \mathrm{ReLU}(A_{L-1} \cdot \mathrm{ReLU}(\dots \mathrm{ReLU}(A_1 x + b_1) \dots) + b_{L-1}) + b_L,$$

where $A_l \in \mathbb{R}^{d_{l-1} \times d_l}$ with $d_l \leq W$ for $l = 0, \ldots, L$ and $\sum_{l=1}^{L} \|A_l\|_0 + \|b_l\|_0 \leq N \}$.

Here $d_0$ represents the data dimension and $d_L$ represents the output dimension. The following theorem establishes approximation guarantees of the ground-truth and empirical score functions.

**Theorem 5.1.** Suppose that the density function of $P_{\text{data}}$ satisfies the sub-Gaussian Hölder density condition in Definition 3.2 with Hölder index $\beta$. For any sufficiently small $\epsilon > 0$, choose the early-stopping time $t_0$ satisfying $\log t_0 = \mathcal{O}(\log \epsilon)$ and the terminal time $T = \mathcal{O}(\log \epsilon^{-1})$. Then there exist network architectures $\mathcal{F}_1(W_1, L_1, N_1)$ and $\mathcal{F}_2(W_2, L_2, N_2)$ giving rise to

$$s_1 \in \mathcal{F}_1(W_1, L_1, N_1) \quad \text{and} \quad s_2 \in \mathcal{F}_2(W_2, L_2, N_2),$$

such that for any $t \in [t_0, T]$, it holds that

$$\mathbb{E}_{\mathcal{D}}\left[\mathbb{E}_{X_t \sim \widehat{P}_t}\left[\left\|s_1(X_t, t) - \nabla \log \widehat{p}_t(X_t)\right\|_2^2\right]\right] \leq \frac{\epsilon}{\sigma_t^4} \quad \text{and} \tag{5.1}$$

$$\mathbb{E}_{\mathcal{D}}\left[\mathbb{E}_{X_t \sim \widehat{P}_t}\left[\left\|s_2(X_t, t) - \nabla \log p_t(X_t)\right\|_2^2\right]\right] \leq \frac{\epsilon}{\sigma_t^2}. \tag{5.2}$$

The configurations of $\mathcal{F}_1$ and $\mathcal{F}_2$ are

$$W_1 = \widetilde{\mathcal{O}}(n \log^3 \epsilon^{-1}), \qquad L_1 = \widetilde{\mathcal{O}}(\log^2 \epsilon^{-1}), \qquad N_1 = \widetilde{\mathcal{O}}(n \log^4 \epsilon^{-1}) \quad \text{and} \tag{5.3}$$

$$W_2 = \widetilde{\mathcal{O}}\left(\epsilon^{-\frac{d}{2\beta}} \log^7 \epsilon^{-1}\right), \qquad L_2 = \widetilde{\mathcal{O}}(\log^4 \epsilon^{-1}), \qquad N_2 = \widetilde{\mathcal{O}}\left(\epsilon^{-\frac{d}{2\beta}} \log^9 \epsilon^{-1}\right). \tag{5.4}$$

The proof is provided in Appendix B. The key idea of the proof is to rewrite the score function as $\nabla \log p_t(x) = \nabla p_t(x)/p_t(x)$ and then construct ReLU networks for approximating the numerator and denominator separately. Note that (5.1) is equivalent to the denoising score matching loss (3.1). Thus, minimizing (3.1) over a sufficiently large network identified in (5.3) using a strong optimizer will bias training toward the empirical score function. Probing the network size upper bounds and the corresponding approximation error, we make the following interpretations.

**Network size depends on sample size** The configuration of the network architecture $\mathcal{F}_1(W_1, L_1, N_1)$ depends on the sample size $n$ and the desired approximation error $\epsilon$, whereas the configuration of the ground-truth network $s_2$ depends on $\epsilon^{-\frac{d}{2\beta}}$. More specifically, as $n$ increases, the required width $W$ and the total number of parameters $N$ for $\mathcal{F}_1$ will increase. This distinction highlights the potentially greater complexity involved in approximating the empirical score function, as it corresponds to a Gaussian mixture distribution with $n$ components.

**Sample Duplication and Memorization** Empirical observations, such as those in Somepalli et al. (2023), show that training sample duplication plays a significant role in memorization. From our insights in Theorem 5.1, sample duplication reduces the complexity required for the network to represent the empirical score function, thereby making memorization more likely. As stated in (5.3) and (5.4), when the dataset contains $n$ i.i.d. samples, approximating the empirical score requires both the network width and the number of non-zero parameters to scale with $n$, while the corresponding quantities for approximating the true score do not depend on the sample size. However, if $m$ samples are duplicates of the remaining $n - m$ i.i.d. samples, then from a theoretical viewpoint, the dataset effectively has size $n - m < n$. This implies that duplication makes the empirical score easier to represent and requires a less expressive network. Consequently, duplication makes memorization more likely to appear.

**Different sensitivity to time $t$** We also observe that the approximation errors in (5.1) and (5.2) exhibit a distinction in the dependence on variance $\sigma_t^2$. The empirical score function reproduces the empirical training data distribution $\widehat{P}_{\text{data}}$, which does not have a smooth density function. Consequently, the empirical score function becomes highly irregular when $t$ approaches 0, making it substantially more difficult to represent. On the contrary, the ground-truth score function possesses better regularity as the data distribution satisfies the sub-Gaussian Hölder condition. We dive deeper into this regularity contrast in the sequel.

**Lipschitz continuity of score functions** We investigate the Lipschitz continuity of score functions by computing the Hessian matrix of log density. As shown in Lemma C.1 in Appendix C, we have

$$\nabla^2 \log p_t(x_t) = -\frac{1}{\sigma_t^2}I + \frac{\alpha_t^2}{\sigma_t^4}\,\mathrm{Cov}[X_0|X_t = x_t].$$

The same result applies to the empirical density $\widehat{p}_t$ by replacing $\mathrm{Cov}[X_0|X_t = x_t]$ with the empirical counterpart induced by training samples. For a small time $t$, we show that the Lipschitz coefficient—the supremum operator norm of the Hessian of the empirical score is bounded by $\Omega(\sigma_t^{-4} \cdot \min_{i \neq j} \|x_i - x_j\|_2^2)$, which depends on the separation of the training samples and variance $\sigma_t^2$. In contrast, the Lipschitz continuity of the ground-truth score of a sub-Gaussian Hölder distribution in Definition 3.2 behaves much better. As a concrete example, for a Gaussian distribution $P_{\mathrm{data}} = \mathcal{N}(\mu, \Sigma)$, denote $\lambda_{\min}(\Sigma)$ as the smallest eigenvalue of $\Sigma$, we have

$$\left\|\nabla^2 \log p_t\right\|_2 = \frac{1}{\sigma_t^2 + \alpha_t^2\,\lambda_{\min}(\Sigma)} = \mathcal{O}(1) \quad \text{for any } t.$$

**Weight decay effectively control the Lipschitz continuity** Weight decay controls the Lipschitz continuity of neural networks by penalizing the Frobenius norms of the weight matrices (Krogh & Hertz, 1991; Loshchilov & Hutter, 2017; Zhang et al., 2018). It has been implemented widely for training large-scale complex neural networks. Motivated by the separation in Lipschitz coefficient, we demonstrate the effectiveness of weight decay for mitigating memorization in Section 6, as the score network can hardly represent the empirical score function with well-controlled smoothness.

## 6 NUMERICAL RESULTS

We conduct experiments on both a simulated Gaussian mixture dataset and CIFAR-10 (Krizhevsky et al., 2009) to validate our theoretical insights and evaluate the effectiveness of our proposed theory-driven memorization mitigation strategies.

### 6.1 EXPERIMENTS ON GAUSSIAN MIXTURE DATASET

We explore how network size, training sample size and data dimension affect generalization and memorization. Additionally, we demonstrate that weight decay and network pruning are effective remedies for memorization, which validates our theoretical insights. For the purpose of evaluating memorization in numerical experiments, following Buchanan et al. (2025); Yoon et al. (2023), we identify memorization as follows.

Given a training dataset $\{x_i\}_{i=1}^n$ and a trained diffusion model $\mathcal{M}$, we say that a sample $x_{\mathrm{new}}$ generated by $\mathcal{M}$ is memorized if $\|x_{\mathrm{new}} - x_{(1)}\|_2^2 \leq \frac{1}{9}\|x_{\mathrm{new}} - x_{(2)}\|_2^2$, where $x_{(k)}$ is the $k$-th nearest neighbor in Euclidean norm to $x_{\mathrm{new}}$ in $(x_i)_{i=1}^n$. Further, we call the proportion of memorized samples within a batch of new samples drawn from $\mathcal{M}$ the memorization ratio.

We specify $P_{\mathrm{data}} = \frac{1}{K}\sum_{k=1}^K \mathcal{N}(\mu^{(k)}, I_d)$, where $\mu^{(k)}, k \in [K]$ are well-separated. As a teaser, we set $d = 2, K = 4$ to visualize how network size affects memorization, which is shown in Figure 2.

In the following experiments, we set $K = 8$, and draw $\mu^{(k)}$ independently from $\mathcal{N}(0, 4I_d)$. We first examine the relationship between memorization ratio, training sample size $n$, and data dimension $d$. The results are shown in Figure 3a. We initially fix the data dimension at $d = 32$ while varying the training sample size

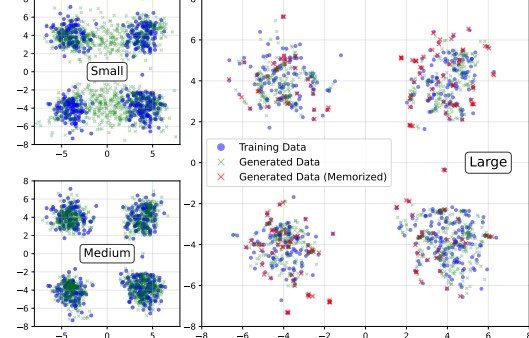

Figure 2: Learning 2D Gaussian mixture with varying network sizes. Increasing the network size leads to a clear progression: from failing to capture the underlying distribution, to partial generalization, and eventually to memorization. Memorized samples generated by the largest network are highlighted in red.

and network size. The results indicate that larger networks exhibit stronger memorization capacity, while more training samples reduce memorization ratio. We then fix the network size (12M parameters) to analyze the effects of training sample size and data dimension. The results show that higher dimension leads to lower memorization as data is harder to replicate.

We then leverage our theoretical insights to explore potential remedies for memorization. Motivated by the theoretical insights in Theorem 5.1, we conduct further experiments to investigate the effects of network width and weight decay. The results are presented in Figure 3b. With sufficient sample size ($n$=10K), memorization is less likely and increasing network width promotes generalization (measured by mean log-likelihood, where higher is better), while strong weight decay is harmful. However, with reduced sample size ($n$=3.2K), wide networks and light weight decay both lead to a high memorization ratio and severely impair generalization, while proper network width and weight decay prevent memorization and improve generalization. These findings validate that choosing appropriate network widths and applying weight decay during training are effective strategies to mitigate memorization.

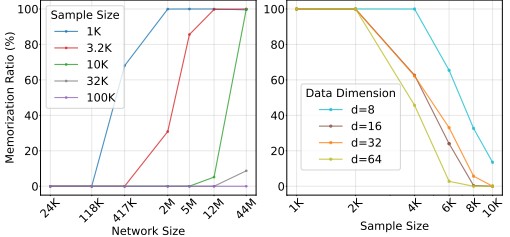 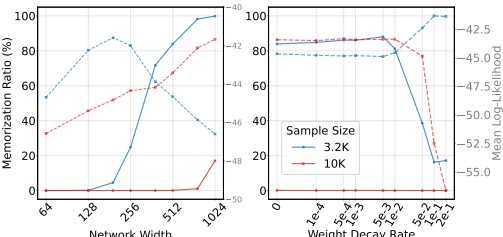

(a) (**Left**): fixed data dimension with varying sample sizes and network sizes. (**Right**): fixed network size with varying sample sizes and data dimensions.

(b) (**Left**): fixed network depth with varying widths and sample sizes. (**Right**): fixed network width with varying weight decay rates and sample sizes.

Figure 3: Comparison of experimental results on Gaussian mixture data. In (b), solid lines show memorization ratio, dashed lines show mean log-likelihood.

## 6.2 EXPERIMENTS ON CIFAR-10

Motivated by our theoretical insights and results on the effect of network width from synthetic experiments above, we propose a pruning method as a plug-and-play approach for trained diffusion models to reduce memorization.

**Pruning to mitigate memorization** Pruning has been widely adopted for trained diffusion models, either to reduce network size for faster inference while maintaining performance (Fang et al., 2025), or to remove specific memorized samples by identifying the responsible neurons (Hintersdorf et al., 2024). We propose a one-shot pruning method for trained Diffusion Transformers (DiTs) (Peebles & Xie, 2023). In particular, motivated by Theorems 4.3 and 5.1, we identify and prune attention heads that contribute least in the small-$t$ regime, followed by fine-tuning. This forces the remaining heads to represent the data with reduced capacity, which in turn encourages the model to learn the ground-truth score rather than overfit to the empirical score. The full procedure is summarized in Algorithm 1. We adapt importance score computation from Liang et al. (2021), with details provided in Appendix D.1.

**Performance of our pruning method** We evaluate our pruning method on the CIFAR-10 (Krizhevsky et al., 2009) dataset. First, we randomly select a subset of 5,000 samples and train a DiT on this dataset. We then apply our pruning method with diffusion time step sampling distribution $\mathcal{T} = \mathrm{Beta}(0.8, 2)$ and set the pruning ratio $\eta = 20\%$. For comparison, we also evaluate the original model and a random pruning baseline with the same pruning ratio. Besides memorization ratio and FID, we also use the precision and recall metrics of Kynkäänniemi et al. (2019). Definitions are provided in Appendix D.2, where recall measures diversity and generation coverage. The results in Table 1 show both our method and random pruning reduce memorization, but our method achieves higher recall and maintains a competitive FID, indicating improved diversity without sacrificing much fidelity. See Figure 4 for a comparison between the images generated by the original model and our pruned model.

---

**Algorithm 1** One-Shot Pruning for Diffusion Transformers

---

1: **Input:**
2:   Dataset $\mathcal{D}$, trained DiT model $\mathcal{M}$ with heads $\mathcal{H} = \{h_1, \ldots, h_H\}$.
3:   Time sampling distribution $\mathcal{T}$, which shall put more density on small $t$.
4:   Pruning percentage $\eta \in [0, 1]$, fine-tuning steps $M$.
5: Compute importance scores $\{I^{(h)}\}_{h \in \mathcal{H}} \leftarrow \text{IMPORTANCESCORE}(\mathcal{M}, \mathcal{D}, \mathcal{T})$.
6: Identify the set $\mathcal{H}_{\text{prune}}$ of $\lfloor \eta \cdot H \rfloor$ heads with the lowest importance scores.
7: Prune all heads $h \in \mathcal{H}_{\text{prune}}$ from the model $\mathcal{M}$.
8: **for** $m = 1, \ldots, M$ **do**
9:   Fine-tune the pruned model $\mathcal{M}$ on a batch from $\mathcal{D}$.
10: **Output:** The pruned model $\mathcal{M}$.

---

Although pruning slightly reduces precision, this is expected, as a high memorization ratio can artificially inflate precision by replicating training samples. For completeness, we also vary the pruning ratio and report additional results in Appendix D.3.

| Model | Precision ($\uparrow$) | Recall ($\uparrow$) | Memorization Ratio (%) ($\downarrow$) | FID ($\downarrow$) |
|---|---|---|---|---|
| Original | $\mathbf{0.39}_{\pm 0.01}$ | $0.08_{\pm 0.01}$ | $73.82_{\pm 1.12}$ | $15.47_{\pm 0.28}$ |
| Our Pruning | $0.33_{\pm 0.02}$ | $\mathbf{0.12}_{\pm 0.01}$ | $68.58_{\pm 0.77}$ | $\mathbf{15.07}_{\pm 0.33}$ |
| Random Pruning | $0.30_{\pm 0.02}$ | $0.09_{\pm 0.01}$ | $\mathbf{66.87}_{\pm 0.94}$ | $17.14_{\pm 0.25}$ |

Table 1: Comparison of the original model, our pruning method, and random pruning. Each value is mean$_{\pm \text{std}}$ over 5 runs. Best results are in bold.

## 7 CONCLUSIONS AND LIMITATIONS

In this work, we present a theoretical framework to explain memorization in diffusion models, examining it from the perspectives of both statistical separation and architectural separation. From the statistical separation side, we show that the ground-truth score function does not minimize the denoising score matching loss, and we quantify this discrepancy for generic sub-Gaussian mixture models. From the architectural separation side, we establish theoretical bounds on the approximation capabilities of neural networks for both the true and empirical score functions, demonstrating the separation of network size. Finally, we validate these theoretical insights through a series of experiments and propose a novel pruning method to mitigate memorization based on our findings.

While our work provides valuable insights, it has a few limitations. First, although we quantify the discrepancy for sub-Gaussian mixture models—a very common case—our theoretical framework does not yet extend to heavy-tailed distributions. Second, while our pruning methods are effective in our experiments, we lack the computational resources to fully validate their performance on larger datasets and models. We hope that future work can address these challenges.

## ACKNOWLEDGMENT

MT is thankful for partial supports by NSF Grants DMS-1847802, DMS-2513699, DOE Grants NA0004261, SC0026274, and Richard Duke Fellowship. The authors also thank Giulio Biroli and Ricardo Baptista for helpful discussions that improved this work.

## ETHICS STATEMENT

This work follows the ICLR Code of Ethics. We do not involve human or animal subjects, and all datasets and models are obtained under proper usage guidelines without compromising privacy. Our study avoids biases and discriminatory outcomes, does not use personally identifiable information, and poses no risks to privacy or security. We are committed to conducting this research with transparency and integrity.

## REPRODUCIBILITY STATEMENT

We take reproducibility seriously and provide all necessary materials to support it. Theoretical results are stated with clear assumptions, and complete proofs are provided in Appendix A, B and C. Experimental settings and implementation details are described in Appendix D. These materials ensure that our claims and results can be verified and reproduced.

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

# A    PROOF OF PROPOSITION 4.1 AND THEOREM 4.3

## A.1    PROOF OF PROPOSITION 4.1

*Proof.* The proof relies on a rewrite of the score functions. For the ground-truth score function and any empirical sample $x_i$, we have

$$\nabla \log p_t(x_t) \overset{(i)}{=} -\frac{1}{\sigma_t^2}(x_t - \alpha_t x_i) - \frac{\alpha_t}{\sigma_t^2} \frac{\int (x_i - x_0)\exp(-\frac{1}{2\sigma_t^2}\|x_t - \alpha_t x_0\|_2^2)\mathrm{d}P_{\mathrm{data}}(x_0)}{\int \exp(-\frac{1}{2\sigma_t^2}\|x_t - \alpha_t x_0\|_2^2)\mathrm{d}P_{\mathrm{data}}(x_0)}$$

$$\overset{(ii)}{=} -\frac{1}{\sigma_t^2}(x_t - \alpha_t x_i) - \frac{\alpha_t}{\sigma_t^2}(x_i - \mu_{0|t}(x_t)), \tag{A.1}$$

where in equality $(i)$, we insert $\alpha_t x_i$, and in equality $(ii)$, we denote

$$\mu_{0|t}(x_t) = \frac{\int x_0 \exp(-\frac{1}{2\sigma_t^2}\|x_t - \alpha_t x_0\|_2^2)\mathrm{d}P_{\mathrm{data}}(x_0)}{\int \exp(-\frac{1}{2\sigma_t^2}\|x_t - \alpha_t x_0\|_2^2)\mathrm{d}P_{\mathrm{data}}(x_0)}.$$

Recalling the definition of $\ell_t(x_i, \cdot)$ in (3.1) and plugging in (A.1), we obtain

$$\frac{1}{n}\sum_{i=1}^{n}\ell_t(x_i, \nabla \log p_t) = \frac{1}{n}\sum_{i=1}^{n}\mathbb{E}_{X_t|x_i}\left[\left\|\frac{\alpha_t}{\sigma_t^2}(x_i - \mu_{0|t}(X_t))\right\|_2^2\right].$$

By analogously denoting

$$\widehat{\mu}_{0|t}(x_t) = \frac{\int x_0 \exp(-\frac{1}{2\sigma_t^2}\|x_t - \alpha_t x_0\|_2^2)\mathrm{d}\widehat{P}_{\mathrm{data}}(x_0)}{\int \exp(-\frac{1}{2\sigma_t^2}\|x_t - \alpha_t x_0\|_2^2)\mathrm{d}\widehat{P}_{\mathrm{data}}(x_0)},$$

we have

$$\frac{1}{n}\sum_{i=1}^{n}\ell_t(x_i, \nabla \log \widehat{p}_t) = \frac{1}{n}\sum_{i=1}^{n}\mathbb{E}_{X_t|x_i}\left[\left\|\frac{\alpha_t}{\sigma_t^2}(x_i - \widehat{\mu}_{0|t}(X_t))\right\|_2^2\right].$$

Combining them, we have

$$\texttt{Loss-Gap}_t = \frac{1}{n}\sum_{i=1}^{n}\mathbb{E}_{X_t|x_i}\left[\left\|\frac{\alpha_t}{\sigma_t^2}(x_i - \mu_{0|t}(X_t))\right\|_2^2\right]$$

$$- \frac{1}{n}\sum_{i=1}^{n}\mathbb{E}_{X_t|x_i}\left[\left\|\frac{\alpha_t}{\sigma_t^2}(x_i - \widehat{\mu}_{0|t}(X_t))\right\|_2^2\right]. \tag{A.2}$$

To compare the terms in A.2, it suffices to fix an arbitrary time $t \in [t_0, T]$. Starting with the ground-truth denoising loss, we have

$$\frac{1}{n}\sum_{i=1}^{n}\mathbb{E}_{X_t|x_i}\left[\left\|\frac{\alpha_t}{\sigma_t^2}(x_i - \mu_{0|t}(X_t))\right\|_2^2\right]$$

$$= \frac{\alpha_t^2}{\sigma_t^4}\frac{1}{n}\sum_{i=1}^{n}\mathbb{E}_{X_t|x_i}\left[\left\|x_i - \widehat{\mu}_{0|t}(X_t) + \widehat{\mu}_{0|t}(X_t) - \mu_{0|t}(X_t)\right\|_2^2\right]$$

$$= \frac{\alpha_t^2}{\sigma_t^4}\frac{1}{n}\sum_{i=1}^{n}\mathbb{E}_{X_t|x_i}\left[\left\|x_i - \widehat{\mu}_{0|t}(X_t)\right\|_2^2\right]$$

$$+ \frac{\alpha_t^2}{\sigma_t^4}\frac{1}{n}\sum_{i=1}^{n}\mathbb{E}_{X_t|x_i}\left[\left\|\widehat{\mu}_{0|t}(X_t) - \mu_{0|t}(X_t)\right\|_2^2\right]$$

$$+ 2\frac{\alpha_t^2}{\sigma_t^4}\underbrace{\frac{1}{n}\sum_{i=1}^{n}\mathbb{E}_{X_t|x_i}\left[\left(x_i - \widehat{\mu}_{0|t}(X_t)\right)^{\top}\left(\widehat{\mu}_{0|t}(X_t) - \mu_{0|t}(X_t)\right)\right]}_{(\spadesuit)}. \tag{A.3}$$

We claim that $(\spadesuit) = 0$. In fact, we have

$$
\begin{aligned}
(\spadesuit) &= 2\frac{\alpha_t^2}{\sigma_t^4}\mathbb{E}_{X_0\sim\widehat{P}_{\text{data}}}\mathbb{E}_{X_t|X_0}\left[\left(X_0 - \widehat{\mu}_{0|t}(X_t)\right)^\top\left(\widehat{\mu}_{0|t}(X_t) - \mu_{0|t}(X_t)\right)\right] \\
&\overset{(i)}{=} 2\frac{\alpha_t^2}{\sigma_t^4}\mathbb{E}_{X_t}\mathbb{E}_{X_0|X_t}\left[\left(X_0 - \widehat{\mu}_{0|t}(X_t)\right)^\top\left(\widehat{\mu}_{0|t}(X_t) - \mu_{0|t}(X_t)\right)\right] \\
&= 2\frac{\alpha_t^2}{\sigma_t^4}\mathbb{E}_{X_t}\left[\left(\widehat{\mu}_{0|t}(X_t) - \widehat{\mu}_{0|t}(X_t)\right)^\top\left(\widehat{\mu}_{0|t}(X_t) - \mu_{0|t}(X_t)\right)\right] \\
&= 0,
\end{aligned}
$$

where equality $(i)$ follows from the tower property of conditional expectation. As a result, comparing (A.2) and (A.3) gives rise to

$$
\texttt{Loss-Gap}_t = \frac{\alpha_t^2}{\sigma_t^4}\cdot\frac{1}{n}\sum_{i=1}^n\mathbb{E}_{X_t|x_i}\left[\left\|\widehat{\mu}_{0|t}(X_t) - \mu_{0|t}(X_t)\right\|_2^2\right]. \tag{A.4}
$$

To further simply the expression, we apply Tweedie's Formula(Robbins, 1992) and have

$$
\mathbb{E}[X_0|X_t = x_t] = \frac{\sigma_t^2\nabla\log p_t(x_t) + x_t}{\alpha_t},
$$

which immediately gives us

$$
\frac{\alpha_t^2}{\sigma_t^4}\frac{1}{n}\sum_{i=1}^n\mathbb{E}_{X_t|x_i}\left[\left\|\widehat{\mu}_{0|t}(X_t) - \mu_{0|t}(X_t)\right\|_2^2\right] = \frac{1}{n}\sum_{i=1}^n\mathbb{E}_{X_t|x_i}\left[\left\|\nabla\log\widehat{p}_t(X_t) - \nabla\log p_t(X_t)\right\|_2^2\right].
$$

Then we can conclude

$$
\begin{aligned}
\texttt{Loss-Gap}_t &= \frac{1}{n}\sum_{i=1}^n\mathbb{E}_{X_t|x_i}\left[\left\|\nabla\log\widehat{p}_t(X_t) - \nabla\log p_t(X_t)\right\|_2^2\right] \\
&= \mathbb{E}_{X\sim\widehat{P}_t}\left[\left\|\nabla\log\widehat{p}_t(X) - \nabla\log p_t(X)\right\|_2^2\right],
\end{aligned}
$$

and we complete the proof. $\qquad\qquad\square$

## A.2 PROOF OF THEOREM 4.3

The proof of Theorem 4.3 proceeds in three steps:

- **Step 1.** After simplifying $\texttt{Loss-Gap}_t$ to the form in (A.4), and assuming $P_{\text{data}}$ follows the mixture model (Mixture Model), we can express $\widehat{\mu}_{0|t}(x_t)$ and $\mu_{0|t}(x_t)$ as weighted sums: $\widehat{\mu}_{0|t}$ weights the contribution of individual samples, while $\mu_{0|t}$ weights the contribution of mixture components.

- **Step 2.** In the small-$t$ regime, on a high-probability event for both the diffusion noise and the samples (where their norms lie in a regular range), we identify the dominant weights in $\widehat{\mu}_{0|t}(x_t)$ and $\mu_{0|t}(x_t)$. If $x_t$ is the diffusion-corrupted version of a training sample $x_i$, then $\widehat{\mu}_{0|t}(x_t)$ is dominated by the contribution of $x_i$, whereas $\mu_{0|t}(x_t)$ is dominated by the contribution of the component that generated $x_i$. We also provide explicit lower bounds on these dominant weights.

- **Step 3.** Separating the dominant and residual terms in the weighted sums yields a lower bound on $\texttt{Loss-Gap}_t$.

We now proceed with the proof step by step.

### A.2.1 **Step 1.** SIMPLIFICATION OF (A.4)

For each $k \in [K]$, let $p_t^{(k)}$ denote the marginal density of the forward diffusion process at time $t$. Equipped with this notation, we can have a simpler discrete version of (A.4).

For $\widehat{\mu}_{0|t}(x_t)$ we have:

$$\widehat{\mu}_{0|t}(x_t) = \frac{\sum_{l=1}^n x_l \exp(-\frac{1}{2\sigma_t^2}\|x_t - \alpha_t x_l\|_2^2)}{\sum_{j=1}^n \exp(-\frac{1}{2\sigma_t^2}\|x_t - \alpha_t x_j\|_2^2)} = \sum_{l=1}^n \widehat{w}_t^{(l)}(x_t)x_l, \tag{A.5}$$

where $\widehat{w}_t^{(l)}(x_t) = \frac{\exp(-\frac{1}{2\sigma_t^2}\|x_t - \alpha_t x_l\|_2^2)}{\sum_{j=1}^n \exp(-\frac{1}{2\sigma_t^2}\|x_t - \alpha_t x_j\|_2^2)}$ for $l = 1, 2, \cdots, n$.

As for $\mu_{0|t}(x_t)$, noticing that

$$p_t^{(k)}(x_t) = (2\pi\sigma_t^2)^{-\frac{d}{2}} \int \exp(-\frac{1}{2\sigma_t^2}\|x_t - \alpha_t x_0\|_2^2) p^{(k)}(x_0) \mathrm{d}x_0,$$

we have

$$
\begin{aligned}
\mu_{0|t}(x_t) &= \frac{\sum_{k=1}^K \int x_0 \exp(-\frac{1}{2\sigma_t^2}\|x_t - \alpha_t x_0\|_2^2) p^{(k)}(x_0) \mathrm{d}x_0}{\sum_{k=1}^K \int \exp(-\frac{1}{2\sigma_t^2}\|x_t - \alpha_t x_0\|_2^2) p^{(k)}(x_0) \mathrm{d}x_0} \\
&= \frac{(2\pi\sigma_t^2)^{-\frac{d}{2}} \sum_{k=1}^K \int x_0 \exp(-\frac{1}{2\sigma_t^2}\|x_t - \alpha_t x_0\|_2^2) p^{(k)}(x_0) \mathrm{d}x_0}{\sum_{j=1}^K p_t^{(j)}(x_t)} \\
&= \sum_{k=1}^K \frac{p_t^{(k)}(x_t)}{\sum_{j=1}^K p_t^{(j)}(x_t)} \int x_0 \left[ (2\pi\sigma_t^2)^{-\frac{d}{2}} \exp(-\frac{1}{2\sigma_t^2}\|x_t - \alpha_t x_0\|_2^2) p^{(k)}(x_0) / p_t^{(k)}(x_t) \right] \mathrm{d}x_0 \\
&= \sum_{k=1}^K w_t^{(k)}(x_t) \mu_{0|t}^{(k)}(x_t), \tag{A.6}
\end{aligned}
$$

where we denote $w_t^{(k)}(x_t) = \frac{p_t^{(k)}(x_t)}{\sum_{j=1}^K p_t^{(j)}(x_t)}, \mu_{0|t}^{(k)}(x_t) = \frac{\int x_0 \exp(-\frac{1}{2\sigma_t^2}\|x_t - \alpha_t x_0\|_2^2) p^{(k)}(x_0)\mathrm{d}x_0}{\int \exp(-\frac{1}{2\sigma_t^2}\|x_t - \alpha_t x_0\|_2^2) p^{(k)}(x_0)\mathrm{d}x_0}$, for $k \in [K]$.

After simplification, $\texttt{Loss-Gap}_t$ can be rewritten as

$$\texttt{Loss-Gap}_t = \frac{\alpha_t^2}{\sigma_t^4} \frac{1}{n} \sum_{i=1}^n \mathbb{E}_{X_t|x_i} \left[ \left\| \sum_{l=1}^n \widehat{w}_t^{(l)}(X_t)x_l - \sum_{k=1}^K w_t^{(k)}(X_t)\mu_{0|t}^{(k)}(X_t) \right\|_2^2 \right].$$

For the sake of simplicity, we further denote

$$\Delta_i \triangleq \mathbb{E}_{X_t|x_i} \left[ \left\| \sum_{l=1}^n \widehat{w}_t^{(l)}(X_t)x_l - \sum_{k=1}^K w_t^{(k)}(X_t)\mu_{0|t}^{(k)}(X_t) \right\|_2^2 \right].$$

### A.2.2  Step 2. BOUNDING THE DOMINANT WEIGHTS WITHIN CERTAIN EVENT

We first denote $\epsilon = \Sigma^{1/2}\xi$, following the notations in Assumption 4.2. We can then write the decomposition of $X^{(k)}$ as

$$X^{(k)} = \mu^{(k)} + \epsilon, \ \epsilon \sim p_\epsilon, \ \mathbb{E}[\epsilon] = 0, \ \mathrm{Cov}(X^{(k)}) = \mathrm{Cov}(\epsilon) = \Sigma.$$

And thus, under Assumption 4.2, there exist some constants $C_1, C_2, C_3 > 0$ such that

$$\epsilon = \Sigma^{1/2}\xi, \ \mathbb{E}[\xi] = 0, \ \mathrm{Cov}[\xi] = I_d, \ \|\xi\|_{\psi_2} \le C_1, \ \|\Sigma\|_F \le C_2\sqrt{d}, \ \|\Sigma\|_2 \le C_3. \tag{A.7}$$

We define a mapping $c : [n] \to [K]$, where $c(i)$ maps $i$ to the index of the component from which it is generated. Equipped with this, we can write $x_i - \mu^{(c(i))} = \epsilon_i$. We now define a high probability event $\mathcal{E}_1$ for sample norm and their well-separation properties. Invoking Corollary A.3, we can specify a high probability event $\mathcal{E}_1$, within which the samples are well separated, and their norms are in a regular range. The statement in the corollary suggests that, for $\delta \in (0, 1)$, with high probability at least $1 - \delta$, we have

$$\min_{i,j \in [n]} \|\epsilon_i - \epsilon_j\|_2^2 \ge \frac{2\, y_l(\delta/2n)}{C} d - \frac{4}{C}\sqrt{\frac{d}{c_0}\log(n^2/\delta)}, \text{ and}$$

$$\frac{y_l(\delta/2n)}{C} d \ \leq \ \inf_{i \in [n]} \|x_i - \mu^{(c(i))}\|_2^2 \leq \ \sup_{i \in [n]} \|x_i - \mu^{(c(i))}\|_2^2 \ \leq \ \frac{y_u(\delta/2n)}{C} d.$$

Thus, the following event holds with probability at least $1 - \delta$:

$$\mathcal{E}_1 \ \triangleq \ \left\{ x_1, \ldots, x_n \ \middle| \ \|\epsilon_i - \epsilon_j\|_2^2 \right.$$

$$\left. \geq \frac{2\,y_l(\delta/2n)}{C} d \ - \ \frac{4}{C} \sqrt{\frac{d}{c_0} \log(n^2/\delta)}, \quad \forall\, i, j \in [n] \right\}$$

$$\cap \ \left\{ x_1, \ldots, x_n \ \middle| \ \frac{y_l(\delta/2n)}{C} d \ \leq \ \inf_{i \in [n]} \|x_i - \mu^{(c(i))}\|_2^2 \right.$$

$$\left. \leq \ \sup_{i \in [n]} \|x_i - \mu^{(c(i))}\|_2^2 \ \leq \ \frac{y_u(\delta/2n)}{C} d \right\}.$$

Similarly, we can also define another similar high probability event for $Z$, the Gaussian noise introduced by diffusion. Invoking Lemma A.1, for $\delta_Z \in (0, 1)$, with high probability at least $1 - \delta_Z$ the following event holds

$$\mathcal{E}_2 \triangleq \left\{ \sqrt{d - 2\sqrt{d \log(2/\delta_Z)}} \leq \|Z\|_2 \leq \sqrt{d + 2\sqrt{d \log(2/\delta_Z)} + 2 \log(2/\delta_Z)} \right\}.$$

First, for the sake of simplicity, we can take $\delta_Z = \frac{\exp(-d/9)}{2}$ and analyze $t$ in a certain range such that $\frac{\sigma_t^2}{\alpha_t^2} \leq \frac{y_l(\delta/2n)}{8C}$. With such constraints, we can easily derive the following relationship:

$$\frac{\alpha_t}{2} \sqrt{\frac{y_u(\delta/2n)}{C}d} \geq \frac{\alpha_t}{2} \sqrt{\frac{y_l(\delta/2n)}{C}d} \geq \sigma_t \sqrt{d + 2\sqrt{d \log(2/\delta_Z)} + 2 \log(2/\delta_Z)}. \tag{A.8}$$

Additionally, we can make first-step simplifications of the weights.

According to (A.5),

$$\widehat{w}_t^{(i)}(X_t) = \frac{1}{1 + \sum_{j \neq i} \exp(-\frac{1}{2\sigma_t^2}(\|X_t - \alpha_t x_j\|_2^2 - \|X_t - \alpha_t x_i\|_2^2))}.$$

According to (A.6),

$$w_t^{(c(i))}(X_t) = \left[ 1 + \sum_{k \neq c(i)} \frac{q_t(X_t - \alpha_t \mu^{(k)})}{q_t(X_t - \alpha_t \mu^{(c(i))})} \right]^{-1}$$

$$\geq \left[ 1 + \sum_{k \neq c(i)} \frac{B}{c_f} \exp\left( -\frac{C(\|X_t - \alpha_t \mu^{(k)}\|_2^2 - \|X_t - \alpha_t \mu^{(c(i))}\|_2^2)}{2(\alpha_t^2 + C\sigma_t^2)} \right) \right]^{-1}.$$

The second inequality invokes Lemma A.5, which provides us an upper bound on the ratio of $q_t$ evaluated at different points.

Consequently, from the first-step simplifications, the analysis of the dominant weights reduces to the comparisons of different distances. Within $\mathcal{E}_1 \cap \mathcal{E}_2$, we can easily conduct such analysis.

**Distance analysis** Conditioned on $\mathcal{E}_1 \cap \mathcal{E}_2$, we discuss the following three kinds of distances for investigating the weight behaviors.

- Case 1: The distance term regarding $X_t = \alpha_t x_i + \sigma_t Z$ and $\mu^{(c(i))}$. We evaluate the distance $\|X_t - \alpha_t \mu^{(k)}\|_2$. According to the forward process, conditioning on $x_i$, we write $X_t$ as $X_t = \alpha_t x_i + \sigma_t Z$, where $Z \sim \mathsf{N}(0, I_d)$ independent of $X_i$. Thus, we derive

$$\|X_t - \alpha_t \mu^{(c(i))}\|_2 \leq \|X_t - \alpha_t x_i\|_2 + \alpha_t \|x_i - \mu^{(c(i))}\|_2$$

$$\leq \sigma_t \|Z\|_2 + \alpha_t \sqrt{\frac{y_u(\delta/2n)}{C} d},$$

where the second inequality leverages the fact that, within $\mathcal{E}_1$ the norm of the samples are controlled. Consequently, we deduce

$$\|X_t - \alpha_t \mu^{(c(i))}\|_2 \leq \sigma_t \sqrt{d + 2\sqrt{d \log(2/\delta_Z)} + 2\log(2/\delta_Z)} + \alpha_t \sqrt{\frac{y_u(\delta/2n)}{C} d}$$

$$\leq \alpha_t \sqrt{d} \left( \sqrt{\frac{y_u(\delta/2n)}{C}} + \frac{1}{2}\sqrt{\frac{y_l(\delta/2n)}{C}} \right), \tag{A.9}$$

where the first inequality leverages the fact that, within $\mathcal{E}_2$ the norm of the diffusion noise is controlled, and the last inequality leverages (A.8).

On the other hand, by the triangle inequality, we have

$$\|X_t - \alpha_t \mu^{(c(i))}\|_2 \geq \max\left\{ \sigma_t \|Z\|_2 - \alpha_t \|x_i - \mu^{(k)}\|_2, \alpha_t \|x_i - \mu^{(k)}\|_2 - \sigma_t \|Z\|_2 \right\}.$$

For the first term in the maximum above, we have

$$\sigma_t \|Z\|_2 - \alpha_t \|x_i - \mu^{(k)}\|_2 \geq \sigma_t \sqrt{d - 2\sqrt{d \log(2/\delta_Z)}} - \alpha_t \sqrt{\frac{y_u(\delta/2n)}{C} d}. \tag{A.10}$$

Similarly, we have

$$\alpha_t \|X_t - \alpha_t \mu^{(c(i))}\|_2 - \sigma_t \|Z\|_2$$

$$\geq \alpha_t \sqrt{\frac{y_l(\delta/2n)}{C} d} - \sigma_t \sqrt{d + 2\sqrt{d \log(2/\delta_Z)} + 2\log(2/\delta_Z)}$$

$$\geq \frac{\alpha_t}{2} \sqrt{\frac{y_l(\delta/2n)}{C} d}, \tag{A.11}$$

where the last inequality leverages (A.8). Taking maximum over (A.10) and (A.11) leads to

$$\|X_t - \alpha_t \mu^{(c(i))}\|_2 \geq \max\left\{ \sigma_t \sqrt{d - 2\sqrt{d \log(2/\delta_Z)}} - \alpha_t \sqrt{\frac{y_u(\delta/2n)}{C} d}, \frac{\alpha_t}{2} \sqrt{\frac{y_l(\delta/2n)}{C} d} \right\}$$

$$\geq \frac{\alpha_t}{2} \sqrt{\frac{y_l(\delta/2n)}{C} d}. \tag{A.12}$$

- Case 2: The distance terms regarding $X_t = \alpha_t x_i + \sigma_t Z$ and $\mu^{(k)}$, $k \neq c(i)$. We only need a lower bound on the distance $\|X_t - \alpha_t \mu^{(j)}\|_2$:

$$\|X_t - \alpha_t \mu^{(k)}\|_2 = \|X_t - \alpha_t \mu^{(c(i))} + \alpha_t \mu^{(c(i))} - \alpha_t \mu^{(k)}\|_2$$

$$\geq \alpha_t \|\mu^{(c(i))} - \mu^{(k)}\|_2 - \|X_t - \alpha_t \mu^{(c(i))}\|_2$$

$$\geq \alpha_t \Delta_{\min} - \alpha_t \sqrt{d} \left( \sqrt{\frac{y_u(\delta/2n)}{C}} + \frac{1}{2}\sqrt{\frac{y_l(\delta/2n)}{C}} \right), \tag{A.13}$$

where the last inequality leverages the definition of $\Delta_{\min}$ and the upper bound in (A.9).

- Case 3: The distance terms regarding $x_i$ and $x_j$. We have

$$\|X_t - \alpha_t x_j\|_2^2 - \|X_t - \alpha_t x_i\|_2^2$$

$$= \|\alpha_t (x_i - x_j) + \sigma_t Z\|_2^2 - \|\sigma_t Z\|_2^2$$

$$\geq \frac{1}{2} \alpha_t^2 \|x_i - x_j\|_2^2 - 2\sigma_t^2 \|Z\|_2^2.$$

If $c(i) = c(j)$, then by the definition of $\mathcal{E}_1$, we have

$$\|x_i - x_j\|_2^2 = \|\epsilon_i - \epsilon_j\|_2^2$$

$$\geq \frac{2\, y_l(\delta/2n)}{C} d - \frac{4}{C}\sqrt{\frac{d}{c_0} \log(n^2/\delta)},$$

and if $c(i) \neq c(j)$, we have

$$\|x_i - x_j\|_2^2 \geq \Delta_{\min}^2 - 2 \sup_{i \in [n]} \|\epsilon_i\|_2^2$$

$$\geq \Delta_{\min}^2 - \frac{2\, y_u(\delta/2n)}{C}\, d.$$

If we set

$$\Delta_{\min}^2 \geq \frac{2\, y_u(\delta/2n)}{C}\, d + \frac{2\, y_l(\delta/2n)}{C}\, d - \frac{4}{C}\sqrt{\frac{d}{c_0}\log(n^2/\delta)},$$

we can then have a union lower bound

$$\|x_i - x_j\|_2^2 \geq \frac{2\, y_l(\delta/2n)}{C}\, d\ - \frac{4}{C}\sqrt{\frac{d}{c_0}\log(n^2/\delta)}.$$

Thus,

$$\|X_t - \alpha_t x_j\|_2^2 - \|X_t - \alpha_t x_i\|_2^2$$

$$\geq \alpha_t^2 \frac{y_l(\delta/2n)}{C} d - \alpha_t^2 \frac{2}{C}\sqrt{\frac{d}{c_0}\log(n^2/\delta)} - 2\sigma_t^2\|Z\|_2^2$$

$$\geq \alpha_t^2 \frac{y_l(\delta/2n)}{C} d - \alpha_t^2 \frac{2}{C}\sqrt{\frac{d}{c_0}\log(n^2/\delta)} - 2\sigma_t^2(d + 2\sqrt{d\log(2/\delta_Z)} + 2\log(2/\delta_Z))$$

$$\geq \alpha_t^2 \frac{y_l(\delta/2n)}{C} d - \alpha_t^2 \frac{2}{C}\sqrt{\frac{d}{c_0}\log(n^2/\delta)} - \frac{1}{2}\alpha_t^2 \frac{y_l(\delta/2n)}{C} d$$

$$\geq \frac{\alpha_t^2}{2C}\left(y_l(\delta/2n)d - 4\sqrt{\frac{d}{c_0}\log(n^2/\delta)}\right), \tag{A.14}$$

where the second inequality leverages the norm range control within $\mathcal{E}_2$, and the third inequality leverages (A.8).

**Lower bounds of dominant weights**  Thus, within $\mathcal{E}_1 \cap \mathcal{E}_2$, we have

$$\widehat{w}_t^{(i)}(X_t) = \frac{1}{1 + \sum_{j \neq i}\exp(-\frac{1}{2\sigma_t^2}(\|X_t - \alpha_t x_j\|_2^2 - \|X_t - \alpha_t x_i\|_2^2))}$$

$$\geq \frac{1}{1 + (n-1)\exp\left(\frac{-\alpha_t^2 d}{2C\sigma_t^2}\left(y_l(\delta/2n)d - 4\sqrt{\frac{d}{c_0}\log(n^2/\delta)}\right)\right)}. \tag{A.15}$$

Leveraging Lemma A.5 and the bounds in (A.9), (A.13), and also setting

$$\Delta_{\min} \geq \left(2\left(\sqrt{\frac{y_u(\delta/2n)}{C}} + \frac{1}{2}\sqrt{\frac{y_l(\delta/2n)}{C}}\right) + 1\right)\sqrt{d},$$

we have

$$w_t^{(c(i))}(X_t) = \left[1 + \sum_{k \neq c(i)}\frac{q_t(X_t - \alpha_t\mu^{(k)})}{q_t(X_t - \alpha_t\mu^{(c(i))})}\right]^{-1}$$

$$\geq \left[1 + \sum_{k \neq c(i)}\frac{B}{c_f}\exp\left(-\frac{C(\|X_t - \alpha_t\mu^{(k)}\|_2^2 - \|X_t - \alpha_t\mu^{(c(i))}\|_2^2)}{2(\alpha_t^2 + C\sigma_t^2)}\right)\right]^{-1}$$

$$\geq \left[1 + \frac{B}{c_f}(K-1)\exp\left(-\frac{C}{2(\alpha_t^2 + C\sigma_t^2)}\right)\right.$$

$$\left.\cdot\left[(\alpha_t\Delta_{\min} - \alpha_t\sqrt{d}(\sqrt{\frac{y_u(\delta/2n)}{C}} + \frac{1}{2}\sqrt{\frac{y_l(\delta/2n)}{C}}))^2\right.\right.$$

$$- \alpha_t^2 d \left( \sqrt{\frac{y_u(\delta/2n)}{C}} + \tfrac{1}{2} \sqrt{\frac{y_l(\delta/2n)}{C}} \right)^2 \Big) \Big]^{-1}, \tag{A.16}$$

where the last inequality leverages the bounds in (A.9) and (A.13).

To further simplify the expressions, we shall notice that if we take $K = \mathrm{poly}(d)$, and $\log(n) = \mathcal{O}(\log(\delta) + d)$, we have the conditions on $\Delta_{\min}$ become $\Delta_{\min} = \mathcal{O}(\sqrt{d})$, and

$$y_l(\delta/2n)d - 4\sqrt{\frac{d}{c_0} \log(n^2/\delta)} = \Omega(d),$$

$$\left( \alpha_t \Delta_{\min} - \alpha_t \sqrt{d} \left( \sqrt{\frac{y_u(\delta/2n)}{C}} + \tfrac{1}{2} \sqrt{\frac{y_l(\delta/2n)}{C}} \right) \right)^2$$

$$- \left( \alpha_t^2 d \left( \sqrt{\frac{y_u(\delta/2n)}{C}} + \tfrac{1}{2} \sqrt{\frac{y_l(\delta/2n)}{C}} \right) \right)^2 = \Omega(d).$$

Thus, the bound in (A.16) can be simplified as

$$w_t^{(c(i))}(X_t) \gtrsim \left[ 1 + \exp\left( -\frac{C\alpha_t^2 d}{2(\alpha_t^2 + C\sigma_t^2)} \right) \right]^{-1}, \tag{A.17}$$

and the bound in (A.15) can be simplified a

$$\widehat{w}_t^{(i)}(X_t) \gtrsim \frac{1}{1 + n \exp\left( \frac{-\alpha_t^2 d}{2C\sigma_t^2} \right)}. \tag{A.18}$$

### A.2.3   **Step 3.** LOWER BOUND OF THE LOSS GAP

In the sequel, to simplify the derivation, we denote $\theta_t = \frac{\alpha_t^2}{\alpha_t^2 + C\sigma_t^2}$.

We now further simplify the loss gap $\texttt{Loss-Gap}_t$ by extracting the weights of dominating sample and component. Within $\mathcal{E}_1$ we can write

$$\Delta_i \geq \mathbb{E}_{X_t|x_i} \Big[ \big\| \widehat{w}_t^{(i)}(X_t) x_i - w_t^{(c(i))}(X_t)\, \mu_{0|t}^{(c(i))}(X_t) $$
$$+ \Big( \sum_{l \neq i} \widehat{w}_t^{(l)}(X_t) x_l - \sum_{k \neq c(i)} w_t^{(k)}(X_t)\, \mu_{0|t}^{(k)}(X_t) \Big) \big\|_2^2 \mathbf{1}\{\mathcal{E}_2\} \Big]$$

$$\geq \frac{1}{2} \mathbb{E}_{X_t|x_i} \Big[ \underbrace{\big\| \widehat{w}_t^{(i)}(X_t) x_i - w_t^{(c(i))}(X_t)\, \mu_{0|t}^{(c(i))}(X_t) \big\|_2^2}_{\mathcal{A}} \mathbf{1}\{\mathcal{E}_2\} \Big]$$

$$- \mathbb{E}_{X_t|x_i} \Big[ \underbrace{\big\| \sum_{l \neq i} \widehat{w}_t^{(l)}(X_t) x_l - \sum_{k \neq c(i)} w_t^{(k)}(X_t)\, \mu_{0|t}^{(k)}(X_t) \big\|_2^2}_{\mathcal{B}} \mathbf{1}\{\mathcal{E}_2\} \Big],$$

where the last inequality leverages the fact that $\|x - y\|_2^2 \geq \frac{1}{2}\|x\|_2^2 - \|y\|_2^2$.

Plugging in the expression of $\mu_{0|t}$ in Lemma A.6 gives rise to

$$\mathbb{E}_{X_t|x_i}[\mathcal{A}\mathbf{1}\{\mathcal{E}_2\}]$$

$$= \mathbb{E}_{X_t|x_i} \Big[ \big( \mathbf{1}\{\mathcal{E}_2\} \big\| \big( \widehat{w}_t^{(i)}(X_t) - w_t^{(c(i))}(X_t)\theta_t \big) x_i$$
$$- w_t^{(c(i))}(X_t)(1 - \theta_t)\mu^{(c(i))} - w_t^{(c(i))}(X_t)\theta_t \cdot \frac{\sigma_t}{\alpha_t} Z - w_t^{(c(i))}(X_t)\boldsymbol{E} \big\|_2^2 \big) \Big]$$

$$\geq \mathbb{E}_{X_t|x_i} \Big[ \big( \mathbf{1}\{\mathcal{E}_2\} \big\| \big( \widehat{w}_t^{(i)}(X_t) - w_t^{(c(i))}(X_t)\theta_t \big) x_i$$
$$- w_t^{(c(i))}(X_t)(1 - \theta_t)\mu^{(c(i))} - w_t^{(c(i))}(X_t)\theta_t \cdot \frac{\sigma_t}{\alpha_t} Z \big\|_2^2 \big) \Big] - 2\|\boldsymbol{E}\|_2^2$$

$$\geq \mathbb{E}_{X_t|x_i} \Big[ \big( \mathbf{1}\{\mathcal{E}_2\} \big\| \big( \widehat{w}_t^{(i)}(X_t) - w_t^{(c(i))}(X_t)\theta_t \big) x_i$$

$$- w_t^{(c(i))}(X_t)(1-\theta_t)\mu^{(c(i))} - w_t^{(c(i))}(X_t)\theta_t \cdot \tfrac{\sigma_t}{\alpha_t} Z \big\|_2^2 \big)\Big] - 2\mathcal{O}(\sigma_t^2/\alpha_t^2)$$

$$= \mathbb{E}_{X_t|x_i}\Big[\Big(\mathbf{1}\{\mathcal{E}_2\}\big\|\big(\widehat{w}_t^{(i)}(X_t) - w_t^{(c(i))}(X_t)\theta_t\big)x_i - w_t^{(c(i))}(X_t)(1-\theta_t)\mu^{(c(i))}\big\|_2^2\big)\Big]$$

$$+ \mathbb{E}_{X_t|x_i}\Big[\big\|w_t^{(c(i))}(X_t)\theta_t \cdot \tfrac{\sigma_t}{\alpha_t} Z\big\|_2^2 \mathbf{1}\{\mathcal{E}_2\}\Big]$$

$$- \mathbb{E}_{X_t|x_i}\Big[\big(w_t^{(c(i))}(X_t)\theta_t \cdot \tfrac{\sigma_t}{\alpha_t} Z\big)^\top \big((\widehat{w}_t^{(i)}(X_t) - w_t^{(c(i))}(X_t)\theta_t)x_i$$

$$- w_t^{(c(i))}(X_t)(1-\theta_t)\mu^{(c(i))}\big) \mathbf{1}\{\mathcal{E}_2\}\Big]$$

$$- 2\mathcal{O}(\sigma_t^2/\alpha_t^2). \tag{A.19}$$

The first term in (A.19) can be simplified as

$$\mathbb{E}_{X_t|x_i}\left[\left(\mathbf{1}\{\mathcal{E}_2\}\left\|\left(\widehat{w}_t^{(i)}(X_t) - \theta_t w_t^{(c(i))}(X_t)\right)x_i - w_t^{(c(i))}(X_t)(1-\theta_t)\mu^{(c(i))}\right\|_2^2\right)\right]$$

$$= \mathbb{E}_{X_t|x_i}\left[\left(\mathbf{1}\{\mathcal{E}_2\}\left\|\left(\widehat{w}_t^{(i)}(X_t) - w_t^{(c(i))}(X_t)\right)x_i - w_t^{(c(i))}(X_t)(1-\theta_t)(x_i - \mu^{(c(i))})\right\|_2^2\right)\right]$$

$$\geq \frac{1}{2}\mathbb{E}_{X_t|x_i}\left[\left(\mathbf{1}\{\mathcal{E}_2\}\left\|w_t^{(c(i))}(X_t)(1-\theta_t)(x_i - \mu^{(c(i))})\right\|_2^2\right)\right]$$

$$- \mathbb{E}_{X_t|x_i}\left[\left\|\left(\widehat{w}_t^{(i)}(X_t) - w_t^{(c(i))}(X_t)\right)x_i\right\|_2^2\right]$$

$$\gtrsim \frac{1}{2}(1-\theta_t)^2\left\|(x_i - \mu^{(c(i))})\right\|_2^2 - \mathbb{E}_{X_t|x_i}\left[\left\|\left(\widehat{w}_t^{(i)}(X_t) - w_t^{(c(i))}(X_t)\right)\right\|_2^2\right]\|x_i\|_2^2$$

$$\gtrsim \frac{1}{2}(1-\theta_t)^2\left\|(x_i - \mu^{(c(i))})\right\|_2^2 - \left[\left(\frac{\exp\left(-\frac{C\theta_t d}{2}\right)}{1 + \exp\left(-\frac{C\theta_t d}{2}\right)}\right)^2\right] \cdot \left(R_{\max}^2 + \frac{y_u(\delta/2n)}{C}d\right),$$

where the second last inequality leverages the fact that in our $t$ range (the condition of Lemma A.6, $\sigma_t \lesssim 1/\sqrt{d}$), $w_t^{c(i)}(X_t) \geq \frac{1}{2}$, the last inequality leverages the lower bound of the weight in (A.17), and the fact that within $\mathcal{E}_1$, $\sup_{i\in[n]}\|x_i\|^2 \leq \sup_{i\in[n]}\|\mu^{(c(i))}\|_2^2 + \|\epsilon_i\|_2^2 \leq R_{\max}^2 + \frac{y_u(\delta/2n)}{C}d$.

The second term in (A.19) can be simplified as

$$\mathbb{E}_{X_t|x_i}\left[\left\|\frac{w_t^{(c(i))}(X_t)\theta_t\sigma_t}{\alpha_t}Z\right\|_2^2 \mathbf{1}\{\mathcal{E}_2\}\right] \gtrsim \theta_t^2 \cdot \frac{\sigma_t^2}{\alpha_t^2} \cdot \left(\frac{1}{1 + \exp\left(-\frac{C\theta_t d}{2}\right)}\right)^2 \cdot d,$$

where the inequality leverages the fact that $\|Z\|_2 \geq \sqrt{d/3}$ within $\mathcal{E}_2$, and the lower bound of the weight in (A.17).

The third term in (A.19) can be simplified as

$$\mathbb{E}_{X_t|x_i}\left[\left(w_t^{(c(i))}(X_t)\theta_t \cdot \tfrac{\sigma_t}{\alpha_t}Z\right)^\top \left((\widehat{w}_t^{(i)}(X_t) - w_t^{(c(i))}(X_t)\theta_t)x_i\right.\right.$$

$$\left.\left. - w_t^{(c(i))}(X_t)(1-\theta_t)\mu^{(c(i))}\right)\mathbf{1}\{\mathcal{E}_2\}\right]$$

$$= \theta_t \cdot \tfrac{\sigma_t}{\alpha_t}\mathbb{E}_{X_t|x_i}\left[w_t^{(c(i))}(X_t)Z^\top\left((\widehat{w}_t^{(i)}(X_t) - \theta_t w_t^{(c(i))}(X_t))x_i\right)\mathbf{1}\{\mathcal{E}_2\}\right]$$

$$- \theta_t \cdot \tfrac{\sigma_t}{\alpha_t}\mathbb{E}_{X_t|x_i}\left[w_t^{(c(i))}(X_t)(1-\theta_t)Z^\top\mu^{(c(i))}\mathbf{1}\{\mathcal{E}_2\}\right].$$

We now decompose this expression by adding and subtracting the term $\theta_t \cdot \tfrac{\sigma_t}{\alpha_t}\mathbb{E}_{X_t|x_i}\left[Z^\top\big((1-\theta_t)(x_i - \mu^{(c(i))})\big)\mathbf{1}\{\mathcal{E}_2\}\right]$. This step is designed to isolate a component that is provably zero due to symmetry, leaving us with a residual term that we can then bound.

$$\mathbb{E}_{X_t|x_i}\left[\left(w_t^{(c(i))}(X_t)\theta_t \cdot \tfrac{\sigma_t}{\alpha_t}Z\right)^\top\left((\widehat{w}_t^{(i)}(X_t) - w_t^{(c(i))}(X_t)\theta_t)x_i\right.\right.$$

$$- w_t^{(c(i))}(X_t)(1 - \theta_t)\mu^{(c(i))}\Big)\mathbf{1}\{\mathcal{E}_2\}\Big]$$

$$= \theta_t \cdot \tfrac{\sigma_t}{\alpha_t}\, \mathbb{E}_{X_t|x_i}\Big[Z^\top\Big((1 - \theta_t)(x_i - \mu^{(c(i))})\Big)\mathbf{1}\{\mathcal{E}_2\}\Big]$$

$$+ \theta_t \cdot \tfrac{\sigma_t}{\alpha_t}\, \mathbb{E}_{X_t|x_i}\Big[Z^\top\Big((\theta_t - 1)(x_i - \mu^{(c(i))})$$

$$+ w_t^{(c(i))}(X_t)(\widehat{w}_t^{(i)}(X_t) - \theta_t w_t^{(c(i))}(X_t))x_i$$

$$- w_t^{(c(i))}(X_t)^2(1 - \theta_t)\mu^{(c(i))}\Big)\mathbf{1}\{\mathcal{E}_2\}\Big].$$

The first term in the equality above is exactly zero. This is because the expectation is over $Z$ and the vector $(1 - \theta_t)(x_i - \mu^{(c(i))})$ is a constant. The event $\mathcal{E}_2$ is symmetric (it depends only on $\|Z\|_2$), and the Gaussian density of $Z$ is also symmetric.

Therefore, the original cross-term is equal to the second term. We now bound the magnitude of this remaining term.

$$\Bigg|\mathbb{E}_{X_t|x_i}\Big[\Big(w_t^{(c(i))}(X_t)\theta_t\tfrac{\sigma_t}{\alpha_t}Z\Big)^\top\Big((\widehat{w}_t^{(i)}(X_t) - w_t^{(c(i))}(X_t)\theta_t)x_i$$

$$- w_t^{(c(i))}(X_t)(1 - \theta_t)\mu^{(c(i))}\Big)\mathbf{1}\{\mathcal{E}_2\}\Big]\Bigg|$$

$$= \tfrac{\theta_t\sigma_t}{\alpha_t}\Bigg|\mathbb{E}_{X_t|x_i}\Big[Z^\top\Big((\widehat{w}_t^{(i)}(X_t)w_t^{(c(i))}(X_t) - w_t^{(c(i))}(X_t)^2\theta_t + \theta_t - 1)x_i$$

$$- (1 - w_t^{(c(i))}(X_t)^2)(1 - \theta_t)\mu^{(c(i))}\Big)\mathbf{1}\{\mathcal{E}_2\}\Big]\Bigg|$$

$$\leq \tfrac{\theta_t\sigma_t}{\alpha_t}\sqrt{\mathbb{E}_{X_t|x_i}[\|Z\|_2^2]\,\mathbb{E}_{X_t|x_i}[(1 - w_t^{(c(i))}(X_t))^2\mathbf{1}\{\mathcal{E}_2\}]}\,(1 - \theta_t)\Big(\|x_i\|_2^2 + \|\mu^{(c(i))}\|_2^2\Big)$$

$$\lesssim \tfrac{\theta_t\sigma_t}{\alpha_t}(1 - \theta_t)\left(\frac{\exp(-\frac{C\theta_t d}{2})}{1 + \exp(-\frac{C\theta_t d}{2})}\right)\Big(R_{\max}^2 + \tfrac{y_u(\delta/2n)}{C}\,d\Big).$$

where the second inequality leverages Cauchy-Schwarz, and the last inequality leverages the fact that within $\mathcal{E}_1$, $\sup_{i\in[n]}\|x_i\|^2 \leq R_{\max}^2 + \tfrac{y_u(\delta/2n)}{C}\,d$ by Corollary A.3.

Collecting all the terms we have

$$\mathbb{E}_{X_t|x_i}[\mathcal{A}\mathbf{1}\{\mathcal{E}_2\}] \gtrsim \theta_t^2 \cdot \tfrac{\sigma_t^2}{\alpha_t^2} \cdot \left(\frac{1}{1 + \exp\left(-\frac{C\theta_t d}{2}\right)}\right)^2 \cdot d$$

$$+ \tfrac{1}{2}(1 - \theta_t)^2\left\|(x_i - \mu^{(c(i))})\right\|_2^2 - \left[\left(\frac{\exp\left(-\frac{C\theta_t d}{2}\right)}{1 + \exp\left(-\frac{C\theta_t d}{2}\right)}\right)^2\right]\cdot\Big(R_{\max}^2 + \tfrac{y_u(\delta/2n)}{C}d\Big)$$

$$- \tfrac{\theta_t\sigma_t}{\alpha_t}(1 - \theta_t)\left(\frac{\exp\left(-\frac{C\theta_t d}{2}\right)}{1 + \exp\left(-\frac{C\theta_t d}{2}\right)}\right)\Big(R_{\max}^2 + \tfrac{y_u(\delta/2n)}{C}d\Big). \tag{A.20}$$

Additionally, by the estimation of $\mu_{0|t}^{(k)}$ derived in Lemma A.6, within $\mathcal{E}_1 \cap \mathcal{E}_2$, we have

$$\mathcal{B} \leq 2(n - 1)\left(\frac{n\exp\left(\frac{-\alpha_t^2 d}{2C\sigma_t^2}\right)}{1 + n\exp\left(\frac{-\alpha_t^2 d}{2C\sigma_t^2}\right)}\right)^2 \cdot \sup_{j\in[n]}\|x_j\|_2^2$$

$$+ 2(K - 1)\left(\frac{\exp\left(-\frac{C\theta_t d}{2}\right)}{1 + \exp\left(-\frac{C\theta_t d}{2}\right)}\right)^2 \cdot \mathbb{E}_{X_t|x_i}\Big[\sup_{k\in[K]}\mu_{0|t}^{(k)}(X_t)\mathbf{1}\{\mathcal{E}_2\}\Big]$$

$$\lesssim \left[ n\left( \frac{n\exp\left(\frac{-\alpha_t^2 d}{2C\sigma_t^2}\right)}{1+n\exp\left(\frac{-\alpha_t^2 d}{2C\sigma_t^2}\right)} \right)^2 + K\left( \frac{\exp\left(-\frac{C\theta_t d}{2}\right)}{1+\exp\left(-\frac{C\theta_t d}{2}\right)} \right)^2 \right] \cdot \left( R_{\max}^2 + \frac{y_u(\delta/2n)}{C}d \right).$$

$$(A.21)$$

We can now summarize all the conditions we have imposed as

$$\Delta_{\min}, R_{\max} = \Theta\left(\sqrt{d}\right), \quad \log(n) = \mathcal{O}(\log(\delta)+d), \quad K = \text{poly}(d).$$

We focus on $t \in [t_0, t_1]$ where $t_0$ is chosen to satisfy $\log(\sigma_{t_0}) \gtrsim -d$, $t_1$ is chosen to satisfy $\log(\sigma_{t_1}) \lesssim -\log d$. With such conditions and time range, and by further noticing that when we take $\log(n) = \mathcal{O}(\log(\delta)+d)$, we have $y_u(\delta/2n), y_l(\delta/2n) = \Theta(1)$ (recalling their definitions in Corollary A.3), we shall have

$$n\left( \frac{n\exp\left(\frac{-\alpha_t^2 d}{2C\sigma_t^2}\right)}{1+n\exp\left(\frac{-\alpha_t^2 d}{2C\sigma_t^2}\right)} \right)^2, K\left( \frac{\exp\left(-\frac{C\theta_t d}{2}\right)}{1+\exp\left(-\frac{C\theta_t d}{2}\right)} \right)^2 = \mathcal{O}(\sigma_t^4),$$

which makes $\mathcal{B}$ and the third and fourth terms in (A.20) negligible. Thus we finally have within $\mathcal{E}_1$, we have

$$\begin{aligned}
\text{Loss-Gap}_t &= \frac{\alpha_t^2}{\sigma_t^4}\frac{1}{n}\sum_{i=1}^n \Delta_i \\
&\geq \frac{\alpha_t^2}{\sigma_t^4}\frac{1}{n}\sum_{i=1}^n \left( \frac{1}{2}\mathbb{E}_{X_t|x_i}[\mathcal{A}\mathbf{1}\{\mathcal{E}_2\}] - \mathbb{E}_{X_t|x_i}[\mathcal{B}\mathbf{1}\{\mathcal{E}_2\}] \right) \\
&\gtrsim \frac{\alpha_t^2}{\sigma_t^4}\left( \theta_t^2 \cdot \frac{\sigma_t^2}{\alpha_t^2} \cdot d + \frac{1}{n}\sum_{i=1}^n (1-\theta_t)^2 \|x_i - \mu^{(c(i))}\|_2^2 \right) \\
&\gtrsim \frac{d}{\sigma_t^2} + \frac{1}{n}\sum_{i=1}^n \|x_i - \mu^{(c(i))}\|_2^2,
\end{aligned}$$

where we shall recall that $\theta_t = \frac{\alpha_t^2}{\alpha_t^2+\sigma_t^2 C}$.

Finally, by taking $\delta = \exp(-d/2c)$ we have

$$\begin{aligned}
\mathbb{E}_{\mathcal{D}}[\text{Loss-Gap}_t] &\geq \mathbb{E}_{\mathcal{D}}\left[ \mathbf{1}\{\mathcal{E}_1\} \cdot \frac{\alpha_t^2}{\sigma_t^4}\frac{1}{n}\sum_{i=1}^n \Delta_i \right] \\
&\gtrsim \mathbb{E}_{\mathcal{D}}\left[ \mathbf{1}\{\mathcal{E}_1\} \cdot \frac{d}{\sigma_t^2} \right] + \frac{1}{n}\sum_{i=1}^n \mathbb{E}_{\mathcal{D}}\left[ (1-\mathbf{1}\{\mathcal{E}_1^c\}) \cdot \|x_i - \mu^{(c(i))}\|_2^2 \right] \\
&\gtrsim \frac{d}{\sigma_t^2} + \text{tr}(\text{Cov}(\epsilon)) - \delta \cdot \sqrt{\frac{1}{n}\sum_{i=1}^n \mathbb{E}_{\mathcal{D}}\left[ \|x_i - \mu^{(c(i))}\|_2^4 \right]} \\
&\gtrsim \frac{d}{\sigma_t^2} + \text{tr}(\Sigma),
\end{aligned}$$

where the second last inequality leverages Cauchy-Schwarz, and we complete the proof.

## A.3 SUPPORTING LEMMAS

We first present the classical lemma of $\chi^2$ concentration bound, to control the range of diffusion noise $Z$.

**Lemma A.1** (Laurent-Massart bound for $\chi^2$ concentration (Laurent & Massart, 2000))**.** Suppose a random variable $X \sim \chi_d^2$ with degrees of freedom $d$. Then for any $t > 0$, it holds that

$$\mathbb{P}[X - d \geq 2\sqrt{dt} + 2t] \leq \exp(-t),$$
$$\mathbb{P}[d - X \leq 2\sqrt{dt}] \leq \exp(-t).$$

We can next derive the following lemma to control the range of $\epsilon$.

**Lemma A.2** (Norm Concentration of $\epsilon$). Under Assumption 4.2 ($\epsilon$ satisfies the conditions in A.7), the following bounds hold:

1. **Upper Tail:** For any $\eta > 1/C - 1$,
$$\mathbb{P}\left(\|\epsilon\|_2^2 \geq (1+\eta)d\right) \leq \frac{B}{c_f} \exp\left(-\frac{d}{2}\left[C(1+\eta) - 1 - \log(C(1+\eta))\right]\right).$$

2. **Lower Tail:** For any $\eta \in (1 - 1/C, 1)$,
$$\mathbb{P}\left(\|\epsilon\|_2^2 \leq (1-\eta)d\right) \leq \frac{B}{c_f} \exp\left(-\frac{d}{2}\left[C(1-\eta) - 1 - \log(C(1-\eta))\right]\right).$$

Additionally, let
$$\tau(\delta) \;=\; \frac{2}{d}\log\left(\frac{2B}{c_f\,\delta}\right),$$
$$y_u(\delta) \;=\; (1+\tau(\delta)) \;+\; \sqrt{\tau(\delta)(2+\tau(\delta))}, y_l(\delta) \;=\; (1+\tau(\delta)) \;-\; \sqrt{\tau(\delta)(2+\tau(\delta))}.$$

Then, for any $\delta \in (0,1)$,
$$\mathbb{P}\left(\tfrac{y_l(\delta)}{C}d \;\leq\; \|\epsilon\|_2^2 \;\leq\; \tfrac{y_u(\delta)}{C}d\right) \;\geq\; 1 - \delta.$$

A corollary induced by Lemma A.2 is that

**Corollary A.3** (Sample Separation and Norm Control). Under Assumption 4.2 ($\epsilon$ satisfies the conditions in A.7), let $\epsilon_1, \ldots, \epsilon_n$ be i.i.d. copies of $\epsilon$. Fix $\delta \in (0,1)$ and define
$$\tau(\delta/2n) = \frac{2}{d}\log\left(\frac{4nB}{c_f\,\delta}\right),$$
$$y_l(\delta/2n) = (1+\tau(\delta/2n)) - \sqrt{\tau(\delta/2n)(2+\tau(\delta/2n))},$$
$$y_u(\delta/2n) = (1+\tau(\delta/2n)) + \sqrt{\tau(\delta/2n)(2+\tau(\delta/2n))}.$$

Then, with probability at least $1 - \delta$, the following holds for all pairs $i \neq j$:
$$\|\epsilon_i - \epsilon_j\|_2^2 \;\geq\; \frac{2\,y_l(\delta/2n)}{C}\,d \;-\; 2\sqrt{\frac{d}{c_0}\log(n^2/\delta)},$$

where $c_0 > 0$ is some constant depending on $C, C_1, C_2, C_3$. Additionally, within the same event, we have
$$\frac{y_l(\delta/2n)}{C}d \leq \|\epsilon_i\|_2^2 \leq \frac{y_u(\delta/2n)}{C}d, \text{ for } i = 1, 2, \cdots, n.$$

We defer the proofs of Lemma A.2 and Corollary A.3 to Appendix A.4.1.

We denote $q_t$ as the density of $\alpha_t\epsilon + \sigma_t Z$. We then provide some useful results that help us to derive useful properties of $q_t$.

**Lemma A.4** (Lemma B.1 and B.8, (Fu et al., 2024)). Let
$$\widehat{\sigma}_t = \frac{\sigma_t}{\left(\alpha_t^2 + C\sigma_t^2\right)^{1/2}}, \quad \widehat{\alpha}_t = \frac{\alpha_t}{\alpha_t^2 + C\sigma_t^2},$$

under sub-Gaussian Hölder density assumption, we have
$$q_t(x) = \frac{1}{\left(\alpha_t^2 + C\sigma_t^2\right)^{d/2}} \exp\left(-\frac{C\|x\|_2^2}{2(\alpha_t^2 + C\sigma_t^2)}\right) h(x,t),$$

where
$$h(x,t) = \int f(z) \frac{1}{(2\pi)^{d/2}\widehat{\sigma}_t^d} \exp\left(-\frac{\|z - \widehat{\alpha}_t x\|_2^2}{2\widehat{\sigma}_t^2}\right) dz, \text{ and } c_f \leq h(x,t) \leq B.$$

Equipped with this, it is also straightforward to obtain the following:

**Lemma A.5** (One–sided upper ratio bound for the channel). For any $x_1, x_2 \in \mathbb{R}^d$, we have

$$\frac{q_t(x_1)}{q_t(x_2)} \leq \frac{B}{c_f} \exp\left(-\frac{C(\|x_1\|_2^2 - \|x_2\|_2^2)}{2(\alpha_t^2 + C\sigma_t^2)}\right).$$

We finally present the following lemma to provide an estimation of $\mu_{0|t}^{(k)}(x_t)$.

**Lemma A.6** (Estimation of $\mu_{0|t}^{(k)}$). For any $t$ satisfying $\frac{\alpha_t}{\sigma_t} = \Omega(\sqrt{d})$, and $x_t = \Theta(\sqrt{d})$, we have

$$\mu_{0|t}^{(k)}(x_t) = \mu^{(k)} + \frac{\alpha_t}{\alpha_t^2 + C\sigma_t^2}(x_t - \alpha_t\mu^{(k)}) + \mathcal{O}\left(\sigma_t/\alpha_t\right),$$

where $E$, the error term, satisfies $\|E\|_2 = \mathcal{O}\left(\frac{\sigma_t}{\alpha_t}\right)$.

The proof is deferred to Appendix A.4.2.

## A.4 PROOF OF SUPPORTING LEMMAS

### A.4.1 PROOF OF LEMMA A.2 AND COROLLARY A.3

*Proof of Lemma A.2.* We define the function $h(x) = x - 1 - \log(x)$, which is positive for $x \neq 1$. The proof proceeds by first bounding the moment-generating function (MGF) of $\|\epsilon\|_2^2$ and then applying a Chernoff bound.

The normalization constant $Z$ is defined as $Z = \int_{\mathbb{R}^d} \exp(-C\|x\|_2^2/2)f(x)dx$. Leveraging $c_f \leq f \leq B$, we can bound $Z$ as

$$Z \geq \int_{\mathbb{R}^d} c_f \cdot \exp(-C\|x\|_2^2/2)dx = c_f \left(\frac{2\pi}{C}\right)^{d/2},$$

$$Z \leq \int_{\mathbb{R}^d} B \cdot \exp(-C\|x\|_2^2/2)dx = B \left(\frac{2\pi}{C}\right)^{d/2}.$$

Let $M(\lambda) = \mathbb{E}[e^{\lambda\|\epsilon\|_2^2}]$ be the MGF of $\|\epsilon\|_2^2$. For $\lambda > 0$:

$$
\begin{aligned}
M(\lambda) &= \int_{\mathbb{R}^d} e^{\lambda\|x\|_2^2} p_\epsilon(x)dx \\
&= \frac{1}{Z}\int_{\mathbb{R}^d} e^{\lambda\|x\|_2^2} \exp(-C\|x\|_2^2/2)f(x)dx \\
&= \frac{1}{Z}\int_{\mathbb{R}^d} \exp\left(-\frac{1}{2}(C - 2\lambda)\|x\|_2^2\right)f(x)dx.
\end{aligned}
$$

For the integral to converge, we require $C - 2\lambda > 0$, i.e., $\lambda < C/2$. Using the upper bound $f(x) \leq B$ and the lower bound on $Z$:

$$
\begin{aligned}
M(\lambda) &\leq \frac{B}{Z}\int_{\mathbb{R}^d} \exp\left(-\frac{1}{2}(C - 2\lambda)\|x\|_2^2\right)dx \\
&\leq \frac{B}{c_f\left(\frac{2\pi}{C}\right)^{d/2}}\left(\frac{2\pi}{C - 2\lambda}\right)^{d/2} \\
&= \frac{B}{c_f}\left(\frac{C}{C - 2\lambda}\right)^{d/2} = \frac{B}{c_f}\left(\frac{1}{1 - 2\lambda/C}\right)^{d/2}.
\end{aligned}
$$

**Part 1: Proof of the Upper Tail Bound.** We seek to bound $\mathbb{P}(\|\epsilon\|_2^2 \geq (1 + \eta)d)$. The Chernoff bound for an upper tail is $\mathbb{P}(X \geq a) \leq \inf_{\lambda>0} e^{-\lambda a}\mathbb{E}[e^{\lambda X}]$.

First, we bound the MGF $M(\lambda) = \mathbb{E}[e^{\lambda \|\epsilon\|_2^2}]$ for $\lambda > 0$. As shown above, this yields:

$$M(\lambda) \leq \frac{B}{c_f} \left(1 - \frac{2\lambda}{C}\right)^{-d/2}, \quad \text{for } 0 < \lambda < C/2.$$

Applying the Chernoff bound with $a = (1 + \eta)d$:

$$\mathbb{P}(\|\epsilon\|_2^2 \geq (1 + \eta)d) \leq \frac{B}{c_f} \inf_{0 < \lambda < C/2} \exp\left(-\lambda(1 + \eta)d - \frac{d}{2}\log(1 - 2\lambda/C)\right).$$

Minimizing the term in the exponent with respect to $\lambda$ yields the optimal value $\lambda^* = \frac{C}{2} - \frac{1}{2(1+\eta)}$. This choice is valid (i.e., $\lambda^* > 0$) if $\eta > 1/C - 1$.

Substituting $\lambda^*$ back into the exponent gives:

$$-\frac{d}{2}[C(1 + \eta) - 1 - \log(C(1 + \eta))] = -\frac{d}{2}h(C(1 + \eta)).$$

This completes the proof of the upper tail bound.

**Part 2: Proof of the Lower Tail Bound.** We seek to bound $\mathbb{P}(\|\epsilon\|_2^2 \leq (1 - \eta)d)$. The Chernoff bound for a lower tail is $\mathbb{P}(X \leq a) \leq \inf_{\lambda > 0} e^{\lambda a} \mathbb{E}[e^{-\lambda X}]$.

First, we bound the MGF for a negative argument, $M(-\lambda) = \mathbb{E}[e^{-\lambda \|\epsilon\|_2^2}]$ for $\lambda > 0$:

$$M(-\lambda) \leq \frac{B}{c_f} \left(1 + \frac{2\lambda}{C}\right)^{-d/2}.$$

Applying the Chernoff bound with $a = (1 - \eta)d$:

$$\mathbb{P}(\|\epsilon\|_2^2 \leq (1 - \eta)d) \leq \frac{B}{c_f} \inf_{\lambda > 0} \exp\left(\lambda(1 - \eta)d - \frac{d}{2}\log\left(1 + \frac{2\lambda}{C}\right)\right).$$

Minimizing the term in the exponent yields the optimal value $\lambda^* = \frac{1}{2}\left(\frac{1}{1-\eta} - C\right)$. This choice is valid (i.e., $\lambda^* > 0$) if $\eta > 1 - 1/C$.

Substituting this $\lambda^*$ back into the exponent gives:

$$-\frac{d}{2}[C(1 - \eta) - 1 - \log(C(1 - \eta))] = -\frac{d}{2}h(C(1 - \eta)).$$

This completes the proof of the lower tail bound.

**Part 3: High Probability Argument.** We finally derive a high probability argument for $\|\epsilon\|_2^2$. Set

$$\tau(\delta) := \frac{2}{d}\log\left(\frac{2B}{c_f \delta}\right), \qquad h(x) := x - 1 - \log x, \quad x > 0.$$

From the one–sided bounds,

$$\mathbb{P}(\|\epsilon\|_2^2 \geq (1 + \eta)d) \leq \frac{B}{c_f}\exp\left(-\frac{d}{2}h(C(1 + \eta))\right),$$

$$\mathbb{P}(\|\epsilon\|_2^2 \leq (1 - \eta)d) \leq \frac{B}{c_f}\exp\left(-\frac{d}{2}h(C(1 - \eta))\right).$$

Imposing each tail to be at most $\delta/2$ is ensured if

$$h(x) \geq \tau(\delta) \quad \text{with} \quad x = C(1 + \eta) \text{ (upper tail)}, \qquad x = C(1 - \eta) \text{ (lower tail)}.$$

Using $h(x) \geq \frac{(x-1)^2}{2x}$ for all $x > 0$, it suffices to require

$$\frac{(x - 1)^2}{2x} \geq \tau(\delta) \iff (x - 1)^2 \geq 2\tau(\delta)x \iff x^2 - 2(1 + \tau(\delta))x + 1 \geq 0.$$

The quadratic has roots

$$y_u(\delta) = (1 + \tau(\delta)) + \sqrt{\tau(\delta)(2 + \tau(\delta))}, \quad y_l(\delta) = (1 + \tau(\delta)) - \sqrt{\tau(\delta)(2 + \tau(\delta))},$$

with $0 < y_l(\delta) < 1 < y_u(\delta)$ (since $\tau(\delta) > 0$). Hence $x^2 - 2(1 + \tau(\delta))x + 1 \geq 0$ is equivalent to

$$x \in (-\infty, y_l(\delta)] \cup [y_u(\delta), \infty).$$

Applying this to each tail:

*Upper tail:* with $x = C(1 + \eta)$, it suffices that $C(1 + \eta) \geq y_u(\delta)$, i.e.

$$\eta \geq \eta_+^{\mathrm{exp}} := \frac{y_u(\delta)}{C} - 1.$$

*Lower tail:* with $x = C(1 - \eta)$, it suffices that $C(1 - \eta) \leq y_l(\delta)$, i.e.

$$\eta \geq \eta_-^{\mathrm{exp}} := 1 - \frac{y_l(\delta)}{C}.$$

Using a union bound with $\delta/2$ on each side yields the two–sided statement

$$\mathbb{P}\left( \frac{y_l(\delta)}{C} d \leq \|\epsilon\|_2^2 \leq \frac{y_u(\delta)}{C} d \right) \geq 1 - \delta,$$

equivalently,

$$(1 - \eta_-^{\mathrm{exp}}) d \leq \|\epsilon\|_2^2 \leq (1 + \eta_+^{\mathrm{exp}}) d,$$

with

$$\eta_-^{\mathrm{exp}} = 1 - \frac{y_l(\delta)}{C}, \qquad \eta_+^{\mathrm{exp}} = \frac{y_u(\delta)}{C} - 1, \qquad \tau(\delta) = \frac{2}{d} \log\left( \frac{2B}{c_f \delta} \right).$$

This finishes the proof. $\square$

*Proof of Corollary A.3.* The proof separately bounds the norms from below and the inner products from above.

From the statement in Lemma A.2, for each $i \in \{1, \ldots, n\}$,

$$\mathbb{P}\left( \frac{y_l(\delta/2n)}{C} d \leq \|\epsilon_i\|_2^2 \leq \frac{y_u(\delta/2n)}{C} d \right) \leq \frac{\delta}{2n}.$$

Let $\mathcal{A}$ be the event that $\frac{y_l(\delta/2n)}{C} d \leq \|\epsilon_i\|_2^2 \leq \frac{y_u(\delta/2n)}{C} d$ for all $i = 1, \ldots, n$. By a union bound over all $n$ samples, the probability of failure is at most $n \cdot \frac{\delta}{2n} = \frac{\delta}{2}$. Therefore, $\mathbb{P}(\mathcal{A}) \geq 1 - \delta/2$.

Here we introduce another lemma:

**Lemma A.7.** Suppose $\epsilon$ satisfies the conditions in A.7. Let $\epsilon_i, \epsilon_j$ be independent copies of $\epsilon$. Then for some universal constant $c_0 > 0$ which depends on $C, C_1, C_2, C_3$, we have

$$P(|\epsilon_i^\top \epsilon_j| \geq t) \leq 2 \exp\left\{ -\frac{c_0 t^2}{d} \right\}.$$

The proof is deferred to Appendix A.4.3.

Let $t_n := \sqrt{\frac{d}{c_0} \log(n^2/\delta)}$. Setting $t = t_n$ makes the tail probability for a single pair $(i, j)$ at most $\frac{\delta}{n^2}$. Let $\mathcal{B}$ be the event that $\epsilon_i^\top \epsilon_j \leq t_n$ for all $i \neq j$. By a union bound over all $\binom{n}{2}$ pairs, the probability of failure is at most $\binom{n}{2} \cdot \frac{\delta}{n^2} \leq \frac{\delta}{2}$. Thus, $\mathbb{P}(\mathcal{B}) \geq 1 - \delta/2$.

We now consider the event $\mathcal{A} \cap \mathcal{B}$, which holds with probability $\mathbb{P}(\mathcal{A} \cap \mathcal{B}) \geq 1 - \mathbb{P}(\mathcal{A}^c) - \mathbb{P}(\mathcal{B}^c) \geq 1 - \delta$. On this event, for all $i \neq j$:

$$
\begin{aligned}
\|\epsilon_i - \epsilon_j\|_2^2 &= \|\epsilon_i\|_2^2 + \|\epsilon_j\|_2^2 - 2\,\epsilon_i^\top \epsilon_j \\
&\geq \frac{y_l(\delta/2n)}{C} d + \frac{y_l(\delta/2n)}{C} d - 2t_n \\
&\geq \frac{2\,y_l(\delta/2n)}{C} d - 2\sqrt{\frac{d}{c_0} \log(n^2/\delta)}.
\end{aligned}
$$

Since this holds with probability at least $1 - \delta$, the claim follows. $\square$

### A.4.2 PROOF OF LEMMA A.6

*Proof of Lemma A.6.* By separating the mean and the random part of the original data $x_0$, we have

$$
\mu_{0|t}^{(k)}(x_t) = \frac{\int x_0 \exp(-\frac{1}{2\sigma_t^2}\|x_t - \alpha_t x_0\|_2^2) p^{(k)}(x_0)\mathrm{d}x_0}{\int \exp(-\frac{1}{2\sigma_t^2}\|x_t - \alpha_t x_0\|_2^2) p^{(k)}(x_0)\mathrm{d}x_0}
$$

$$
= \frac{\int (\epsilon + \mu^{(k)}) \exp(-\frac{1}{2\sigma_t^2}\|x_t - \alpha_t(\epsilon + \mu^{(k)})\|_2^2) p_\epsilon(\epsilon)\mathrm{d}\epsilon}{\int \exp(-\frac{1}{2\sigma_t^2}\|x_t - \alpha_t(\epsilon + \mu^{(k)})\|_2^2) p_\epsilon(\epsilon)\mathrm{d}\epsilon}
$$

Plugging in the expression of $p_\epsilon$, we have

$$
\exp\left(-\frac{1}{2\sigma_t^2}\|x_t - \alpha_t(\epsilon + \mu^{(k)})\|_2^2\right) p_\epsilon(\epsilon)
$$

$$
= \exp\left(-\frac{1}{2\sigma_t^2}\|x_t - \alpha_t(\epsilon + \mu^{(k)})\|_2^2 - \frac{C}{2}\|\epsilon\|_2^2 + \log f(\epsilon)\right)
$$

$$
= \exp\left(-\frac{1}{2\sigma_t^2}\left(\|x_t - \alpha_t\mu^{(k)}\|_2^2 - 2\alpha_t(x_t - \alpha_t\mu^{(k)})^\top\epsilon + \alpha_t^2\|\epsilon\|_2^2\right) - \frac{C}{2}\|\epsilon\|_2^2 + \log f(\epsilon)\right)
$$

$$
= \exp\left(-\frac{1}{2}\left(\frac{\alpha_t^2}{\sigma_t^2} + C\right)\|\epsilon\|_2^2 + \frac{\alpha_t}{\sigma_t^2}(x_t - \alpha_t\mu^{(k)})^\top\epsilon - \frac{1}{2\sigma_t^2}\|x_t - \alpha_t\mu^{(k)}\|_2^2 + \log f(\epsilon)\right)
$$

$$
= \exp\left(-\frac{\gamma_t^2}{2}\|\epsilon - \widetilde{\mu}_\epsilon\|_2^2 + \frac{\gamma_t^2}{2}\|\widetilde{\mu}_\epsilon\|_2^2 - \frac{1}{2\sigma_t^2}\|x_t - \alpha_t\mu^{(k)}\|_2^2 + \log f(\epsilon)\right)
$$

$$
= \exp(C(t, x_t)) \cdot \exp\left(-\frac{\gamma_t^2}{2}\|\epsilon - \widetilde{\mu}_\epsilon\|_2^2\right) f(\epsilon),
$$

where

$$
\gamma_t^2 := \frac{\alpha_t^2}{\sigma_t^2} + C,
$$

$$
\widetilde{\mu}_\epsilon := \frac{\alpha_t}{\sigma_t^2\gamma_t^2}(x_t - \alpha_t\mu^{(k)}),
$$

$$
C(t, x_t) := \frac{\gamma_t^2}{2}\|\widetilde{\mu}_\epsilon\|_2^2 - \frac{1}{2\sigma_t^2}\|x_t - \alpha_t\mu^{(k)}\|_2^2.
$$

By substituting the simplified kernel back into the expression for $\mu_{0|t}^{(k)}(x_t)$, the constant term $\exp(C(t, x_t))$ cancels from the numerator and denominator, yielding:

$$
\mu_{0|t}^{(k)}(x_t) = \frac{\int (\epsilon + \mu^{(k)}) \exp\left(-\frac{\gamma_t^2}{2}\|\epsilon - \widetilde{\mu}_\epsilon\|_2^2\right) f(\epsilon)\mathrm{d}\epsilon}{\int \exp\left(-\frac{\gamma_t^2}{2}\|\epsilon - \widetilde{\mu}_\epsilon\|_2^2\right) f(\epsilon)\mathrm{d}\epsilon}
$$

$$
= \mu^{(k)} + \frac{\int \epsilon \exp\left(-\frac{\gamma_t^2}{2}\|\epsilon - \widetilde{\mu}_\epsilon\|_2^2\right) f(\epsilon)\mathrm{d}\epsilon}{\int \exp\left(-\frac{\gamma_t^2}{2}\|\epsilon - \widetilde{\mu}_\epsilon\|_2^2\right) f(\epsilon)\mathrm{d}\epsilon}.
$$

This expression is the expectation of $\epsilon$ with respect to a new posterior distribution, whose unnormalized density is given by $q(\epsilon|x_t, k) \propto \exp\left(-\frac{\gamma_t^2}{2}\|\epsilon - \widetilde{\mu}_\epsilon\|_2^2\right) f(\epsilon)$. We provide a more rigorous justification for the approximation, starting from the exact expression for the posterior mean:

$$
\mu_{0|t}^{(k)}(x_t) = \mu^{(k)} + \mathbb{E}_{\epsilon \sim q}[\epsilon] = \mu^{(k)} + \widetilde{\mu}_\epsilon + \mathbb{E}_{\epsilon \sim q}[\epsilon - \widetilde{\mu}_\epsilon].
$$

Our goal is to analyze the term $\mathbb{E}_{\epsilon \sim q}[\epsilon - \widetilde{\mu}_\epsilon]$. Writing it as a ratio of integrals:

$$
\mathbb{E}_{\epsilon \sim q}[\epsilon - \widetilde{\mu}_\epsilon] = \frac{\int (\epsilon - \widetilde{\mu}_\epsilon) \exp\left(-\frac{\gamma_t^2}{2}\|\epsilon - \widetilde{\mu}_\epsilon\|_2^2\right) f(\epsilon)\mathrm{d}\epsilon}{\int \exp\left(-\frac{\gamma_t^2}{2}\|\epsilon - \widetilde{\mu}_\epsilon\|_2^2\right) f(\epsilon)\mathrm{d}\epsilon}.
$$

Let $\phi_{\widetilde{\mu}_\epsilon, \gamma_t^{-2}}(\epsilon) = \exp\left(-\frac{\gamma_t^2}{2}\|\epsilon - \widetilde{\mu}_\epsilon\|_2^2\right)$ denote the unnormalized Gaussian density. We apply multivariate integration by parts to the numerator, which yields the exact identity:

$$\int (\epsilon - \widetilde{\mu}_\epsilon) \phi_{\widetilde{\mu}_\epsilon, \gamma_t^{-2}}(\epsilon) f(\epsilon) \mathrm{d}\epsilon = \frac{1}{\gamma_t^2} \int \phi_{\widetilde{\mu}_\epsilon, \gamma_t^{-2}}(\epsilon) \nabla f(\epsilon) \mathrm{d}\epsilon.$$

Substituting this into our expression, and letting $Z$ be a random variable with density proportional to the Gaussian part, i.e., $Y \sim \mathcal{N}(\widetilde{\mu}_\epsilon, (\gamma_t^2)^{-1} I_d)$, we obtain the exact relation:

$$\mathbb{E}_{\epsilon \sim q}[\epsilon - \widetilde{\mu}_\epsilon] = \frac{1}{\gamma_t^2} \frac{\mathbb{E}_Y[\nabla f(Y)]}{\mathbb{E}_Y[f(Y)]},$$

and we further have

$$\|\mathbb{E}_{\epsilon \sim q}[\epsilon - \widetilde{\mu}_\epsilon]\|_2 \leq \frac{B\sqrt{d}}{\gamma_t^2 c_f}.$$

By the condition $\alpha_t/\sigma_t = \Omega(\sqrt{d})$, we finally have

$$\mu_{0|t}^{(k)}(x_t) = \underbrace{\mu^{(k)} + \frac{\alpha_t}{\alpha_t^2 + C\sigma_t^2}(x_t - \alpha_t \mu^{(k)})}_{\text{Gaussian Posterior Mean}} + \mathcal{O}(\sigma_t/\alpha_t),$$

and we complete the proof.

$\square$

### A.4.3 Proof of Lemma A.7

*Proof.* First, we rewrite the inner product as a bilinear form in terms of the independent vectors $\xi_i$ and $\xi_j$, which are entrywise independent sub-Gaussian random vectors with zero mean and unit variance as stated in Assumption 4.2:

$$\epsilon_i^\top \epsilon_j = (\Sigma^{1/2} \xi_i)^\top (\Sigma^{1/2} \xi_j) = \xi_i^\top \Sigma \xi_j.$$

The expression $\xi_i^\top \Sigma \xi_j$ is a bilinear form with a deterministic matrix $\Sigma$ and independent sub-gaussian vectors $\xi_i, \xi_j$. We can now directly apply the Hanson-Wright inequality (see Vershynin (2018) Theorem 6.2.2), which states that for any fixed matrix $A$:

$$P(|\xi_i^\top A \xi_j| \geq t) \leq 2\exp\left\{-C_0 \min\left(\frac{t^2}{C_1^4 \|A\|_F^2}, \frac{t}{C_1^2 \|A\|_{\text{op}}}\right)\right\},$$

for some constant $C_0 > 0$. By setting $A = \Sigma$ in the inequality and invoking our condition $\|\xi\|_{\psi_2} \leq C_1$, $\|\Sigma\|_F \leq C_2\sqrt{d}$, $\|\Sigma\|_2 \leq C_3$, we immediately arrive at the final bound:

$$P(|\epsilon_i^\top \epsilon_j| \geq t) \leq 2\exp\left\{-\frac{c_0 t^2}{d}\right\},$$

where $c_0 > 0$ is a constant depending on $C, C_0, C_1, C_2$. $\square$

## B Representing Empirical and Ground-truth Score Function using Deep Neural Networks

We follow the idea of network approximation in Fu et al. (2024) to build our proof.

We express the empirical score function as

$$\nabla \log \widehat{p}_t(x) = \frac{\nabla \widehat{p}_t(x)}{\widehat{p}_t(x)},$$

similarly for the ground-truth score function, and we approximate the numerator $\nabla \widehat{p}_t(x)$ and denominator $\widehat{p}_t(x)$ separately. To ensure uniform approximation, we restrict the domain of $x$ to a bounded set. In addition, we impose a lower threshold $\epsilon_{\text{low}}$ on $p_t(x)$ to prevent instability caused

by extremely small density values. Finally, within the overlapping regions of these two truncated domains, we employ ReLU networks for approximation.

We organize this section as follows. Appendix B.1 presents the main lemmas and propositions that form the foundation for the proof of Theorem 5.1, and uses these results to give a complete proof of Theorem 5.1. Appendix B.2 provides the proof of Proposition B.4, which establishes the network approximation of both the numerator $\nabla \widehat{p}_t(x)$ and the denominator $\widehat{p}_t(x)$. Appendix B.3 collects the proofs of the auxiliary lemmas used throughout this section. Finally, Appendix B.4 details the network architecture and analyzes the error propagation of the score approximation network.

### B.1 PROOF OF THEOREM 5.1

We begin by stating the main lemmas and propositions needed for the proof.

We first establish that the $\ell_\infty$-norm of the empirical score function can be bounded in terms of the $\ell_\infty$-norm of $x$. We denote $B_D = \max_{1 \leq i \leq n} \|x_i\|_\infty$.

**Lemma B.1.** The empirical score function satisfies

$$\|\nabla \log \widehat{p}_t(x)\|_\infty \leq \frac{\|x\|_\infty + B_D}{\sigma_t^2}.$$

The proof is provided in Appendix B.3.1. This lemma shows that the $\ell_\infty$-norm of the score function is controlled by both the input magnitude and the magnitude of the dataset.

Next, we establish some results on complement of the bounded domain of $x$.

**Lemma B.2.** Suppose $B > \max(2B_D, 2\sqrt{d})$. For a fixed time $t \in [0, T]$, it holds that

$$\int_{\|x\|_\infty > B} \|\nabla \log \widehat{p}_t(x)\|_2^2 \widehat{p}_t(x) dx \lesssim \frac{1}{\sigma_t^4} B^d \exp\left(-\frac{B^2}{8}\right),$$

$$\int_{\|x\|_\infty > B} \widehat{p}_t(x) dx \lesssim \frac{1}{\sigma_t^4} B^{d-2} \exp\left(-\frac{B^2}{8}\right).$$

The proof is given in Appendix B.3.2. Lemma B.2 follows from the light-tailed nature of the empirical distribution, which ensures exponential decay outside the bounded domain.

In a similar fashion, we show that analogous bounds hold when the empirical density $\widehat{p}_t$ is truncated by a threshold.

**Lemma B.3.** For any $B > 2B_D$ and $\epsilon_{\text{low}} > 0$, we have

$$\int_{\|x\|_\infty \leq B} \mathbb{1}\left\{|\widehat{p}_t(x)| < \epsilon_{\text{low}}\right\} \widehat{p}_t(x) \, dx \lesssim B^d \, \epsilon_{\text{low}}, \tag{B.1}$$

$$\int_{\|x\|_\infty \leq B} \mathbb{1}\left\{|\widehat{p}_t(x)| < \epsilon_{\text{low}}\right\} \|\nabla \log \widehat{p}_t(x)\|_2^2 \widehat{p}_t(x) \, dx \lesssim \frac{\epsilon_{\text{low}}}{\sigma_t^4} B^{d+2}. \tag{B.2}$$

The proof is provided in Appendix B.3.3.

By combining Lemmas B.2 and B.3, we complete the truncation step. We introduce our network approximation result in Proposition B.4.

**Proposition B.4.** Suppose that the density function of $P_{\text{data}}$ satisfies the sub-Gaussian Hölder density condition in Definition 3.2. For any sufficiently small $\epsilon > 0$. Define the early-stopping time $t_0$ satisfying $\log t_0 = \mathcal{O}(\log \epsilon)$ and the terminal time $T = \mathcal{O}(\log \epsilon^{-1})$. We constrain $x \in [-2\sqrt{2 \log \epsilon^{-1}}, 2\sqrt{2 \log \epsilon^{-1}}]^d$. Then there exist ReLU neural network architectures $\mathcal{F}_1(W_1, L_1, N_1)$, such that $\exists \widehat{s} \in \mathcal{F}_1(W_1, L_1, N_1)$ satisfying for all $t \in [t_0, T]$

$$\widehat{p}_t(x)\|\nabla \log \widehat{p}_t(x) - \widehat{s}(x, t)\|_\infty \lesssim \frac{\epsilon}{\sigma_t^2}.$$

The configuration of $\mathcal{F}_1$ is

$$L = \mathcal{O}(\log^2 \epsilon^{-1}), \quad W = \mathcal{O}(n \log^3 \epsilon^{-1}), \quad N = \mathcal{O}(n \log^4 \epsilon^{-1}).$$

The proof is provided in Appendix B.2.

Now we start to prove the approximation bound for empirical distribution. We claim $\widehat{s}(x, t)$ is a $L_2(\widehat{P}_t)$ approximator of the score fucntion. In order to prove it, we choose $B = 2\sqrt{2 \log \epsilon^{-1}}$, and $\epsilon_{\mathrm{low}} = 4\epsilon$. We decompose the score approxiamtion error into three parts

$$\int_{\mathbb{R}^d} \left\| \widehat{s}(x, t) - \nabla \log \widehat{p}_t(x) \right\|_2^2 \widehat{p}_t(x) \, dx$$

$$= \underbrace{\int_{\|x\|_\infty > B} \left\| \widehat{s}(x, t) - \nabla \log \widehat{p}_t(x) \right\|_2^2 \widehat{p}_t(x) \, dx}_{(D_1)}$$

$$+ \underbrace{\int_{\|x\|_\infty \leq B} \mathbb{1}\left\{ |\widehat{p}_t(x)| < \epsilon_{\mathrm{low}} \right\} \left\| \widehat{s}(x, t) - \nabla \log \widehat{p}_t(x) \right\|_2^2 \widehat{p}_t(x) \, dx}_{(D_2)}$$

$$+ \underbrace{\int_{\|x\|_\infty \leq B} \mathbb{1}\left\{ |\widehat{p}_t(x)| \geq \epsilon_{\mathrm{low}} \right\} \left\| \widehat{s}(x, t) - \nabla \log \widehat{p}_t(x) \right\|_2^2 \widehat{p}_t(x) \, dx}_{(D_3)}.$$

We bound three parts separately.

**Bounding $D_1$** By Proposition B.4, we know $\|\widehat{s}(x, t)\|_\infty \leq \frac{2\sqrt{2 \log \epsilon^{-1}} + B_D}{\sigma_t^2}$

$$\int_{\|x\|_\infty > B} \left\| \widehat{s}(x, t) - \nabla \log \widehat{p}_t(x) \right\|_2^2 \widehat{p}_t(x) \, dx$$

$$\leq \int_{\|x\|_\infty > B} \left( 2\|\widehat{s}(x, t)\|_2^2 + 2\|\nabla \log \widehat{p}_t(x)\|_2^2 \right) \widehat{p}_t(x) \, dx$$

$$\lesssim \frac{1}{\sigma_t^4} (\log \epsilon^{-1})^{d/2} \epsilon. \tag{B.3}$$

We invoke Lemma B.2 in the second inequality.

**Bounding $D_2$** Similar to what we did in bounding $D_1$, we have

$$\int_{\|x\|_\infty \leq B} \mathbb{1}\left\{ |\widehat{p}_t(x)| < \epsilon_{\mathrm{low}} \right\} \left\| \widehat{s}(x, t) - \nabla \log \widehat{p}_t(x) \right\|_2^2 \widehat{p}_t(x) \, dx$$

$$\leq \int_{\|x\|_\infty \leq B} \left( 2\|\widehat{s}(x, t)\|_2^2 + 2\|\nabla \log \widehat{p}_t(x)\|_2^2 \right) \mathbb{1}\left\{ |\widehat{p}_t(x)| < \epsilon_{\mathrm{low}} \right\} \widehat{p}_t(x) \, dx$$

$$\lesssim \frac{\epsilon_{\mathrm{low}}}{\sigma_t^4} (\log \epsilon^{-1})^{d/2+1}. \tag{B.4}$$

We invoke Lemma B.3 in the second inequality.

**Bounding $D_3$** By Proposition B.4, we have

$$\int_{\|x\|_\infty \leq B} \mathbb{1}\left\{ |\widehat{p}_t(x)| \geq \epsilon_{\mathrm{low}} \right\} \left\| \widehat{s}(x, t) - \nabla \log \widehat{p}_t(x) \right\|_2^2 \widehat{p}_t(x) \, dx$$

$$\leq \int_{\|x\|_\infty \leq B} \mathbb{1}\left\{ |\widehat{p}_t(x)| \geq \epsilon_{\mathrm{low}} \right\} d\|\widehat{s}(x, t) - \nabla \log \widehat{p}_t(x)\|_\infty^2 \widehat{p}_t(x) \, dx$$

$$\lesssim \int_{\|x\|_\infty \leq B} \mathbb{1}\left\{ |\widehat{p}_t(x)| \geq \epsilon_{\mathrm{low}} \right\} \frac{d}{\widehat{p}_t(x)\sigma_t^4} \epsilon^2 \, dx$$

$$= \frac{\epsilon^2}{\epsilon_{\mathrm{low}}} \int_{\|x\|_\infty \leq B} \mathbb{1}\left\{ |\widehat{p}_t(x)| \geq \epsilon_{\mathrm{low}} \right\} \frac{d\epsilon_{\mathrm{low}}}{\widehat{p}_t(x)\sigma_t^4} \, dx$$

$$\lesssim \frac{\epsilon^2}{\epsilon_{\mathrm{low}}\sigma_t^4} (\log \epsilon^{-1})^{d/2}. \tag{B.5}$$

Combining (B.3), (B.4) and (B.5) together gives us

$$\int_{\mathbb{R}^d} \left\| \widehat{s}(x,t) - \nabla \log \widehat{p}_t(x) \right\|_2^2 \widehat{p}_t(x)\, dx$$

$$\lesssim \frac{1}{\sigma_t^4}(\log \epsilon^{-1})^{d/2}\epsilon + \frac{\epsilon}{\sigma_t^4}(\log \epsilon^{-1})^{d/2+1} + \frac{\epsilon}{\sigma_t^4}(\log \epsilon^{-1})^{d/2}$$

$$\lesssim \frac{\epsilon}{\sigma_t^4}(\log \epsilon^{-1})^{d/2+1}, \tag{B.6}$$

here we plug in $\epsilon_{\text{low}} = 4\epsilon$.

Set $\epsilon' = C_\epsilon \epsilon(\log \epsilon^{-1})^{d/2+1}$, where $C_\epsilon$ represents the constant hidden in $\lesssim$ in (B.6). Also, when $\epsilon$ goes to zero, $\epsilon'$ will go to zero. Then we immediately derive

$$\int_{\mathbb{R}^d} \left\| \widehat{s}(x,t) - \nabla \log \widehat{p}_t(x) \right\|_2^2 \widehat{p}_t(x)\, dx \lesssim \frac{\epsilon'}{\sigma_t^4},$$

it implies

$$\mathbb{E}_{\mathcal{D}}\left[ \mathbb{E}_{x \sim \widehat{P}_t}\left[ \left\| \widehat{s}(x,t) - \nabla \log \widehat{p}_t(x) \right\|_2^2 \right] \right] \lesssim \frac{\epsilon'}{\sigma_t^4},$$

The network configuration of the entire network architecture satisfies

$$W = \widetilde{\mathcal{O}}\big(n \log^3(\epsilon')^{-1}\big), \qquad L = \widetilde{\mathcal{O}}\big(\log^2(\epsilon')^{-1}\big), \qquad N = \widetilde{\mathcal{O}}\big(n \log^4(\epsilon')^{-1}\big).$$

For the approximation of ground-truth score function, we apply the Theorem 3.4 in Fu et al. (2024) with $d_y = 0$.

**Theorem B.5.** (Theorem 3.4 in Fu et al. (2024)) Suppose $P_{\text{data}}$ has a sub-Gaussian Hölder density with Hölder index $\beta$. For sufficiently large $N_1$ and constants $C_\sigma, C_\alpha > 0$, by taking the early-stopping time $t_0 = N_1^{-C_\sigma}$ and the terminal time $T = C_\alpha \log N_1$, there exists

$$s \in \mathcal{F}\big(W, L, N\big)$$

such that for any $t \in [t_0, T]$, it holds that

$$\int_{\mathbb{R}^d} \left\| s(x,t) - \nabla \log p_t(x) \right\|_2^2 p_t(x)\, \mathrm{d}x = \mathcal{O}\left( \frac{1}{\sigma_t^2} \cdot N_1^{-\frac{2\beta}{d}} \cdot (\log N_1)^{\beta+1} \right). \tag{B.7}$$

The hyperparameters in the ReLU neural network class $\mathcal{F}$ satisfy

$$W = \mathcal{O}\left( N_1 \log^7 N_1 \right), \qquad L = \mathcal{O}\left( \log^4 N_1 \right), \qquad N = \mathcal{O}\left( N_1 \log^9 N_1 \right). \tag{B.8}$$

We set $\epsilon_{\text{true}} = C'_\epsilon \cdot N_1^{-\frac{2\beta}{d}} \cdot (\log N_1)^{\beta+1}$, where $C'_\epsilon$ denote the constant hidden by $\mathcal{O}$, when $N$ is sufficiently large, $\epsilon_{\text{true}}$ will be sufficiently small. Then we immediately have

$$\int_{\mathbb{R}^d} \left\| s(x,t) - \nabla \log p_t(x) \right\|_2^2 p_t(x)\, \mathrm{d}x \leq \frac{\epsilon_{\text{true}}}{\sigma_t^2}.$$

Namely

$$\mathbb{E}_{\mathcal{D}}\left[ \mathbb{E}_{X_t \sim \widehat{P}_t}\left[ \left\| s(X_t,t) - \nabla \log p_t(X_t) \right\|_2^2 \right] \right] \leq \frac{\epsilon_{\text{true}}}{\sigma_t^2}.$$

The network configuration is

$$W_2 = \widetilde{\mathcal{O}}\left( (\epsilon_{\text{true}})^{-\frac{d}{2\beta}} \log^7 \epsilon_{\text{true}}^{-1} \right), \qquad L_2 = \widetilde{\mathcal{O}}\big( \log^4 \epsilon_{\text{true}}^{-1} \big), \qquad N_2 = \widetilde{\mathcal{O}}\left( (\epsilon_{\text{true}})^{-\frac{d}{2\beta}} \log^9 \epsilon_{\text{true}}^{-1} \right).$$

We complete our proof.

## B.2 Proof of Proposition B.4

We denote the first coordinate of a vector $x \in R^d$ as $[x]_1$. Without loss of generality, we focus on the $j$-th coordinate of the empirical score function. The explicit form of it is

$$[\nabla \log \widehat{p}_t(x)]_j = \frac{1}{\sigma_t} \frac{\overbrace{\left[\sum_{i=1}^{n} \frac{1}{n} \frac{(\alpha_t x_i - x)}{\sigma_t} \exp\left(-\frac{1}{2\sigma_t^2}\|x - \alpha_t x_i\|_2^2\right)\right]_j}^{D_5}}{\underbrace{\sum_{i=1}^{n} \frac{1}{n} \exp\left(-\frac{1}{2\sigma_t^2}\|x - \alpha_t x_i\|_2^2\right)}_{D_4}}.$$

We approximate the denominator $D_4$ and numerator $D_5$ with ReLU networks, and subsequently combine these approximations to construct a score estimator.

**Lemma B.6.** (ReLU approximation of $D_4$) For any sufficiently small $\epsilon_{f_1} > 0$, there exists a ReLU network architecture $\mathcal{F}(W, L, N)$, such that $\exists f_1^{\text{ReLU}}(x, t) \in \mathcal{F}$ satisfying

$$\left| \sum_{i=1}^{n} \frac{1}{n} \exp\left(-\frac{1}{2\sigma_t^2}\|x - \alpha_t x_i\|_2^2\right) - f_1^{\text{ReLU}}(x, t) \right| \leq \epsilon_{f_1}, \qquad (\text{B}.9)$$

for any $x \in \left[-2\sqrt{2\log \epsilon_{f_1}^{-1}}, 2\sqrt{2\log \epsilon_{f_1}^{-1}}\right]^d$, and $t \in [t_0, T]$, where $\log t_0 = \mathcal{O}(\log \epsilon_{f_1})$, and $T = \mathcal{O}(\log \epsilon_{f_1}^{-1})$, and the network configuration is

$$L = \mathcal{O}(\log^2 \epsilon_{f_1}^{-1}), \quad W = \mathcal{O}(n \log^3 \epsilon_{f_1}^{-1}), \quad N = \mathcal{O}(n \log^4 \epsilon_{f_1}^{-1}).$$

The proof is provided in Appendix B.3.4. We also have the following result to approximate $D_5$.

**Lemma B.7.** (ReLU approximation of $D_5$) For any sufficiently small $\epsilon_{f_2} > 0$, and $j \in [d]$, there exists a ReLU network architecture $\mathcal{F}_j(W, L, N)$, such that $\exists f_2^{\text{ReLU}}(x, t, j) \in \mathcal{F}_j$ satisfying

$$\left| \sum_{i=1}^{n} \frac{1}{n} \frac{[\alpha_t x_i - x]_j}{\sigma_t} \exp\left(-\frac{1}{2\sigma_t^2}\|x - \alpha_t x_i\|_2^2\right) - f_2^{\text{ReLU}}(x, t, j) \right| \leq \epsilon_{f_2}, \qquad (\text{B}.10)$$

for any $x \in \left[-2\sqrt{2\log \epsilon_{f_2}^{-1}}, 2\sqrt{2\log \epsilon_{f_2}^{-1}}\right]^d$, and $t \in [t_0, T]$, where $\log t_0 = \mathcal{O}(\log \epsilon_{f_2})$, and $T = \mathcal{O}(\log \epsilon_{f_2}^{-1})$, and the network configuration is

$$L = \mathcal{O}(\log^2 \epsilon_{f_2}^{-1}), \quad W = \mathcal{O}(n \log^3 \epsilon_{f_2}^{-1}), \quad N = \mathcal{O}(n \log^4 \epsilon_{f_2}^{-1}).$$

The proof is provided in Appendix B.3.5. Now we are ready to finish the proof.

*Proof.* Let $\epsilon_{\text{low}} = 4\epsilon$, and set $\epsilon_{f_1} = \epsilon_{f_2} = \epsilon$. Then when $\widehat{p}_t(x) > \epsilon_{\text{low}}$, we have $f_1^{\text{ReLU}}(x, t) > \frac{1}{2}\widehat{p}_t(x)$. Using Lemmas B.6 and B.7, we denote the clipped version of $f_1$ by $f_{1,\text{clip}} = \max(f_1^{\text{ReLU}}, \epsilon_{\text{low}})$, and for $j \in [d]$, define the score approximator as

$$f_3(x, t, j) = \min\left(\frac{f_2^{\text{ReLU}}(x, t, j)}{\sigma_t f_{1,\text{clip}}(x, t)}, \frac{2\sqrt{2\log \epsilon^{-1}} + B_D}{\sigma_t^2}\right)$$

By the definition of $f_3(x, t, j)$, we know $|f_3(x, t, j)| \lesssim \frac{2\sqrt{2\log \epsilon^{-1}} + B_D}{\sigma_t^2}$, this actually matches the upper bound of $\|\nabla \log \widehat{p}_t(x)\|_\infty$ when $\|x\|_\infty \leq B$. Next, we bound the difference between $[\nabla \log \widehat{p}_t(x)]_j$ and $f_3(x, t, j)$

$$|[\nabla \log \widehat{p}_t(x)]_j - f_3(x, t, j)| \leq \left|[\nabla \log \widehat{p}_t(x)]_j - \frac{f_2^{\text{ReLU}}(x, t, j)}{\sigma_t f_{1,\text{clip}}(x, t)}\right|$$

$$\leq \left| \frac{[\nabla \widehat{p}_t(x)]_j}{\widehat{p}_t(x)} - \frac{[\nabla \widehat{p}_t(x)]_j}{f_{1,\text{clip}}(x,t)} \right| + \left| \frac{[\nabla \widehat{p}_t(x)]_j}{f_{1,\text{clip}}(x,t)} - \frac{f_2^{\text{ReLU}}(x,t,j)}{\sigma_t f_{1,\text{clip}}(x,t)} \right|$$

$$\leq [\nabla \widehat{p}_t(x)]_j \left| \frac{1}{\widehat{p}_t(x)} - \frac{1}{f_{1,\text{clip}}(x,t)} \right|$$

$$+ \frac{\left| \sigma_t [\nabla \widehat{p}_t(x)]_j - \sigma_t f_2^{\text{ReLU}}(x,t,j) \right|}{\sigma_t f_{1,\text{clip}}(x,t)}.$$

From $\|\nabla \log \widehat{p}_t(x)\|_\infty \leq \frac{2\sqrt{2\log \epsilon^{-1}} + B_D}{\sigma_t^2}$, we derive $[\nabla \widehat{p}_t(x)]_j \leq \frac{B+B_D}{\sigma_t^2} \widehat{p}_t$, for $\widehat{p}_t \geq \epsilon_{\text{low}}$, we have

$$|[\nabla \log \widehat{p}_t(x)]_j - f_3(x,t,j)|$$

$$\leq \frac{2\sqrt{2\log \epsilon^{-1}} + B_D}{\sigma_t^2} \widehat{p}_t \left| \frac{1}{\widehat{p}_t(x)} - \frac{1}{f_{1,\text{clip}}} \right| + \frac{\left| \sigma_t [\nabla \widehat{p}_t(x)]_j - \sigma_t f_2^{\text{ReLU}}(x,t,j) \right|}{\sigma_t f_{1,\text{clip}}}$$

$$\lesssim \frac{1}{f_{1,\text{clip}}} \left( \frac{\left( 2\sqrt{2\log \epsilon^{-1}} + B_D \right) |\widehat{p}_t(x) - f_{1,\text{clip}}|}{\sigma_t^2} + \frac{\left| [\nabla \widehat{p}_t(x)]_j - f_2^{\text{ReLU}}(x,t,j) \right|}{\sigma_t} \right)$$

$$\lesssim \frac{2\sqrt{2\log \epsilon^{-1}}\epsilon}{\widehat{p}_t \sigma_t^2}.$$

Then we can obtain a mapping $\mathbf{f}_3(x,t)$ to approximate $\nabla \log \widehat{p}_t(x)$

$$\|\nabla \log \widehat{p}_t(x) - \mathbf{f}_3(x,t)\|_\infty \leq \frac{2\sqrt{2\log \epsilon^{-1}}\epsilon}{\widehat{p}_t \sigma_t^2}.$$

Here $\mathbf{f}_3(x,t)$ is defined as

$$\mathbf{f}_3(x,t) = [f_3(x,t,1), f_3(x,t,2), ... f_3(x,t,d)]^\top.$$

We now construct a ReLU network $\mathbf{f}_3^{\text{ReLU}}(x,t)$ to approximate $\mathbf{f}_3(x,t)$, namely

$$\left\| \mathbf{f}_3(x,t) - \mathbf{f}_3^{\text{ReLU}}(x,t) \right\|_\infty \leq \epsilon.$$

Given ReLU realizations $f_1$ and $f_2$, we build upon them by implementing the following basic operations via ReLU networks: the inverse function, the product function, a ReLU-based approximation of $\sigma_t$, and entrywise $\min/\max$ operators. Details on determining the network size and analyzing error propagation are deferred to the Appendix B.4. Once we construct $\mathbf{f}_3^{\text{ReLU}}(x,t)$, we have

$$\widehat{p}_t(x)\|\nabla \log \widehat{p}_t(x) - \mathbf{f}_3^{\text{ReLU}}(x,t)\|_\infty \lesssim \frac{\epsilon}{\sigma_t^2}.$$

where $\mathbf{f}_3^{\text{ReLU}}(x,t) \in \mathcal{F}_{f_3}$, the network configuration of $\mathcal{F}_{f_3}$ satisfies

$$L = \mathcal{O}(\log^2 \epsilon^{-1}), \quad W = \mathcal{O}(n\log^3 \epsilon^{-1}), \quad N = \mathcal{O}(n\log^4 \epsilon^{-1}).$$

We complete our proof. $\qquad\square$

## B.3 PROOF OF LEMMAS

### B.3.1 PROOF OF LEMMA B.1

*Proof.*

$$\|\nabla \log \widehat{p}_t(x)\|_\infty = \frac{1}{\sigma_t^2} \frac{\sum_{i=1}^n \|x - \alpha_t x_i\|_\infty \exp\left(-\frac{1}{2\sigma_t^2}\|x - \alpha_t x_i\|_2^2\right)}{\sum_{i=1}^n \exp\left(-\frac{1}{2\sigma_t^2}\|x - \alpha_t x_i\|_2^2\right)}$$

$$\leq \frac{1}{\sigma_t^2} \frac{\sum_{i=1}^n \left( (\|x\|_\infty + \|\alpha_t x_i\|_\infty) \exp\left(-\frac{1}{2\sigma_t^2}\|x - \alpha_t x_i\|_2^2\right) \right)}{\sum_{i=1}^n \exp\left(-\frac{1}{2\sigma_t^2}\|x - \alpha_t x_i\|_2^2\right)}$$

$$\leq \frac{\|x\|_\infty + B_D}{\sigma_t^2}.$$

$\qquad\square$

### B.3.2   PROOF OF LEMMA B.2

*Proof.* We first prove the inequality for the score function.

$$\int_{\|x\|_\infty > B} \|\nabla \log \widehat{p}_t(x)\|_2^2 \widehat{p}_t(x) dx$$

$$= \sum_{i=1}^n \frac{1}{n} \frac{1}{\sigma_t^d (2\pi)^{d/2}} \int_{\|x\|_\infty > B} \|\nabla \log \widehat{p}_t(x)\|_2^2 \exp\left(-\frac{\|x - \alpha_t x_i\|_2^2}{2\sigma_t^2}\right) dx.$$

We only need to bound this term

$$\frac{1}{\sigma_t^d (2\pi)^{d/2}} \int_{\|x\|_\infty > B} \|\nabla \log \widehat{p}_t(x)\|_2^2 \exp\left(-\frac{\|x - \alpha_t x_i\|_2^2}{2\sigma_t^2}\right) dx.$$

By applying Lemma B.1, we have

$$\frac{1}{\sigma_t^d (2\pi)^{d/2}} \int_{\|x\|_\infty > B} \|\nabla \log \widehat{p}_t(x)\|_2^2 \exp\left(-\frac{\|x - \alpha_t x_i\|_2^2}{2\sigma_t^2}\right) dx$$

$$\leq \frac{d}{\sigma_t^{d+4}(2\pi)^{d/2}} \int_{\|x\|_\infty > B} (\|x\|_\infty + B_D)^2 \exp\left(-\frac{\|x - \alpha_t x_i\|_2^2}{2\sigma_t^2}\right) dx$$

$$\leq \frac{d}{\sigma_t^{d+4}(2\pi)^{d/2}} \int_{\|x\|_2 > B} (\|x\|_2 + B_D)^2 \exp\left(-\frac{\|x - \alpha_t x_i\|_2^2}{2\sigma_t^2}\right) dx$$

$$= \frac{d}{\sigma_t^4 (2\pi)^{d/2}} \int_{\|\sigma_t \xi_i + \alpha_t x_i\|_2 > B} (\|\sigma_t \xi_i + \alpha_t x_i\|_2 + B_D)^2 \exp\left(-\frac{\|\xi_i\|_2^2}{2}\right) d\xi_i$$

$$\leq \frac{d}{\sigma_t^4 (2\pi)^{d/2}} \int_{\|\xi_i\|_2 > (B - B_D)/\sigma_t} (\|\sigma_t \xi_i\|_2 + 2B_D)^2 \exp\left(-\frac{\|\xi_i\|_2^2}{2}\right) d\xi_i$$

$$= \frac{d}{\sigma_t^4 (2\pi)^{d/2}} \int_{r > (B - B_D)/\sigma_t} \int_\omega (\sigma_t r + 2B_D)^2 \exp\left(-\frac{r^2}{2}\right) r^{d-1} dr d\omega. \tag{B.11}$$

The third inequality follows from the change of variable $\xi_i = \frac{x - \alpha_t x_i}{\sigma_t}$. The last equality follows from changing variables to spherical coordinates. Next, we consider give a upper bound for (B.11), we derive it by firstly substituting $r$ with $m = r^2$, then (B.11) becomes

$$\frac{d}{\sigma_t^4 (2\pi)^{d/2}} \int_{r > (B - B_D)/\sigma_t} \int_\omega (\sigma_t r + 2B_D)^2 \exp\left(-\frac{r^2}{2}\right) r^{d-1} dr d\omega \tag{B.12}$$

$$= \frac{d}{\sigma_t^4 (2\pi)^{d/2}} \int_{m > (B - B_D)^2/\sigma_t^2} \int_\omega (\sigma_t^2 m + 4\sigma_t B_D \sqrt{m} + 4B_D^2) \exp\left(-\frac{m}{2}\right) \frac{m^{\frac{d-2}{2}}}{2} dm d\omega. \tag{B.13}$$

We bound this integral using Theorem 1.1 and Proposition 2.6 in (Pinelis, 2020).

**Lemma B.8.** Let $G_a(x)$ be defined as

$$G_a(x) := \begin{cases} x^{-2} e^{-x}, & \text{if } a = -1, \\ \dfrac{(x + b_a)^a - x^a}{a b_a} e^{-x}, & \text{if } a \in (-1, \infty) \setminus \{0\}, \\ e^{-x} \log \dfrac{x+1}{x}, & \text{if } a = 0. \end{cases}$$

where

$$b_a := \begin{cases} \Gamma(a + 1)^{1/(a-1)}, & \text{if } a \in (-1, \infty) \setminus \{1\}, \\ e^{1-\gamma}, & \text{if } a = 1, \end{cases}$$

and $\gamma$ is the Euler constant.

Then, for $-1 \leq a \leq 1$, it holds that

$$\int_x^\infty t^{a-1} e^{-t} dt \leq G_a(x).$$

Moreover, for any real $a > 1$, we have

$$\int_x^\infty t^{a-1}e^{-t}dt \leq \frac{x^{a-1}e^{-x}}{1 - \frac{a-1}{x}}, \qquad \text{for all real } x > a - 1.$$

By applying Lemma B.8, we obtain the following estimates. When $a = 0$, one has

$$\int_x^\infty t^{a-1}e^{-t}dt \leq G_a(x) \leq x^{-a}e^{-x}, \qquad x > 0, \tag{B.14}$$

since $\log\left(\frac{1+x}{x}\right) \leq \frac{1}{x}$. For $a \in (-1, 1] \setminus \{0\}$, it holds that

$$\int_x^\infty t^{a-1}e^{-t}dt \leq G_a(x) \lesssim x^{a-1}e^{-x}. \tag{B.15}$$

Furthermore, for $a > 1$ and $x > a - 1$, we have

$$\int_x^\infty t^{a-1}e^{-t}dt \leq \frac{x^{a-1}e^{-x}}{1 - \frac{a-1}{x}} \lesssim x^{a-1}e^{-x}. \tag{B.16}$$

Combining (B.14) ,(B.15), (B.16) and (B.13) together, we can conclude, when $B > \max(2B_D, 2\sqrt{d})$,

$$\frac{1}{\sigma_t^d(2\pi)^{d/2}}\int_{\|x\|_\infty > B} \|\nabla \log \widehat{p}_t(x)\|_2^2 \exp\left(-\frac{\|x - \alpha_t x_i\|_2^2}{2\sigma_t^2}\right)$$

$$\leq \frac{d}{\sigma_t^4(2\pi)^{d/2}}\int_{m > (B-B_D)^2/\sigma_t^2}\int_\omega (\sigma_t^2 m + 4\sigma_t B_D\sqrt{m} + 4B_D^2)\exp\left(-\frac{m}{2}\right)\frac{m^{\frac{d-2}{2}}}{2}dmd\omega$$

$$\lesssim \frac{1}{\sigma_t^4}\int_{m > (B-B_D)^2/\sigma_t^2}\int_\omega (\sigma_t^2 m + 4\sigma_t B_D\sqrt{m} + 4B_D^2)\exp\left(-\frac{m}{2}\right)\frac{m^{\frac{d-2}{2}}}{2}dmd\omega$$

$$\lesssim \frac{1}{\sigma_t^4}\int_{m > B^2/4}\int_\omega (\sigma_t^2 m + 4\sigma_t B_D\sqrt{m} + 4B_D^2)\exp\left(-\frac{m}{2}\right)\frac{m^{\frac{d-2}{2}}}{2}dmd\omega$$

$$\lesssim \frac{1}{\sigma_t^4}B^d\exp\left(-\frac{B^2}{8}\right).$$

Then we can conclude

$$\int_{\|x\|_\infty > B} \|\nabla \log \widehat{p}_t(x)\|_2^2 \widehat{p}_t(x)dx$$

$$\lesssim \frac{1}{\sigma_t^4}B^d\exp\left(-\frac{B^2}{8}\right).$$

Similarly we have

$$\int_{\|x\|_\infty > B} \widehat{p}_t(x)dx$$

$$\lesssim \sum_{i=1}^n \frac{1}{n}\frac{1}{\sigma_t^{d+4}}\int_{\|x\|_2 > B}\exp\left(-\frac{\|x - \alpha_t x_i\|_2^2}{2\sigma_t^2}\right)dx$$

$$\lesssim \frac{1}{\sigma_t^4}B^{d-2}\exp\left(-\frac{B^2}{8}\right).$$

$\square$

### B.3.3  PROOF OF LEMMA B.3

*Proof.* For the first inequality, we have

$$\int_{\|x\|_\infty \leq B} \mathbb{1}\{|\widehat{p}_t(x)| < \epsilon_{\text{low}}\}\,\widehat{p}_t(x)\,dx$$

$$\leq \int_{\|x\|_\infty \leq B} \epsilon_{\text{low}}\, dx$$

$$\lesssim B^d \epsilon_{\text{low}}.$$

For the second inequality, by Lemma B.1, we have

$$\int_{\|x\|_\infty \leq B} \mathbb{1}\{|\widehat{p}_t(x)| < \epsilon_{\text{low}}\} \|\nabla \log \widehat{p}_t(x)\|_2^2 \widehat{p}_t(x)\, dx$$

$$\leq \frac{1}{\sigma_t^4} \int_{\|x\|_\infty \leq B} \epsilon_{\text{low}} (\|x\|_\infty + B_D)^2\, dx$$

$$\lesssim \frac{\epsilon_{\text{low}}}{\sigma_t^4} B^{d+2}.$$

$\square$

### B.3.4 PROOF OF LEMMA B.6

*Proof.* For any $\epsilon > 0$, let $U_x$ be the set satisfies

$$U_x = \left\{ i \in [n] \,\middle|\, \left\| \frac{(x - \alpha_t x_i)}{\sigma_t} \right\|_2 \leq \sqrt{2 \log \epsilon^{-1}} \right\}.$$

It immediately gives us

$$\sum_{i=1}^n \frac{1}{n} \exp\left( -\frac{1}{2\sigma_t^2} \|x - \alpha_t x_i\|_2^2 \right) - \sum_{i \in U_x} \frac{1}{n} \exp\left( -\frac{1}{2\sigma_t^2} \|x - \alpha_t x_i\|_2^2 \right)$$

$$= \sum_{i \notin U_x} \frac{1}{n} \exp\left( -\frac{1}{2\sigma_t^2} \|x - \alpha_t x_i\|_2^2 \right)$$

$$\leq \sum_{i \notin U_x} \frac{1}{n} \epsilon$$

$$\leq \epsilon. \tag{B.17}$$

Then, we approximate $\exp\left( -\frac{1}{2\sigma_t^2} \|x - \alpha_t x_i\|_2^2 \right)$ for $i \in U_x$. We already have $\frac{1}{2\sigma_t^2} \|x - \alpha_t x_i\|_2^2 \leq \log \epsilon^{-1}$. By Taylor expansions, we have

$$\left| \exp\left( -\frac{1}{2\sigma_t^2} \|x - \alpha_t x_i\|_2^2 \right) - \sum_{k<p} \frac{1}{k!} \left( -\frac{1}{2\sigma_t^2} \|x - \alpha_t x_i\|_2^2 \right)^k \right| \leq \frac{\log^p \epsilon^{-1}}{p!},$$

where we use the fact $|e^{-x} - \sum_{k<p} \frac{1}{k!} x^k| \leq \frac{x^p}{p!}$ when $x > 0$. Let $p = \lceil 3u \log \epsilon^{-1} \rceil$, where $u$ satisfies $3u \log u = 1$, and invoking the equality $p! \geq (\frac{p}{3})^p$, it yields

$$\left| \exp\left( -\frac{1}{2\sigma_t^2} \|x - \alpha_t x_i\|_2^2 \right) - \sum_{k<p} \frac{1}{k!} \left( -\frac{1}{2\sigma_t^2} \|x - \alpha_t x_i\|_2^2 \right)^k \right| \leq \frac{\log^p \epsilon^{-1}}{p!} \leq u^{-3u \log \epsilon^{-1}} = \epsilon. \tag{B.18}$$

By (B.17) and (B.18), we have

$$\left| \sum_{i=1}^n \frac{1}{n} \exp\left( -\frac{1}{2\sigma_t^2} \|x - \alpha_t x_i\|_2^2 \right) - \sum_{i \in U_x} \frac{1}{n} \sum_{k<p} \frac{1}{k!} \left( -\frac{1}{2\sigma_t^2} \|x - \alpha_t x_i\|_2^2 \right)^k \right|$$

$$\leq \left| \sum_{i=1}^n \frac{1}{n} \exp\left( -\frac{1}{2\sigma_t^2} \|x - \alpha_t x_i\|_2^2 \right) - \sum_{i \in U_x} \frac{1}{n} \exp\left( -\frac{1}{2\sigma_t^2} \|x - \alpha_t x_i\|_2^2 \right) \right|$$

$$+ \left| \sum_{i \in U_x} \frac{1}{n} \exp\left( -\frac{1}{2\sigma_t^2} \|x - \alpha_t x_i\|_2^2 \right) - \sum_{i \in U_x} \frac{1}{n} \sum_{k<p} \frac{1}{k!} \left( -\frac{1}{2\sigma_t^2} \|x - \alpha_t x_i\|_2^2 \right)^k \right|$$

$$\leq 2\epsilon. \tag{B.19}$$

We set $B = 2\sqrt{2 \log \epsilon^{-1}}$ for convenience. We denote $\sum_{k<p} \frac{1}{k!} \left( -\frac{1}{2\sigma_t^2} \|x - \alpha_t x_i\|_2^2 \right)^k$ as $f_{p,i}(x,t)$, and $h_{p,i}(x,t) = f_{p,i}(x,t) \mathbb{1}_{\{i \in U_x\}}$, for any $i \in U_x$, we can approximate the Taylor expansion using ReLU network.

**Lemma B.9** (Concatenation, Remark 13 of (Nakada & Imaizumi, 2020))**.** For a series of ReLU networks $f_1 : \mathbb{R}^{d_1} \to \mathbb{R}^{d_2}, f_2 : \mathbb{R}^{d_2} \to \mathbb{R}^{d_3}, \ldots, f_k : \mathbb{R}^{d_k} \to \mathbb{R}^{d_{k+1}}$ with $f_i \in \mathcal{F}(W_i, L_i, N_i)$ $(i = 1, 2, \ldots, k)$, there exists a neural network $f \in \mathcal{F}(W, L, N)$ satisfying

$$f(x) = f_k \circ f_{k-1} \circ \cdots \circ f_1(x), \qquad \forall x \in \mathbb{R}^{d_1},$$

with

$$L = \sum_{i=1}^{k} L_i, \quad W \leq 2 \sum_{i=1}^{k} W_i, \quad N \leq 2 \sum_{i=1}^{k} N_i.$$

**Lemma B.10** (Identity function, Lemma F.2 of (Fu et al., 2024))**.** Given $d \in \mathbb{N}$ and $L \geq 2$, there exists $f_{\mathrm{id}}^L \in \mathcal{F}(2d, L, 2dL)$ that realizes an $L$–layer $d$-dimensional identity map

$$f_{\mathrm{id}}^L(x) = x, \quad x \in \mathbb{R}^d.$$

**Lemma B.11** (Parallelization and Summation, Lemma F.3 of (Oko et al., 2023))**.** For any neural networks $f_1, f_2, \ldots, f_k$ with $f_i : \mathbb{R}^{d_i} \to \mathbb{R}^{d_i'}$ and $f_i \in \mathcal{F}(W_i, L_i, N_i)$ $(i = 1, 2, \ldots, k)$, there exists a neural network $f \in \mathcal{F}(W, L, N)$ satisfying

$$f(x) = \left[ f_1(x_1)^\top f_2(x_2)^\top \cdots f_k(x_k)^\top \right]^\top : \mathbb{R}^{d_1+d_2+\cdots+d_k} \to \mathbb{R}^{d_1'+d_2'+\cdots+d_k'},$$

for all $x = (x_1^\top x_2^\top \cdots x_k^\top)^\top \in \mathbb{R}^{d_1+d_2+\cdots+d_k}$ (here $x_i$ can be shared), with

$$L = \max_{1 \leq i \leq k} L_i, \qquad W \leq 2 \sum_{i=1}^{k} W_i, \qquad N \leq 2 \sum_{i=1}^{k} (N_i + Ld_i').$$

Moreover, for $x_1 = x_2 = \cdots = x_k = x \in \mathbb{R}^d$ and $d_1' = d_2' = \cdots = d_k' = d'$, there exists $f_{\mathrm{sum}}(x) \in \mathcal{F}(W, L, N)$ that expresses $f_{\mathrm{sum}}(x) = \sum_{i=1}^{k} f_i(x)$, with

$$L = \max_{1 \leq i \leq k} L_i + 1, \qquad W \leq 4 \sum_{i=1}^{k} W_i, \qquad N \leq 4 \sum_{i=1}^{k} (N_i + Ld_i') + 2W. \tag{F.3}$$

**Lemma B.12** (Entry-wise Minimum and Maximum, Lemma F.4 of Fu et al. (2024))**.** For any two neural networks $f_1, f_2$ with $f_i : \mathbb{R}^d \to \mathbb{R}^{d'}$, $f_i \in \mathcal{F}(W_i, L_i, N_i)$ $(i = 1, 2)$ and $L_1 \geq L_2$, there exists a neural network $f \in \mathcal{F}(W, L, N)$ satisfying

$$f(x) = \min(f_1(x), f_2(x)) \quad (\text{or } \max(f_1(x), f_2(x))) \text{ for all } x \in \mathbb{R}^d,$$

with

$$L = L_1 + 1, \quad W \leq 2(W_1 + W_2), \quad N \leq 2(N_1 + N_2) + 2(L_1 - L_2)d'.$$

**Lemma B.13** (Approximating the product, Lemma F.6 of (Oko et al., 2023))**.** Let $d \geq 2$, $C \geq 1$. For any $\epsilon_{\mathrm{product}} > 0$, there exists $f_{\mathrm{mult}}(x_1, x_2, \ldots, x_d) \in \mathcal{F}(W, L, N)$ with

$$L = \mathcal{O}\big( \log d (\log \epsilon_{\mathrm{product}}^{-1} + d \log C) \big), \qquad W = 48d, \qquad N = \mathcal{O}(d \log \epsilon_{\mathrm{product}}^{-1} + d \log C),$$

such that

$$\left| f_{\mathrm{mult}}(x_1', x_2', \ldots, x_d') - \prod_{i=1}^{d} x_i \right| \leq \epsilon_{\mathrm{product}} + dC^{d-1}\epsilon_1. \tag{B.20}$$

for all $x \in [-C, C]^d$ and $x' \in \mathbb{R}^d$ with $\|x - x'\|_\infty \leq \epsilon_1$. Moreover, $|f_{\mathrm{mult}}(x)| \leq C^d$ for all $x \in \mathbb{R}^d$, and $f_{\mathrm{mult}}(x_1', x_2', \ldots, x_d') = 0$ if at least one of $x_i' = 0$.

We note that if $d = 2$ and $x_1 = x_2 = x$, it approximates the square of $x$. We denote the network by $f_{\mathrm{square}}(x)$ and the corresponding $\epsilon_{\mathrm{product}}$ by $\epsilon_{\mathrm{square}}$. Moreover, for any $x \in \mathbb{R}^d$ and $\mathbf{n} \in \mathbb{N}^d$, we denote the approximation of $x^{\mathbf{n}} = \prod_{i=1}^{d} x_i^{n_i}$ by $f_{\mathrm{poly},\mathbf{n}}(x)$ and the corresponding error by $\epsilon_{\mathrm{poly}}$.

**Lemma B.14** (Lemma F.7 of (Oko et al., 2023)). For any $0 < \epsilon_{\text{inv}} < 1$, there exists $f_{-1} \in \mathcal{F}(W, L, N)$ with

$$L = \mathcal{O}(\log^2 \epsilon_{\text{inv}}^{-1}), \quad W = \mathcal{O}(\log^3 \epsilon_{\text{inv}}^{-1}), \quad N = \mathcal{O}(\log^4 \epsilon_{\text{inv}}^{-1})$$

such that

$$\left| f_{-1}(x') - \frac{1}{x} \right| \leq \epsilon_{\text{inv}} + \frac{|x' - x|}{\epsilon_{\text{inv}}^2}, \qquad \text{for all } x \in [\epsilon_{\text{inv}}, \epsilon_{\text{inv}}^{-1}] \text{ and } x' \in \mathbb{R}. \qquad \text{(B.21)}$$

**Lemma B.15** (Lemma F.8 of (Fu et al., 2024)). For $\epsilon_\alpha \in (0, 1)$, there exists $f_\alpha \in \mathcal{F}(W, L, N)$ with

$$L = \mathcal{O}(\log^2 \epsilon_\alpha^{-1}), \quad W = \mathcal{O}(\log \epsilon_\alpha^{-1}), \quad N = \mathcal{O}(\log^2 \epsilon_\alpha^{-1}),$$

such that

$$|f_\alpha(t) - \alpha_t| \leq \epsilon_\alpha, \qquad \text{for all } t \geq 0. \qquad \text{(B.22)}$$

We can readily extend the approximation of $\alpha_t$ to $\alpha_t^2 = e^{-t}$ by doubling the coefficients in the first linear layer.

**Lemma B.16** (Lemma F.10 of (Fu et al., 2024)). For $\epsilon_\sigma \in (0, 1)$, there exists $f_\sigma \in \mathcal{F}(W, L, N)$ with

$$L = \mathcal{O}(\log^2 \epsilon_\sigma^{-1}), \quad W = \mathcal{O}(\log^3 \epsilon_\sigma^{-1}), \quad N = \mathcal{O}(\log^4 \epsilon_\sigma^{-1})$$

such that

$$\left| f_\sigma(t) - \sigma_t \right| \leq \epsilon_\sigma, \qquad \text{for all } t \geq \epsilon_\sigma. \qquad \text{(B.23)}$$

**Lemma B.17.** For any $\epsilon_{\sigma'} \in (0, 1)$, there exists $f_{\sigma'} \in \mathcal{F}(W, L, N)$ such that

$$\left| f_{\sigma'}(t) - \frac{1}{\sigma_t} \right| \leq \epsilon_{\sigma'}, \qquad \text{for all } t \geq \epsilon_{\sigma'},$$

with network parameters satisfying

$$L = \mathcal{O}(\log^2 \epsilon_{\sigma'}^{-1}), \quad W = \mathcal{O}(\log^3 \epsilon_{\sigma'}^{-1}), \quad N = \mathcal{O}(\log^4 \epsilon_{\sigma'}^{-1}).$$

*Proof.* We define the network by composition

$$f_{\sigma'}(t) = f_{-1}(f_\sigma(t)),$$

where $f_{-1}$ approximates the reciprocal function (Lemma B.14) and $f_\sigma$ approximates $\sigma_t = \sqrt{1 - e^{-t}}$ (Lemma B.16).

By Lemma B.14, the approximation error of $f_{-1}$ satisfies

$$\left| f_{\sigma'}(t) - \frac{1}{\sigma_t} \right| \leq \epsilon_{\text{inv}} + \frac{\epsilon_\sigma}{\epsilon_{\text{inv}}}.$$

Now we set

$$\epsilon_{\text{inv}} = \min\left( \frac{\epsilon_{\sigma'}}{2}, \frac{1}{\sqrt{1 - e^{-\epsilon_{\sigma'}}}} \right) = \mathcal{O}(\epsilon_{\sigma'}), \qquad \epsilon_\sigma = \frac{\epsilon_{\text{inv}} \epsilon_{\sigma'}}{2}.$$

With this choice, the total error is bounded by $\epsilon_{\sigma'}$ for all $t \geq \epsilon_{\sigma'}$. Finally, according to Lemma B.9, we can verify the network parameters $\mathcal{F}(W, L, N)$ satisfy

$$L = \mathcal{O}(\log^2 \epsilon_{\sigma'}^{-1}), \quad W = \mathcal{O}(\log^3 \epsilon_{\sigma'}^{-1}), \quad N = \mathcal{O}(\log^4 \epsilon_{\sigma'}^{-1}).$$

$\square$

**Lemma B.18** (ReLU approximation of the interval indicator). Fix $B > 0$ and a margin parameter $\tau(\delta) \in (0, 1]$. Let $\sigma(u) = \max\{0, u\}$ and define the "unit–ramp"

$$r_\tau(\delta)(u) = \sigma\left( \frac{u}{\tau(\delta)} \right) - \sigma\left( \frac{u}{\tau(\delta)} - 1 \right) \in [0, 1].$$

Consider

$$f_{B,\tau(\delta)}(x) = r_\tau(\delta)(x + B) - r_\tau(\delta)(x - B), \quad x \in \mathbb{R}.$$

Then $f_{B,\tau(\delta)} : \mathbb{R} \to [0, 1]$ is realized by a two–layer ReLU network with width 4, and it satisfies

$$f_{B,\tau(\delta)}(x) = \begin{cases} 0, & |x| \geq B + \tau(\delta), \\ 1, & |x| \leq B, \\ \text{linear in } x, & x \in [-B - \tau(\delta), -B] \cup [B, B + \tau(\delta)]. \end{cases}$$

Moreover, $f_{B,\tau(\delta)} \in \mathcal{F}(W, L, N)$ with

$$L = 2, \quad W = 4, \quad N = 1.$$

*Proof.* Since $r_\tau(\delta)(u)$ requires two ReLUs, the entire construction uses four ReLU units in parallel in a single hidden layer, followed by a linear output combination. This corresponds to a two–layer ReLU network (one hidden nonlinear layer plus the output layer) with width $W = 4$. Because all nonlinearities appear in one hidden layer, we have $N = 1$. Thus the stated bounds hold. $\qquad\square$

With these lemmas established, we are ready to approximate the Taylor series using a ReLU network. By Lemmas B.9, B.10, B.11, B.15, and B.17, we define the network as

$$\widehat{h}_{p,i}(x,t) = f_{\text{mult}}\left(f_{\text{sum},k<p}\left(\frac{(-1/2)^k}{k!}f_{\text{poly},k}(g_i(x,t))\right), f_{\text{indicator}}(x,t)\right),$$

where

$$g_i(x,t) = \sum_{j=1}^{d} f_{\text{mult}}(f_{\sigma'}, f_{\sigma'}, f_{\text{id}}^2([x]_j) - f_\alpha(t)[x_i]_j, f_{\text{id}}^2([x]_j) - f_\alpha(t)[x_i]_j) \quad (k \geq 1)$$

$$f_{\text{poly},0} = 1, \quad f_{\text{indicator}}(x,t) = f_{\sqrt{2\log\epsilon^{-1}},\tau(\delta)}(g_i(x,t)).$$

We further define

$$\widehat{f}_{p,i}(x,t) := f_{\text{sum},k<p}\left(\frac{(-1/2)^k}{k!}f_{\text{poly},k}(g_i(x,t))\right).$$

We first compute the approximation error between $\widehat{f}_{p,i}(x,t)$ and $f_{p,i}(x,t)$, which is

$$\epsilon_{p,i} \leq \sum_{k<p}\frac{\epsilon_{\text{poly},k}}{2^k k!} = e\epsilon_{\text{poly},k},$$

where

$$\epsilon_{\text{poly},k} = \epsilon_{\text{product},k,1} + C_{k,1}\epsilon_{k,1}, \quad \epsilon_{k,1} = d(\epsilon_{\text{product},k,2} + 4C_{k,2}^3\epsilon_{k,2})$$

$$C_{k,1} = k\left(\frac{\sqrt{d}(B+B_D)}{\sigma_{t_0}}\right)^{2(k-1)} \quad C_{k,2} = \max\left(\frac{1}{\sigma_{t_0}}, \sqrt{d}(B+B_D)\right), \quad \epsilon_{k,2} = \max(B_D\epsilon_\alpha, \epsilon_{\sigma'}).$$

We set $\epsilon^\star = \frac{\epsilon_{\exp}}{e}$, and take

$$\epsilon_{\text{product},k,1} = \frac{\epsilon^\star}{2}, \quad \epsilon_{\text{product},k,2} = \frac{\epsilon^\star}{4dC_{k,1}}, \quad \epsilon_\alpha = \frac{\epsilon^\star}{4C_{k,2}^3 B_D C_{k,1} d}, \quad \epsilon_{\sigma'} = \frac{\epsilon^\star}{4C_{k,2}^3 C_{k,1} d}.$$

Then, by the definition of $\epsilon_{\text{product},1}$, we can verify $\epsilon_{p,i} \leq \epsilon_{\exp}$. We decompose the total error into three parts

$$|\widehat{h}_{p,i}(x,t) - h_{p,i}(x,t)|$$
$$\leq \underbrace{|\widehat{h}_{p,i}(x,t) - \widehat{f}_{p,i}(x,t) \times f_{\text{indicator}}(x,t)|}_{D_{6,1}}$$
$$+ \underbrace{|\widehat{f}_{p,i}(x,t) \times f_{\text{indicator}}(x,t) - f_{p,i}(x,t) \times f_{\text{indicator}}(x,t)|}_{D_{6,2}}$$
$$+ \underbrace{|f_{p,i}(x,t) \times f_{\text{indicator}}(x,t) - f_{p,i}(x,t) \times \mathbb{1}_{\{i \in U_x\}}|}_{D_{6,3}}.$$

The first part arises from multiplying two networks. The second part comes from the approximation error of the Taylor expansion $\widehat{f}_{p,i}(x,t)$. The third part is due to the approximation error of the indicator function $\mathbb{1}_{\{i \in U_x\}}$. We now bound these three contributions separately. For $D_{6,1}$, by Lemma B.13, it implies

$$\left|\widehat{h}_{p,i}(x,t) - \widehat{f}_{p,i}(x,t) \times f_{\text{indicator}}(x,t)\right| \leq \epsilon_{\text{product},3}. \tag{B.24}$$

For $D_{6,2}$

$$\left|\widehat{f}_{p,i}(x,t) \times f_{\text{indicator}}(x,t) - f_{p,i}(x,t) \times f_{\text{indicator}}(x,t)\right| \leq |\widehat{f}_{p,i}(x,t) - f_{p,i}(x,t)| = \epsilon_{p,i} \leq \epsilon_{\exp}. \tag{B.25}$$

For $D_{6,3}$, when $\|\frac{x-\alpha_t x_i}{\sigma_t}\| \in [0, \sqrt{2\log\epsilon^{-1}}] \cup [\sqrt{2\log\epsilon^{-1}} + \tau(\delta), \infty]$, $f_{\text{indicator}}(x,t) = \mathbb{1}_{\{i\in U_x\}}$, then

$$|f_{p,i}(x,t) \times f_{\text{indicator}}(x,t) - f_{p,i}(x,t) \times \mathbb{1}_{\{i\in U_x\}}| = 0.$$

When $\|\frac{x-\alpha_t x_i}{\sigma_t}\| \in (\sqrt{2\log\epsilon^{-1}}, \sqrt{2\log\epsilon^{-1}} + \tau(\delta))$

$$|f_{p,i}(x,t) \times f_{\text{indicator}}(x,t) - f_{p,i}(x,t) \times \mathbb{1}_{\{i\in U_x\}}|$$
$$\leq |f_{p,i}(x,t)|$$
$$\leq \frac{(\log\epsilon^{-1} + 2\tau(\delta)\sqrt{\log\epsilon^{-1}} + \tau(\delta)^2)^p}{p!}$$
$$= \exp\left(3u\log\epsilon^{-1}\left(\log\left(1 + \frac{\tau(\delta)^2}{(\log\epsilon^{-1})} + 2\frac{\tau(\delta)}{\sqrt{\log\epsilon^{-1}}}\right) - \log u\right)\right)$$
$$= \exp\left(3u\log\epsilon^{-1}\left(2\log\left(1 + \frac{\tau(\delta)}{\sqrt{\log\epsilon^{-1}}}\right) - \log u\right)\right)$$
$$\leq \exp\left(3u\left(2\tau(\delta)\sqrt{\log\epsilon^{-1}} - \log u\log\epsilon^{-1}\right)\right). \tag{B.26}$$

Set $\tau(\delta) = \frac{1}{6u\sqrt{\log\epsilon^{-1}}}$, then from (B.26), we can conclude

$$|f_{p,i}(x,t) \times f_{\text{indicator}}(x,t) - f_{p,i}(x,t) \times \mathbb{1}_{\{i\in U_x\}}| \leq e\epsilon. \tag{B.27}$$

Combining (B.24), (B.25), and (B.27) together gives us

$$|\widehat{h}_{p,i}(x,t) - h_{p,i}(x,t)| \leq \epsilon_{\text{product},3} + \epsilon_{\exp} + e\epsilon. \tag{B.28}$$

We choose $\epsilon_{\exp} = \epsilon_{\text{product},3} = \epsilon$, and define $f_1^{\text{ReLU}}$ as

$$f_1^{\text{ReLU}} = f_{\text{mult}}(1/n, f_{\text{sum},1\leq i\leq n}(\widehat{h}_{p,i}(x,t))).$$

Consequently, from (B.19) and (B.28), we have

$$\left|\sum_{i=1}^n \frac{1}{n}\exp\left(-\frac{1}{2\sigma_t^2}\|x-\alpha_t x_i\|_2^2\right) - f_1^{\text{ReLU}}(x,t)\right| \leq (e+4)\epsilon + \epsilon_{\text{product},f_1}.$$

We choose $\epsilon_{\text{product},f_1} = \epsilon$, by Lemmas B.9, B.11, B.13, B.15, B.17, we have

$$\left|\sum_{i=1}^n \frac{1}{n}\exp\left(-\frac{1}{2\sigma_t^2}\|x-\alpha_t x_i\|_2^2\right) - f_1^{\text{ReLU}}(x,t)\right| \leq (e+5)\epsilon.$$

The network size parameters of $f_1^{\text{ReLU}}(x,t)$ satisfy

$$L = \widetilde{\mathcal{O}}(\log^2\epsilon^{-1}), \quad W = \widetilde{\mathcal{O}}(n\log^3\epsilon^{-1}), \quad N = \widetilde{\mathcal{O}}(n\log^4\epsilon^{-1}).$$

Substituting $\epsilon$ with $\frac{\epsilon_{f_1}}{e+5}$ immediately give us (B.9), and proof is complete. $\qquad\square$

### B.3.5 PROOF OF LEMMA B.7

*Proof.* This lemma serves as the counterpart of Lemma B.6. The proof follows a similar structure, and is same for every entry $j \in [d]$, with the only difference lying in the construction of $U_x$. Therefore, I will focus on elaborating this part. Let $U_x'$ be the set satisfies

$$U_x' = \left\{i \in [n] \left| \left\|\frac{(x-\alpha_t x_i)}{\sigma_t}\right\|_2 \leq 2\sqrt{\log\epsilon^{-1}}\right.\right\}.$$

It immediately gives us

$$\left|\sum_{i=1}^n \frac{[\alpha_t x_i - x]_j}{\sigma_t n}\exp\left(-\frac{1}{2\sigma_t^2}\|x-\alpha_t x_i\|_2^2\right) - \sum_{i\in U_x'}\frac{[\alpha_t x_i - x]_j}{\sigma_t n}\exp\left(-\frac{1}{2\sigma_t^2}\|\alpha_t x_i - x\|_2^2\right)\right|$$

$$= \left| \sum_{i \notin U'_x} \frac{1}{n} \frac{[\alpha_t x_i - x]_j}{\sigma_t} \exp\left(-\frac{1}{2\sigma_t^2} \|x - \alpha_t x_i\|_2^2\right) \right|$$

$$\leq \sum_{i \notin U'_x} \frac{2}{n} \sqrt{\log \epsilon^{-1}} \epsilon^2$$

$$\leq \epsilon.$$

The last inequality holds because $\epsilon$ is sufficiently small, ensuring that $2\epsilon\sqrt{\log \epsilon^{-1}} \leq 1$ (in fact, this condition is satisfied whenever $\epsilon \leq \frac{1}{e}$).

Then, we construct the network approximation in a similar manner. First, for each $1 \leq i \leq n$, we approximate the exponential function $\exp\left(-\frac{1}{2\sigma_t^2} \|x - \alpha_t x_i\|_2^2\right)$ and the term $\frac{[x - \alpha_t x_i]_j}{\sigma_t}$ separately using ReLU networks. Next, we combine these components using Lemma B.13. We then sum the resulting functions and multiply by $\frac{1}{n}$, applying Lemmas B.11 and B.13 as needed. Finally, we obtain the network configuration, completing the proof. □

### B.4 CONSTRUCTION OF $\mathbf{f}_3(x, t)$

We denote the entry-wise maximum function in Lemma B.12 as $f_{\max}$, and entry-wise minimum function in Lemma B.12 as $f_{\min}$. By Lemmas B.9, B.11, B.12, B.13, B.14, and B.17.

We define

$$f_3^{\text{ReLU}}(x, t, j)$$
$$= f_{\min}\left(f_{\text{mult}}(f_{\sigma'}, f_2^{\text{ReLU}}(x, t, j), f_{-1}(f_{\max}(f_1^{\text{ReLU}}(x, t), \epsilon_{\text{low}})), \frac{2\sqrt{2\log \epsilon^{-1}} + B_D}{f_{\sigma'}^2}\right),$$

We have

$$\left| f_3^{\text{ReLU}}(x, t, j) - \frac{f_2^{\text{ReLU}}}{\sigma_t f_{1, \text{clip}}} \right| \leq \max\left(\epsilon_{\text{mult},3} + 3C_{f,1}^2 \epsilon_{\sigma'}, \epsilon_{\text{product},f_3} + 3C_{f,2}^2(\epsilon_{\text{inv}} + \epsilon_{\sigma'})\right).$$

where

$$C_{f,1} = \max\left(2\sqrt{2\log \epsilon^{-1}} + B_D, \frac{1}{\sigma_{t_0}^2}\right), \quad C_{f,2} = \max\left(\frac{1}{\epsilon_{\text{low}}}, \frac{1}{\sigma_{t_0}}, \frac{2\sqrt{2\log \epsilon^{-1}} + B_D}{\sigma_{t_0}^2}\right).$$

We choose

$$\epsilon_{\text{mult},3} = \epsilon_{\text{product},f_3} = \frac{\epsilon}{2}, \quad \epsilon_{\sigma'} = \frac{\epsilon}{6C_{f,1}^2}, \quad \epsilon_{\text{inv}} = \epsilon_{\sigma'} = \epsilon_{\sigma'} = \frac{\epsilon}{12C_{f,2}^2}.$$

Then we can conclude

$$\left| f_3^{\text{ReLU}}(x, t, j) - \frac{f_2^{\text{ReLU}}}{\sigma_t f_{1, \text{clip}}} \right| \leq \epsilon.$$

Using Lemma B.11, we can construct

$$\mathbf{f}_3^{\text{ReLU}}(x, t) = [f_3^{\text{ReLU}}(x, t, 1), f_3^{\text{ReLU}}(x, t, 2), ..., f_3^{\text{ReLU}}(x, t, d)],$$

such that

$$\left\| \mathbf{f}_3^{\text{ReLU}}(x, t) - \mathbf{f}_3(x, t) \right\|_\infty \leq \epsilon.$$

The hyperparameters $(L, W, N)$ of the entire network satisfy

$$L = \mathcal{O}(\log^2 \epsilon^{-1}), \quad W = \mathcal{O}(n \log^3 \epsilon^{-1}), \quad N = \mathcal{O}(n \log^4 \epsilon^{-1}).$$

## C STATEMENT AND PROOF OF LEMMA C.1

In the following lemma, we investigate the Lipschitz continuity of score functions by computing the Hessian matrix of log density.

**Lemma C.1.** The Hessian of $\log p_t(x_t)$ admits the following explicit form:

$$\nabla^2 \log p_t(x_t) = -\frac{I}{\sigma_t^2} + \frac{\alpha_t^2}{\sigma_t^4} \operatorname{Cov}[X_0|X_t = x_t], \tag{C.1}$$

where the covariance is taken with respect to the posterior distribution of $X_0$ given $X_t$. Define the Lipschitz constant of the empirical score function $\nabla \log \widehat{p}_t(x_t)$ as

$$C_t = \sup_{x_t} \left\| \nabla^2 \log \widehat{p}_t(x_t) \right\|_2.$$

Assume that $n > 2$, and the minimum pairwise distance between data points satisfies

$$\min_{i \neq j, i, j \in [n]} \|x_i - x_j\|_2 \geq \frac{2\sigma_t}{\alpha_t} \sqrt{\log\left(\frac{n-2}{2}\right)},$$

Under this assumption, the Lipschitz constant $C_t$ satisfies the bounds

$$-\frac{1}{\sigma_t^2} + \frac{\alpha_t^2}{16\sigma_t^4} \min_{i \neq j, i, j \in [n]} \|x_i - x_j\|_2^2 \ \leq \ C_t \ \leq \ \frac{1}{\sigma_t^2} + \frac{\alpha_t^2}{4\sigma_t^4} \max_{i \neq j, i, j \in [n]} \|x_i - x_j\|_2^2. \tag{C.2}$$

When $t$ is small, we can conclude $C_t = \Omega(\sigma_t^{-4} \cdot \min_{i \neq j} \|x_i - x_j\|_2^2)$.

The proof is provided in Appendix C. Lemma C.1 provides a characterization of the Lipschitz constant of the score function. In particular, via (C.1), the posterior covariance $\operatorname{Cov}[X_0 \mid X_t = x_t]$ controls the smoothness of the score function.

For the empirical score $\nabla \log \widehat{p}_t(x_t)$, the covariance term is replaced by an empirical covariance computed from the sample. This empirical covariance varies significantly across $x_t$ and depends on the sample configuration, especially the pairwise distances between data points. As shown in Lemma C.1, under a separation condition on the data, the Lipschitz constant of the empirical score satisfies (C.2). This bound shows that $C_t$ can grow sharply when there are widely separated clusters ($\min_{i,j \in [n]} \|x_i - x_j\|_2$ large), especially at small noise levels $\sigma_t$, where the $\sigma_t^{-4}$ term strongly amplifies these effects.

*Proof.* We first write the explicit form of the Hessian of $\log p_t(x_t)$:

$$\nabla^2 \log p_t(x_t)$$
$$= -\frac{I}{\sigma_t^2} + \frac{\frac{1}{\sigma_t^4} \int (x_t - \alpha_t x_0)(x_t - \alpha_t x_0)^\top \exp\left(-\frac{\|x_t - \alpha_t x_0\|_2^2}{2\sigma_t^2}\right) p_{\text{data}}(x_0) \, dx_0}{\int \exp\left(-\frac{\|x_t - \alpha_t x_0\|_2^2}{2\sigma_t^2}\right) p_{\text{data}}(x_0) \, dx_0}$$
$$- \frac{\frac{1}{\sigma_t^4} e(x_t)(e(x_t))^\top}{\left(\int \exp\left(-\frac{\|x_t - \alpha_t x_0\|_2^2}{2\sigma_t^2}\right) p_{\text{data}}(x_0) \, dx_0\right)^2}.$$

where we define

$$e(x_t) = \int (x_t - \alpha_t x_0) \exp\left(-\frac{\|x_t - \alpha_t x_0\|_2^2}{2\sigma_t^2}\right) p_{\text{data}}(x_0) \, dx_0.$$

Notice that density function of the posterior distribution of $X_0$ given $X_t$ is

$$p(x_0|x_t) = \frac{\exp\left(-\frac{\|x_t - \alpha_t x_0\|_2^2}{2\sigma_t^2}\right) p_{\text{data}}(x_0)}{\int \exp\left(-\frac{\|x_t - \alpha_t x_0\|_2^2}{2\sigma_t^2}\right) p_{\text{data}}(x_0) \, dx_0}.$$

Using this posterior, the Hessian simplifies to

$$\nabla^2 \log p_t(x_t) = -\frac{I}{\sigma_t^2} + \frac{1}{\sigma_t^4} \operatorname{Cov}\left[X_t - \alpha_t X_0 | X_t = x_t\right],$$

where the covariance is taken with respect to $p(x_0|x_t)$. Since $X_t$ is constant given $x_t$, this further reduces to

$$\nabla^2 \log p_t(x_t) = -\frac{I}{\sigma_t^2} + \frac{\alpha_t^2}{\sigma_t^4} \operatorname{Cov}[X_0|X_t = x_t], \tag{C.3}$$

which is the form in (C.1).

To derive the upper bound for the Lipschitz constant of the empirical score function, we first obtain the expression for $\nabla^2 \log \widehat{p}_t(x_t)$ in a similar manner, using equation (C.3).

$$\nabla^2 \log \widehat{p}_t(x_t) = -\frac{I}{\sigma_t^2} + \frac{\alpha_t^2}{\sigma_t^4} \operatorname{Cov}[X_i|X_t = x_t],$$

where $X_i|X_t$ denotes the posterior distribution of $X_i$ given $X_t$.

For any $u \in R^d$ satisfying $\|u\|_2 = 1$,

$$|u^\top \nabla^2 \log \widehat{p}_t(x_t)u| \leq \frac{1}{\sigma_t^2} + \frac{\alpha_t^2}{\sigma_t^4} \operatorname{Var}(u^\top X_i|X_t = x_t).$$

To bound the variance term on the right-hand side, we introduce the following lemma.

**Lemma C.2** (Variance bound on a bounded interval). Let $X$ be a real random variable supported on $[a, b]$ (i.e., $a \leq X \leq b$ almost surely), and set $L = b - a$. Then

$$\operatorname{Var}(X) \leq \frac{L^2}{4}.$$

*Proof.* Fix $m = \mathbb{E}[X]$. Since $X \in [a, b]$ a.s. and $m \in [a, b]$, we have the pointwise bound

$$(X - m)^2 \leq \max\{(a - m)^2, (b - m)^2\}.$$

The function $m \mapsto \max\{(a - m)^2, (b - m)^2\}$ on $[a, b]$ is minimized at $m = \frac{a+b}{2}$ and its minimum value is $\left(\frac{b-a}{2}\right)^2$. Hence, for the actual $m = \mathbb{E}[X] \in [a, b]$,

$$(X - \mathbb{E}[X])^2 \leq \left(\frac{b - a}{2}\right)^2 \quad \text{a.s.}$$

Taking expectations yields

$$\operatorname{Var}(X) = \mathbb{E}\big[(X - \mathbb{E}[X])^2\big] \leq \frac{(b - a)^2}{4}.$$

$\square$

By Lemma C.2, we conclude that

$$|u^\top \nabla^2 \log \widehat{p}_t(x_t)u| \leq \frac{1}{\sigma_t^2} + \frac{\alpha_t^2(\max_i u^\top x_i - \min_i u^\top x_i)^2}{4\sigma_t^4}$$

$$\leq \frac{1}{\sigma_t^2} + \frac{\alpha_t^2 \max_{a,b} \|x_a - x_b\|_2^2}{4\sigma_t^4}.$$

By definition of $C_t$, we have $C_t = \sup_{\|u\|_2=1} |u^\top \nabla^2 \log \widehat{p}_t(x_t)u|$, and then we immediately derive the upper bound for $C_t$.

$$C_t \leq \frac{1}{\sigma_t^2} + \frac{\alpha_t^2 \max_{a,b} \|x_a - x_b\|_2^2}{4\sigma_t^4}$$

To establish the lower bound, we begin by expressing $\nabla^2 \log \widehat{p}_t(x_t)$ in a more explicit form.

$$\nabla^2 \log \widehat{p}_t(x_t)$$
$$= -\frac{I}{\sigma_t^2} + \frac{\frac{1}{\sigma_t^4} \sum_{i=1}^n (x_t - \alpha_t x_i)(x_t - \alpha_t x_i)^\top \exp\left(-\frac{\|x_t - \alpha_t x_i\|_2^2}{2\sigma_t^2}\right)}{\sum_{i=1}^n \exp\left(-\frac{\|x_t - \alpha_t x_i\|_2^2}{2\sigma_t^2}\right)}$$

$$-\frac{\frac{1}{\sigma_t^4}\left(\sum_{i=1}^n (x_t - \alpha_t x_i)\exp\left(-\frac{\|x_t - \alpha_t x_i\|_2^2}{2\sigma_t^2}\right)\right)\left(\sum_{i=1}^n (x_t - \alpha_t x_i)^\top \exp\left(-\frac{\|x_t - \alpha_t x_i\|_2^2}{2\sigma_t^2}\right)\right)}{\left(\sum_{i=1}^n \exp\left(-\frac{\|x_t - \alpha_t x_i\|_2^2}{2\sigma_t^2}\right)\right)^2}.$$

Denote $\mu(x_t) = \frac{\sum_{i=1}^n (x_t - \alpha_t x_i)\exp\left(-\frac{\|x_t - \alpha_t x_i\|_2^2}{2\sigma_t^2}\right)}{\left(\sum_{i=1}^n \exp\left(-\frac{\|x_t - \alpha_t x_i\|_2^2}{2\sigma_t^2}\right)\right)}$, $w_i(x_t) = \frac{\exp\left(-\frac{\|x_t - \alpha_t x_i\|_2^2}{2\sigma_t^2}\right)}{\sum_{i=1}^n \exp\left(-\frac{\|x_t - \alpha_t x_i\|_2^2}{2\sigma_t^2}\right)}$, we can rewrite $\nabla^2 \log \widehat{p}_t(x_t)$ as

$$\nabla^2 \log \widehat{p}_t(x_t) = -\frac{I}{\sigma_t^2} + \frac{1}{\sigma_t^4}\left(\sum_{i=1}^n (x_t - \alpha_t x_i)(x_t - \alpha_t x_i)^\top w_i(x_t) - \mu(x_t)\mu(x_t)^\top\right)$$

$$= -\frac{I}{\sigma_t^2} + \frac{1}{\sigma_t^4}\left(\sum_{i=1}^n (x_t - \alpha_t x_i - \mu(x_t))(x_t - \alpha_t x_i - \mu(x_t))^\top w_i(x_t)\right).$$

For any $u \in R^d$ satisfying $\|u\|_2 = 1$ we have

$$u^\top \nabla^2 \log \widehat{p}_t(x_t)u = -\frac{1}{\sigma_t^2} + \frac{1}{\sigma_t^4}\left(\sum_{i=1}^n w_i(x_t)\left((x_t - \alpha_t x_i - \mu(x_t))^\top u\right)^2\right).$$

We choose $(i,j)$ such that $\|x_i - x_j\| = \min_{i \neq j, i,j \in [n]} \|x_i - x_j\|_2$. At the midpoint $x_t = (x_i + x_j)/2$, we have

$$w_i(x_t) = w_j(x_t) = \frac{1}{2 + \sum_{h \neq i, h \neq j}\exp\left(-\frac{\alpha_t^2\left(\|x_t - x_h\|_2^2 - \|(x_i - x_j)/2\|_2^2\right)}{2\sigma_t^2}\right)}.$$

We introduce two lemmas to bound the difference $\|x_t - x_h\|_2^2 - \|(x_i - x_j)/2\|_2^2$ in terms of the minimum pairwise distance $\min_{a,b \in [n], a \neq b}\|x_a - x_b\|_2$.

**Lemma C.3.** Let $a, b, t \in \mathbb{R}^d$, set the midpoint $m = \frac{a+b}{2}$ and $r = \frac{1}{2}\|a - b\|_2$. Then

$$\|t - m\|_2^2 = \frac{1}{2}\left(\|t - a\|_2^2 + \|t - b\|_2^2\right) - r^2.$$

*Proof.* Observe that $t - m = \frac{1}{2}\left((t - a) + (t - b)\right)$, hence

$$4\|t - m\|_2^2 = \|(t - a) + (t - b)\|_2^2 = \|t - a\|_2^2 + \|t - b\|_2^2 + 2\langle t - a, t - b\rangle.$$

Also,

$$\|(t - a) - (t - b)\|_2^2 = \|a - b\|_2^2 = \|t - a\|_2^2 + \|t - b\|_2^2 - 2\langle t - a, t - b\rangle,$$

so

$$2\langle t - a, t - b\rangle = \|t - a\|_2^2 + \|t - b\|_2^2 - \|a - b\|_2^2.$$

Substitute into the first display:

$$4\|t - m\|_2^2 = 2\left(\|t - a\|_2^2 + \|t - b\|_2^2\right) - \|a - b\|_2^2.$$

Divide by 4 and note $r^2 = \frac{1}{4}\|a - b\|_2^2$ to obtain

$$\|t - m\|_2^2 = \frac{1}{2}\left(\|t - a\|_2^2 + \|t - b\|_2^2\right) - r^2. \qquad \square$$

**Lemma C.4.** Let

$$\widehat{\Delta}_{\min} = \min_{a \neq b}\|x_a - x_b\|_2.$$

Then we have

$$\|x_t - x_h\|_2^2 - \|(x_i - x_j)/2\|_2^2 \geq \frac{\widehat{\Delta}_{\min}^2}{2}, \quad h \neq i, h \neq j$$

where $x_t = \frac{x_i + x_j}{2}$, $(i,j)$ satisfies $\|x_i - x_j\| = \widehat{\Delta}_{\min}$,

*Proof.* By Lemma C.3

$$\|x_t - x_h\|_2^2 - \|(x_i - x_j)/2\|_2^2 = \frac{1}{2}\Big(\|x_h - x_i\|_2^2 + \|x_h - x_j\|_2^2\Big) - \frac{\widehat{\Delta}_{\min}^2}{2}$$

$$\geq \widehat{\Delta}_{\min}^2 - \frac{\widehat{\Delta}_{\min}^2}{2}$$

$$= \frac{\widehat{\Delta}_{\min}^2}{2}.$$

$\square$

By Lemma C.4, we obtain $\|x_t - x_h\|_2^2 - \|(x_i - x_j)/2\|_2^2 \geq \frac{1}{2}\min_{a,b\in[n],a\neq b}\|x_a - x_b\|_2^2$. Since $\min_{a,b\in[n],a\neq b}\|x_a - x_b\|_2 \geq \frac{2\sigma_t}{\alpha_t}\sqrt{\log\left(\frac{n-2}{2}\right)}$, then we have $w_i(x_t) = w_j(x_t) \geq \frac{1}{4}$. Let $u = \frac{x_i - x_j}{\|x_i - x_j\|_2}$.

$$u^\top \nabla^2 \log \widehat{p}_t(x_t) u$$

$$= -\frac{1}{\sigma_t^2} + \frac{1}{\sigma_t^4}\left(\sum_{i=1}^n w_i(x_t)\left((x_t - \alpha_t x_i - \mu(x_t))^\top u\right)^2\right)$$

$$\geq -\frac{1}{\sigma_t^2} + \frac{1}{4\sigma_t^4}\left(\left((\alpha_t(x_i - x_j)/2 + \mu(x_t))^\top u\right)^2 + \left((\alpha_t(x_i - x_j)/2 - \mu(x_t))^\top u\right)^2\right)$$

$$= -\frac{1}{\sigma_t^2} + \frac{1}{4\sigma_t^4}\left(\left(\mu(x_t)^\top u\right)^2 + \left((\alpha_t(x_i - x_j)/2)^\top u\right)^2\right)$$

$$\geq -\frac{1}{\sigma_t^2} + \frac{\alpha_t^2}{16\sigma_t^4}\|x_i - x_j\|_2^2$$

$$= -\frac{1}{\sigma_t^2} + \frac{\alpha_t^2}{16\sigma_t^4}\min_{i\neq j, i,j\in[n]}\|x_i - x_j\|_2^2.$$

Therefore we can conclude

$$\nabla^2 \log \widehat{p}_t(x_t) \succeq \left(-\frac{1}{\sigma_t^2} + \frac{\alpha_t^2}{16\sigma_t^4}\min_{i\neq j, i,j\in[n]}\|x_i - x_j\|_2^2\right) I,$$

which immediately implies

$$\|\nabla^2 \log \widehat{p}_t(x_t)\|_2 \geq \left(-\frac{1}{\sigma_t^2} + \frac{\alpha_t^2}{16\sigma_t^4}\min_{i\neq j, i,j\in[n]}\|x_i - x_j\|_2^2\right),$$

and it implies the lower bound for $C_t$

$$C_t \geq -\frac{1}{\sigma_t^2} + \frac{\alpha_t^2}{16\sigma_t^4}\min_{i\neq j, i,j\in[n]}\|x_i - x_j\|_2^2.$$

Moreover, when $t$ is small, we can conclude $C_t = \Omega(\sigma_t^{-4} \cdot \min_{i\neq j}\|x_i - x_j\|_2^2)$. $\square$

# D EXPERIMENTAL DETAILS

## D.1 COMPUTING THE IMPORTANCE SCORE

To formalize the computation of importance scores, we follow the masking-based framework of (Liang et al., 2021). In each Transformer layer of the diffusion model, we associate a binary mask variable $\xi_h \in \{0, 1\}$ with every attention head $h$. Setting $\xi_h = 1$ keeps the head active, while $\xi_h = 0$ prunes it away. Let $\mathcal{L}(x, t; \mathcal{M})$ denote the training loss of the model $\mathcal{M}$ on input $x$ at diffusion step $t$. The sensitivity of $\mathcal{L}$ with respect to $\xi_h$ quantifies how important head $h$ is to the model's predictions. We thus define the importance score of $h$ as the expected gradient magnitude of $\mathcal{L}$ with respect to $\xi_h$, averaged over data and timesteps, and layerwise $\ell_2$ normalized:

$$I^{(h)} = \frac{\mathbb{E}_{x\sim\mathcal{D},\, t\sim\mathcal{T}}\left[\left|\frac{\partial \mathcal{L}(x,t;\mathcal{M})}{\partial \xi_h}\right|\right]}{\sqrt{\sum_{h'\in\text{layer}(h)}\left(\mathbb{E}_{x\sim\mathcal{D},\, t\sim\mathcal{T}}\left[\left|\frac{\partial \mathcal{L}(x,t;\mathcal{M})}{\partial \xi_h}\right|\right]\right)^2}} \quad \in [0, 1].$$

---

**Algorithm 2** IMPORTANCESCORE($\mathcal{M}, \mathcal{D}, \mathcal{T}$)

---

1: **Input:**
2:     Model $\mathcal{M}$ with mask variables $\{\xi_h\}$ for all heads $h \in \mathcal{H}$.
3:     Dataset $\mathcal{D}$, Time Sampling Distribution $\mathcal{T}$.
4: **Initialize:** Accumulated scores $S^{(h)} \leftarrow 0$ for all $h \in \mathcal{H}$.
5: **for** each batch of data $x \sim \mathcal{D}$ **do**
6:     Sample timestep $t \sim \mathcal{T}$.
7:     Compute loss $\mathcal{L}(x, t; \mathcal{M})$.
8:     Backpropagate to obtain all gradients $\left\{ \frac{\partial \mathcal{L}}{\partial \xi_h} \right\}_{h \in \mathcal{H}}$.
9:     Accumulate scores: $S^{(h)} \leftarrow S^{(h)} + \left| \frac{\partial \mathcal{L}}{\partial \xi_h} \right|$ for all $h \in \mathcal{H}$.
10: **for** each layer $l$ in the model **do**
11:     Compute layer-wise norm: $N_l \leftarrow \sqrt{\sum_{h' \in l} (S^{(h')})^2}$.
12:     **for** each head $h$ in layer $l$ **do**
13:         Normalize score: $I^{(h)} \leftarrow S^{(h)} / N_l$.
14: **Output:** Importance scores $\{I^{(h)}\}_{h \in \mathcal{H}}$.

---

## D.2 COMPUTING PRECISION AND RECALL

We follow the definitions in Kynkäänniemi et al. (2019) to compute precision and recall. Let the real and generated feature sets (obtained from InceptionV3 feature embeddings) be

$$\Phi_r = \{\phi_r^{(i)}\}_{i=1}^{N_r}, \qquad \Phi_g = \{\phi_g^{(j)}\}_{j=1}^{N_g}.$$

For a feature vector $\phi$ and a set $\Phi$, define the indicator function

$$f(\phi, \Phi) = \begin{cases} 1, & \text{if } \exists \phi' \in \Phi \text{ such that } \|\phi - \phi'\|_2 \leq \|\phi' - \text{NN}_k(\phi', \Phi)\|_2, \\ 0, & \text{otherwise}, \end{cases}$$

where $\text{NN}_k(\phi', \Phi)$ denotes the $k$-th nearest neighbor of $\phi'$ within $\Phi$. Precision and recall are then defined as

$$\text{Precision}(\Phi_r, \Phi_g) = \frac{1}{|\Phi_g|} \sum_{\phi_g \in \Phi_g} f(\phi_g, \Phi_r), \qquad \text{Recall}(\Phi_r, \Phi_g) = \frac{1}{|\Phi_r|} \sum_{\phi_r \in \Phi_r} f(\phi_r, \Phi_g).$$

Intuitively, each feature set induces a manifold via local $k$-nearest-neighbor neighborhoods. Precision measures the fraction of generated samples that lie within the manifold of real data, while recall measures the fraction of real samples that lie within the manifold of generated data.

## D.3 MODEL CONFIGURATION AND TRAINING

For numerical simulations on Gaussian mixture data, we build standard MLPs and vary their width, depth and diffusion time embedding dimension to achieve different parameter settings. Detailed configurations are presented in Table 3.

For the experiments on CIFAR-10, we adapt the implementation of DiT (Peebles & Xie, 2023) from https://github.com/ArchiMickey/Just-a-DiT. Our training set is a randomly chosen subset of CIFAR-10 containing 5,000 images. The model has hidden dimension 384, 12 layers, and 6 heads per layer. We use a learning rate of $2 \times 10^{-4}$ with a cosine scheduler and train for 100,000 steps without weight decay to obtain the original model. After pruning, the model is further trained for 5,000 steps to obtain the results. When sampling, we use a deterministic sampler with 50 steps, classifier free guidance scale 2.0, and randomly generated labels for each sample. Both memorization ratio and FID are evaluated using 50K generated samples.

Additional results including the case with pruning ratio $\eta = 40\%$ are summarized in Table 2.

| Model | Precision (↑) | Recall (↑) | Memorization Ratio (%) (↓) | FID (↓) |
|---|---|---|---|---|
| Original | $0.39_{\pm 0.01}$ | $0.08_{\pm 0.01}$ | $73.82_{\pm 1.12}$ | $15.47_{\pm 0.28}$ |
| Our Pruning (20%) | $0.33_{\pm 0.02}$ | $0.12_{\pm 0.01}$ | $68.58_{\pm 0.77}$ | $15.07_{\pm 0.33}$ |
| Random Pruning (20%) | $0.30_{\pm 0.02}$ | $0.09_{\pm 0.01}$ | $66.87_{\pm 0.94}$ | $17.14_{\pm 0.25}$ |
| Our Pruning (40%) | $0.25_{\pm 0.02}$ | $0.08_{\pm 0.00}$ | $58.63_{\pm 1.18}$ | $16.53_{\pm 0.36}$ |
| Random Pruning (40%) | $0.24_{\pm 0.02}$ | $0.06_{\pm 0.01}$ | $55.72_{\pm 0.99}$ | $20.16_{\pm 0.41}$ |

Table 2: Additional results including pruning ratio $s = 40\%$. We report precision, recall, memorization ratio, and FID. Each value is shown as mean$_{\pm \text{std}}$ over 5 random seeds.

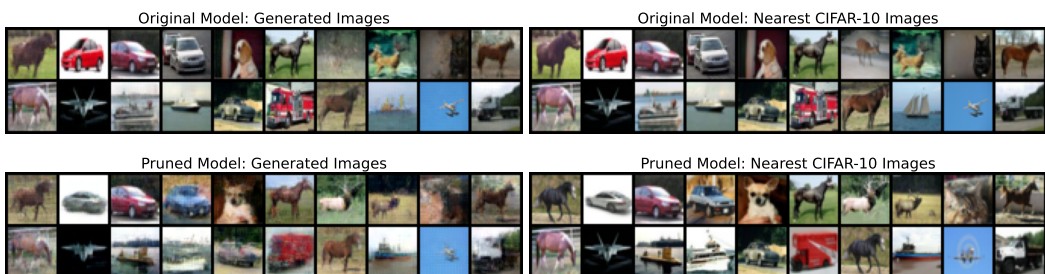

Figure 4: **Left:** Generated images from the same random noise, with the original model (top) and our pruned model (bottom). **Right:** Nearest neighbors of the generated images in the CIFAR-10 training set. At a comparable level of quality, the pruned model shows greater diversity, while the original model tends to replicate training samples.

| Parameter Count | Network Configuration | | |
|---|---|---|---|
| | Width | # Depth | Time Embedding Dim |
| 24,128 | 32 | 3 | 32 |
| 118,144 | 64 | 4 | 64 |
| 417,344 | 128 | 6 | 64 |
| 2,180,224 | 256 | 8 | 128 |
| 5,009,920 | 384 | 10 | 128 |
| 12,888,320 | 512 | 12 | 256 |
| 44,396,288 | 1024 | 14 | 256 |

Table 3: Network configurations corresponding to parameter counts.

## THE USE OF LARGE LANGUAGE MODELS (LLMS)

We used LLMs as assistants for writing and coding tasks such as formatting results into tables, refining phrasing, polishing standard sections, and assisting with code logging and debugging, but not for generating research ideas, designing experiments, or analyzing raw results; all substantive contributions were carried out by the authors, who take full responsibility for the final content.

