# OpenReview forum: "Provable Separations between Memorization and Generalization in Diffusion Models"
_ICLR.cc/2026/Conference — ICLR 2026 Poster_

### Official Review · Reviewer_95eC · 2025-10-22

**Soundness:** 4
**Presentation:** 3
**Contribution:** 3
**Rating:** 8
**Confidence:** 3

**Summary:**

The paper proposes a theoretical analysis of the memorization phenomenon in diffusion models. The analysis offers two perspectives: a statistical and an architectural one. The statistical perspective distinguishes between the ground-truth score function, which reflects the underlying data distribution, and the empirical score function that minimizes the loss on the (sampled) training data. Optimizing the diffusion model pushes the model to learn the empirical score function, which minimizes the training loss, instead of approximating the ground-truth score function, thereby leading to memorization. The subsequent analysis of the network complexity (number of parameters) demonstrates that over-parameterized diffusion models learn the complex empirical score function, also leading to memorization. Based on these theoretical insights, the paper explores mitigation strategies by reducing the model complexity by pruning attention heads or weight decay. These experiments are conducted on CIFAR-10 and Gaussian mixtures.

**Strengths:**

- The paper provides fascinating insights into the training dynamics and influencing factors favoring memorization in diffusion models. Thorough mathematical foundations support the proposed framework. All necessary proofs are provided in the main paper or the appendix.
- The theoretical analysis of the loss gap between the ground-truth and the empirical score function offers an intuitive explanation of why diffusion models can inadvertently memorize and replicate parts of their training data.
- While, to some extent, this is to be expected, the pruning- and regularization-based mitigation strategies support model developers in their design decisions for building diffusion models and setting the training parameters.

**Weaknesses:**

- While the paper provides an interesting perspective on diffusion memorization, the core insight that a model tends to minimize the empirical loss instead of the ground-truth loss feels like the traditional bias-variance trade-off and overfitting phenomena in statistical machine learning. I do not want to reduce the paper’s contribution, but want to emphasize that the high-level message of the paper might have limited novelty. However, it might be the case that I overlooked some crucial novelty of the paper here.
- The paper focuses on unconditional and low-resolution diffusion models trained on comparably small datasets (a couple of thousand training samples). Whereas I think this is appropriate for the provided analysis, it remains unclear to what extent the results transfer to more complex distributions, such as LAION data, and sample counts in the billions. Yet, I acknowledge that running such experiments is not feasible for small research labs, which is why I do not request such experiments. However, a more thorough discussion on how the authors expect their findings to scale or behave in such regimes would benefit the paper.
- The results of the proposed pruning mitigation support an improvement in memorization mitigation. However, the reduction is relatively small (about 5 percentage points compared to the baseline, still reporting a 68% memorization ratio). Therefore, it feels less suitable for practical application.

**Questions:**

- Do the findings also apply to large-scale diffusion models, e.g., Stable Diffusion trained on billions of data samples?
- As a follow-up question: We know that Stable Diffusion has memorized some of its training samples. Does this mean the capacity of Stable Diffusion is too large? Or can we assume, since only a small share of duplicated training samples has been memorized, that Stable Diffusion already has a sufficient size, and the issue is the duplicated dataset?

---

> ### Author Response · Authors · 2025-11-22
>
> Thank you for your kind recognition and inspiring comments on our work, we appreciate your time and insightful suggestions. We now summarize the weaknesses and questions you raised and address them as follows:
>
> > **Q1**: While the paper provides an interesting perspective on diffusion memorization, the core insight that a model tends to minimize the empirical loss instead of the ground-truth loss feels like the traditional bias-variance trade-off and overfitting phenomena in statistical machine learning.
>
> **A1:** This is a very sharp observation. For the statistical separation part, our starting point is indeed closely related to classical overfitting and empirical risk minimization phenomena in statistical learning. However, as discussed in the paragraph following Proposition 4.1, our statistical separation analysis is neither a generalization bound nor an upper bound of the type typically encountered in the classical bias–variance trade-off framework. Instead, we analyze the loss under the empirical distribution, which is not the generalization error, and makes the standard bias–variance decomposition inapplicable. Establishing the lower bound for the loss gap in Theorem 4.3 requires new analytical techniques—for example, isolating the dominant sample or component of the empirical and true score functions. Please see Appendix A.2 for the full derivation.
>
> Beyond statistical separation, our second contribution — the network architectural separation — provides further novelty by characterizing what kinds of neural networks are more likely to represent the empirical score and thus memorize. Our work provides new **non-asymptotic analysis** under **realistic deep-network settings** (deep ReLU architectures) and **general sub-Gaussian mixture distributions**, which cover many real-world scenarios.
>
> > **Q2**: The paper focuses on unconditional and low-resolution diffusion models trained on comparably small datasets (a couple of thousand training samples). Whereas I think this is appropriate for the provided analysis, it remains unclear to what extent the results transfer to more complex distributions, such as LAION data, and sample counts in the billions. Yet, I acknowledge that running such experiments is not feasible for small research labs, which is why I do not request such experiments. However, a more thorough discussion on how the authors expect their findings to scale or behave in such regimes would benefit the paper.
>
> **A2:** Thank you for the suggestion, and we also appreciate your understanding that it is not feasible for us to scale the experiments to LAION-level datasets or billion-parameter diffusion models. Adding such a discussion will definitely strengthen the paper. We would like to clarify that our main theoretical results, both the statistical separation and network architectural separation, do not inherently require small datasets or simple distributions. For example, the sample requirement in Theorem 4.3, $\log(n) = \mathcal{O}(d)$, is quite mild and is typically satisfied in modern large-scale training regimes. Similarly, our data distribution assumptions—sub-Gaussian mixtures—capture a broad class of real-world data. As discussed in Section 4, mixture models align well with the multi-modality observed in large-scale datasets: different categories (e.g., dogs, airplanes, buildings) naturally correspond to different mixture components, and these components generally exhibit sub-Gaussian tails.

---

> ### Author Response · Authors · 2025-11-22
>
> > **Q3**: The results of the proposed pruning mitigation support an improvement in memorization mitigation. However, the reduction is relatively small. Therefore, it feels less suitable for practical application.
>
> **A3**: Thank you for the comments. We mainly focus on theory of provable separations between memorization and generalization. The pruning procedure is directly motivated by our theoretical insights (as discussed in Section 6.2) and intentionally kept simple. Its purpose is to provide empirical validation of the theoretical findings rather than to introduce a new SOTA mitigation technique. For this reason, we believe the current experiments and results are sufficient for supporting our theoretical claims. For large-scale, real-world applications, we acknowledge that such an intentionally simple algorithm would require additional engineering heuristics. We welcome further discussion and insights on how these ideas might be adapted in more applied settings.
>
> > **Q4**: As a follow-up question: We know that Stable Diffusion has memorized some of its training samples. Does this mean the capacity of Stable Diffusion is too large? Or can we assume, since only a small share of duplicated training samples has been memorized, that Stable Diffusion already has a sufficient size, and the issue is the duplicated dataset?
>
> **A4:** Thank you for raising this insightful question — it touches on a subtle aspect. Data duplication, network capacity, and dataset size can all contribute, and we address these factors below.
>
> First, regarding your intuition that duplicated samples are more likely to be memorized: this is consistent with our theoretical framework. As discussed in Q3 & A3 for Reviewer M2PN, duplicated samples effectively receive larger weights in the empirical loss. During training, the model minimizes the empirical objective $\int_{t_0}^T \frac{1}{n} \ell_t(x_i, s) dt.$ If the dataset contains $n$ unique points, each contributes $1/n$ to the loss. But if a point $x_j$ appears many times, it contributes a disproportionately large share of the loss. The optimizer naturally prioritizes minimizing the most influential terms, so multiple copies of $\ell\_t(x\_j, s)$ encourage the model to reduce this particular loss as much as possible. This makes duplicated samples more prone to memorization.
>
> Second, regarding the relationship between dataset size, network capacity, and the **partial memorization** phenomenon (i.e., only some training samples are memorized): this is a more interesting regime. Even without duplication, partial memorization may emerge when the network size is in a medium-to-large range. Prior work such as [1] observes this effect and provides valuable insights. However, their analysis is conducted under restrictive assumptions — Gaussian mixture data with a parametric Gaussian mixture denoiser family — which limits how far their conclusions can generalize.
>
> The question of which samples are memorized, why those samples in particular, and under what conditions partial memorization occurs remains largely open. Identifying the underlying mechanisms — especially under realistic data distributions and architectures — is still an active area of research. We view this as a promising direction, and we are currently developing statistical tools (though still facing technical challenges) to better understand when partial memorization arises, what network configurations are critical, and what sample characteristics make memorization more likely.
>
> Thank you again for raising this point — we appreciate the opportunity to discuss it, and we welcome further insights or suggestions you may have.
>
>
> [1] Buchanan, Sam, et al. "On the edge of memorization in diffusion models." arXiv preprint arXiv:2508.17689 (2025).

---

> > ### Comment · Reviewer_95eC · 2025-11-25
> >
> > I thank the authors for the clarifications and detailed responses. After reading all reviews and author responses, I decided to keep my initial score of accepting the paper. I think other reviewers also made valid points, and some improvements/updates of the paper will improve it. For example, as reviewer XbEb pointed out, the assumptions and hypothesis could be made a bit clearer in the paper to facilitate understanding for the reader. But I think the additional page for the final version provides sufficient space for such adjustments. I also appreciate that the current updated version already includes most details and analysis requested by the reviewers.

---

> > > ### Author Response · Authors · 2025-11-25
> > >
> > > Thank you for your recognition and for taking the time to revisit our work. We appreciate your constructive insights throughout the review process. We will definitely incorporate the thoughtful suggestions raised by you and the other reviewers into the final version.

---

### Official Review · Reviewer_ENVr · 2025-10-29

**Soundness:** 3
**Presentation:** 4
**Contribution:** 3
**Rating:** 6
**Confidence:** 3

**Summary:**

This paper theoretically proves a fundamental separation between memorization and generalization in diffusion models. It shows that finite-sample Fisher divergence and network capacity jointly cause models to fit empirical rather than true score functions. Experiments demonstrate that pruning and regularization can reduce memorization while preserving generation quality.

**Strengths:**

1. The paper provides a rigorous theoretical analysis of diffusion models by quantifying the loss gap between the empirical and ground-truth scores under sub-Gaussian, Hölder-smooth data distributions, and further establishes an architectural separation showing that approximating the empirical score requires higher model complexity.
2. The paper proposes a practical one-shot pruning method for Diffusion Transformers that removes low-importance heads in the small-t regime, effectively reducing memorization while maintaining or even improving generation quality, as supported by CIFAR-10 experiments.

**Weaknesses:**

1. The analysis mainly focuses on the small-t regime, but the paper does not empirically verify whether memorization indeed concentrates in this phase during real generation — an experiment comparing large-t and small-t generations could strengthen the claims.
2. I think The theoretical results rely on strong assumptions such as sub-Gaussianity, which limit their applicability to real-world data.
3. The proposed pruning method is tested on a single dataset, its generalization to other datasets or more complex diffusion models is uncertain, and its ability to substantially reduce memorization appears limited.

**Questions:**

See Weakness. Further:
    1. The results rely on sub-Gaussianity and Hölder smoothness assumptions. How sensitive are your main theorems to these conditions?

---

> ### Author Response · Authors · 2025-11-22
>
> Thank you for taking the time to provide a detailed review, and we appreciate your recognition of our contributions. We now summarize the weaknesses and questions you raised and address them as follows:
>
> > **Q1:** The theoretical analysis mainly focuses on the small-$t$ regime, while the paper does not empirically verify whether memorization indeed concentrates in this phase during real generation.
>
> **A1:** Thank you for raising this point. Theorem 4.3 shows that the loss gap between the empirical and true score functions is large in the small-$t$ regime. As mentioned in the discussion following the theorem, the overall training loss gap $\mathbb{E}\_{\mathcal{D}}[\hat{\mathcal{L}}(\nabla \log p\_t) - \hat{\mathcal{L}}(\nabla \log \hat{p}\_t)]
> = \int\_{t\_0}^T \mathbb{E}\_{\mathcal{D}}[\texttt{Loss-Gap}\_t]  dt$
> is therefore influenced by the small-$t$ contributions. When combined with an expressive neural network (see Theorem 5.1 and accompanying discussion) and a strong optimizer, the model is more likely to be driven toward the overall empirical minimizer, which leads to memorization. However, this does not imply that memorization during the generation process occurs exclusively in the small-$t$ regime. Making such a claim would go beyond our statistical guarantee and is not the focus of our current framework. Memorization during sampling is an accumulated effect across the full denoising trajectory, and studying it would require a detailed analysis of the backward generation dynamics. For reference, some works such as [1] specifically examine generation dynamics and provide insightful results.
>
> We nevertheless appreciate your suggestion to provide numerical evidence supporting the claim in Theorem 4.3. In the revised version, we include an empirical evaluation of $\texttt{Loss-Gap}\_t$ under a Gaussian mixture setting, varying both $t$ and the sample size $n$. The results (see *updated PDF, Figure 1*) align with our theoretical predictions: the gap increases as $t$ becomes smaller, and insufficient sample size $n$ also yields a large gap.
>
> > **Q2**: The results' reliance on sub-Gaussianity and Hölder smoothness assumptions seems limiting. How sensitive are your main theorems to these conditions?
>
> **A2:** Thank you for the request for clarification. First, our statistical separation analysis can also be adapted to mixtures with bounded support (a relaxation of the support assumptions in Assumption 4.2 on sub-Gaussian Hölder densities), as noted in the accompanying discussion of Theorem 4.3. More importantly, regarding the assumptions themselves, sub-Gaussianity is common in real-world data; and mixture models, as discussed in Section 4, also provide a flexible and realistic way to capture the multi-modality seen in real datasets. For instance, image datasets often contain distinct categories—such as dogs and airplanes—that correspond to different mixture components, and these components typically satisfy sub-Gaussian tail behavior.
>
>
>
> As noted in Section 7, our analysis does not directly apply to heavy-tailed distributions. Nonetheless, relevant study of diffusion models for heavy-tailed data is limited, and incorporating such cases into our analysis is even more challenging. We view this as a promising direction, although we currently face technical challenges in extending our tools to this regime. We are grateful that you raised this point, and we would welcome further discussion if you have suggestions or insights on this topic.
>
> > **Q3**: The proposed pruning method is tested on a single dataset, its generalization to other datasets or more complex diffusion models is uncertain.
>
> **A3**: We mainly focus on theory of provable separations between memorization and generalization. The pruning procedure is directly motivated by these theoretical insights (as discussed in Section 6.2) and intentionally kept simple. Its purpose is to provide empirical validation of the theoretical findings rather than to introduce a new SOTA mitigation technique. For this reason, we believe the current experiments and results are sufficient to support our theoretical claims.
>
>
> [1] Biroli, Giulio, et al. "Dynamical regimes of diffusion models." Nature Communications 15.1 (2024): 9957.

---

### Official Review · Reviewer_XbEb · 2025-10-30

**Soundness:** 2
**Presentation:** 2
**Contribution:** 2
**Rating:** 4
**Confidence:** 4

**Summary:**

This paper develops a theoretical framework for explaining memorization and generalization in diffusion models. First, it shows that the denoising score matching loss with respect to an empirical distribution is not minimized by the ground-truth score function, due to a statistical lower bound.
Second, it proves that under the universal approximation perspective, representing the empirical score requires larger networks than the case of ground-truth score.
These results establish quantitative separations between the regimes where diffusion models memorize versus generalize.
The authors further propose a pruning scheme for mitigating memorization while preserving generation quality.

**Strengths:**

The paper provides a solid theoretical analysis necessary for understanding the phenomenon of memorization and generalization in diffusion models. It rigorously establishes, in a statistical sense, the loss separation between the true and empirical score functions on empirical data, and quantitatively characterizes the network complexity required to learn each by leveraging results from universal approximation theory, which are appreciable theoretical contributions. Moreover, the authors effectively convey the high-level intuition behind their theorems, enabling readers to grasp the core ideas without following every technical detail.

**Weaknesses:**

- **Unclear theoretical hypothesis and motivation.** It is not entirely clear what specific hypothesis the theory aims to formalize or explain. Judging from the experiments, the paper seems to intend to relate sample size, network size, and weight decay to memorization and generalization in diffusion models, but the central claim or insight remains vague. The analysis does show that limited sample size can cause the empirical loss to favor memorization over generalization, but this is well-known and qualitatively unsurprising. It is unclear whether the authors' goal is to quantify the transition (e.g., identifying a regime such as $\log n = O(d)$) or to argue its relevance to large-scale diffusion models. The theoretical results are not explicitly connected to the empirical observations in Section 6.1, making it difficult to understand what the experiments are meant to test or demonstrate.

- **Weak empirical validation.**
The experiments do not concretely demonstrate the quantitative connection to the theory, e.g., the right balance between sample and network size is not discussed. Some claims, such as Lines 402–403 (“With sufficient sample size, network width increase promotes generalization”), depend heavily on narrow parameter ranges and is not properly substantiated (network width much greater than 1024 may eventually lead to memorization even for n=10K samples). The discussion of weight decay remains tautological (“proper network width and weight decay prevent memorization”) without clarifying what constitutes a proper scaling or whether such relationships can be derived from the theory.


- **Ambiguous role and limited contribution of the pruning method.**
The pruning scheme introduced in Section 6 seems disconnected from the main theoretical narrative. Reducing model size to mitigate memorization seems to be a straightforward idea and is not clearly tied to the presented theoretical framework. Its empirical impact appears modest, and many existing fine-tuning or guidance-based unlearning approaches are likely more effective. Its inclusion blurs the paper’s focus between theoretical analysis and engineering heuristics.


Overall, the paper seems uncertain about its main contribution: as a purely theoretical paper, the results are technically sound but conceptually expected; as a scientific study linking theory and empirical phenomena, it lacks a clear hypothesis-testing structure; and as an applied work, the proposed pruning method does not add substantial novelty.

**Questions:**

1. What is the central hypothesis that the experiments are designed to test? How are the theoretical results meant to explain the observed effects of sample size, network size, and weight decay on memorization and generalization?

2. Does the theory predict a quantitative regime (e.g., for sample size, $\log n = O(d)$ or larger) in which generalization can emerge, and are the chosen sample sizes in experiments intended to probe that regime?

3. Regarding the statement that “proper network width and weight decay prevent memorization,” can your theory specify how network size and weight decay should quantitatively relate to achieve generalization?

---

> ### Author Response · Authors · 2025-11-22
>
> Thank you for your recognition of our theoretical analysis on establishing the statistical separation and network complexity bounds related to memorization behaviors. We also appreciate your sharp observations and rigorous thoughts in the weaknesses and questions section — they are valuable for improving our work. We now summarize and address them as follows:
>
>
> > **Q1:** It is not entirely clear what the paper’s main contributions are, what specific hypothesis the theory aims to formalize or explain. The analysis does show that limited sample size can cause the empirical loss to favor memorization over generalization, but this is well-known and unsurprising.
>
> **A1:** Our main contribution is to rigorously establish separations between memorization and generalization from the dual lens of statistical learning and function approximation by neural networks. In particular, we formalize this through two theorems:
>
> 1. **Theorem 4.3 — Statistical Separation.** We establish non-asymptotic statistical bounds on denoising score matching, fundamentally reveal that perfectly optimizing the empirical denoising loss leads to memorization. Specifically, when the sample size satisfies $\log(n) = \mathcal{O}(d)$, the denoising score matching loss admits an inherent gap between the ground-truth score function and the empirical score function. This statistical gap leads to memorization.
> 2. **Theorem 5.1 — Network Architectural Separation.** We establish bounds for deep ReLU networks approximating the ground-truth and empirical score functions, showing that more expressive networks are needed to represent the empirical score than the truth score. This characterizes the network-size requirement for memorization.
>
> Our work provides new **non-asymptotic analysis** under **realistic deep-network settings** (deep ReLU architectures) and **general sub-Gaussian mixture distributions**, which cover many real-world scenarios. For instance, image distributions across multiple classes often follow such mixture structures.
>
> We acknowledge that it is unsurprising that limited sample size can lead to memorization. However, formalizing this intuition into a rigorous and non-asymptotic theory is highly nontrivial, especially under our general setting (deep ReLU networks and sub-Gaussian mixture distributions). Moreover, our analysis goes substantially beyond the basic intuition: as discussed after Theorem 4.3, large within-component variance can also influence memorization; The network architectural separation also provides additional insights connecting detailed network configurations to memorization behavior, as detailed in Theorem 5.1, approximating the empirical score requires both the network width and the number of non-zero parameters to scale with $n$, while the corresponding quantities for approximating the true score do not depend on the sample size.
>
>
>
>
> > **Q2:** The theoretical results are not explicitly connected to the empirical observations in Section 6.1.
>
> **A2:** The experiments in Section 6.1 are designed to qualitatively test the theoretical insights on sample size, network size, and weight decay:
>
> - **Sample size:** From Theorem 4.3, insufficient $n$ introduces a statistical loss gap that leads to memorization → we vary $n$ to validate this effect and observe a sharp transition from generalization to memorization as $n$ increases.
> - **Network size:** From Theorem 5.1, representing the empirical score requires larger networks; and if the network becomes expressive enough to represent the empirical score, memorization will occur → we test this by increasing the network size.
> - **Weight decay:** Theorem 5.1 also shows that representing the empirical score requires weight parameters with larger magnitude; thus, if the network attains such large norms, it is more likely to memorize → we vary the weight decay strength to examine its role in preventing memorization.
>
> Thus, the experiments are intended as strong demonstrations of the theoretical conclusions.

---

> ### Author Response · Authors · 2025-11-22
>
> > **Q3:** Does the theory predict a quantitative regime such as a required sample size or network size for generalization? And can the experiments demonstrate a quantitative connection to the theory, like specifying how network size and weight decay should quantitatively relate to achieve generalization?
>
> **A3:** Thank you for raising this point. In our setting, we have already identified a small sample zone where the statistical separation is valid, i.e., $\log n = \mathcal{O}(d)$, which leads to memorization. However, under our general assumptions (sub-Gaussian mixture data and deep ReLU networks), a precise asymptotic-type threshold is not given due to the technical nature of the non-asymptotic analysis.
>
> In simplified settings, such as asymptotic analysis using 2-layer random feature networks or studies assuming Gaussian mixture data with parameterized Gaussian mixture denoisers (see [1,2]), more precise phase transitions can indeed be characterized. Unfortunately, these techniques do not directly generalize to the general settings considered in our work. This motivates our choice to conduct experiments that qualitatively validate the theoretical connections, as explained in Q2 & A2.
>
> To the best of our knowledge, our analysis is already a meaningful step toward understanding these phenomena in a broader regime. Nevertheless, we agree that deriving more precise thresholds is an important future direction, and we are actively exploring this using more advanced statistical tools. We sincerely appreciate your suggestion and would welcome further discussion on potential approaches.
>
>
> > **Q4:** The inclusion of the pruning-based mitigation method blurs the paper’s focus between theoretical analysis and engineering heuristics, and the method itself is not clearly tied to the presented theoretical framework.
>
> **A4:** As we emphasized in Q1 & A1, we mainly focus on theory of provable separations between memorization and generalization. The pruning procedure is developed based on the theoretical insights (as discussed in Section 6.2) and intentionally kept simple. Its purpose is to provide empirical validation of the theoretical findings rather than to introduce a new SOTA mitigation technique, and in this sense the connection is strong.
>
> Both aspects of our theoretical framework directly guide the design of the pruning procedure. The statistical separation result (Theorem 4.3) shows that the loss gap between the empirical and true score functions is large especially in the small-$t$ regime. Therefore, in Section 6.2, we sample $t$ from a $\mathrm{Beta}(0.8, 2)$ distribution to emphasize small values of $t$ when computing importance scores for attention heads. The network architectural separation (Theorem 5.1) states that representing the empirical score requires a more expressive network. By pruning attention heads, we reduce the network’s expressiveness, making it less capable of representing the empirical score and thus less prone to memorization. And our experimental results serve their intended purpose: it validates that operations directly motivated by our theoretical insights can mitigate memorization.
>
>
>
>
> **References**
>
> [1] Buchanan, Sam, et al. *On the edge of memorization in diffusion models.* arXiv:2508.17689 (2025).
>
> [2] Bonnaire, Tony, et al. *Why Diffusion Models Don’t Memorize: The Role of Implicit Dynamical Regularization in Training.* arXiv:2505.17638 (2025).

---

### Official Review · Reviewer_M2PN · 2025-10-31

**Soundness:** 3
**Presentation:** 3
**Contribution:** 3
**Rating:** 6
**Confidence:** 4

**Summary:**

The paper proposes a theoretical approach to understand and mitigate memorization in diffusion models. The paper shows that there is (even with a polynomial number of training samples) a non-negligible loss gap for smaller t, which leads to the empirical loss being minimized, which in turn leads to memorization. As a solution, a pruning-based approach is presented, pruning the attention heads with the lowest importance score.

**Strengths:**

- The paper presents a strong theoretical understanding of why memorization happens in diffusion models.
- The paper shows that even with more training data, the loss gap will be present, showing that adding more training data to prevent memorization is not sufficient.

**Weaknesses:**

- In the experimental section, it is not mentioned which model architecture was used.
- The experiments are only conducted on small datasets such as the CIFAR-10 dataset and a synthetic Gaussian mixture dataset
- The pruning-based method is not evaluated against other SOTA pruning-based methods.

Misc:
- In line 95 there is a typo in "correspnding"
- Line 103 "an" -> "a"

**Questions:**

Q1: How does this theoretical approach support the empirical observation that deduplication seems to help with memorization mitigation?
Q2: Why is the fine-tuning after pruning the attention heads necessary? And more importantly, how do you make sure that no new samples in the fine-tuning set are memorized?
Q3: How are recall and precision calculated in Table 1?
Q4: Does this only hold for unconditioned text-to-image diffusion models? Does this also hold for other noise schedulers?

**Details Of Ethics Concerns:**

There are no concerns.

---

> ### Author Response · Authors · 2025-11-22
>
> Thank you for your recognition of our work and for kindly pointing out the typos. We now summarize the weaknesses and questions you raised and address them as follows:
>
> > **Q1:** Clarify the network architecture used in experiments.
>
> **A1:** Regarding the architectures we used: for the CIFAR-10 experiments, as mentioned in Appendix D.2, we adopt the DiT official implementation with hidden size 384, 12 layers, and 6 heads per layer. For the numerical simulations on Gaussian mixture data, we build standard MLPs and varied their width, depth and diffusion time embedding dimension to achieve different parameter settings. Detailed configurations can be found in the updated PDF, _Appendix D.3, Table 3_.
>
> > **Q2**: The pruning-based method is not evaluated against other SOTA pruning-based methods.
>
> **A2:** Firstly, we acknowledge that many strong pruning methods exist, yet their primary purpose is to reduce memory consumption and training cost; we instead use the pruning method as a plug-and-play mitigation to memorization. Using pruning to promote generalization and mitigate memorization is a new perspective driven by our theoretical insights (as discussed in Section 6.2) on the separations established in Theorems 4.3 and 5.1.
>
> Secondly, we mainly focus on theory of provable separations between memorization and generalization. The pruning procedure is **intentionally kept simple**, and its purpose is to provide empirical validation of the theoretical findings rather than to introduce a new SOTA mitigation technique.
>
>
> > **Q3**: Discussion on the relationship between theoretical results and the effect of sample duplication.
>
> **A3:** This is a very insightful question. We can discuss the effect of duplication from two perspectives.
>
> First, duplication can reduce the complexity of the network for representing the empirical score function, making memorization more likely. As stated in Theorem 5.1, when the dataset contains $n$ i.i.d. samples (and under our sub-Gaussian mixture assumptions, they are well-separated with high probability), approximating the empirical score requires both the network width and the number of non-zero parameters to scale with $n$, while the corresponding quantities for approximating the true score do not depend on the sample size. However, if $m$ samples are duplicates of the remaining $n - m$ i.i.d. samples, then from the theoretical viewpoint, the dataset effectively has size $n - m < n$. This implies that duplication makes the empirical score become easier to represent, and it only requires a less expressive network. Consequently, duplication makes memorization more likely to appear. We have also added this discussion to _Section 5_ of our updated PDF.
>
>
> Second, duplicated samples are more likely to be memorized. During training, we minimize the empirical loss $\int_{t_0}^T \frac{1}{n}\ell_t(x_i,s) dt$. If we have $n$ unique points, each data point $x_i$ contributes $1/n$ to the total loss. However, if a point $x_j$ is duplicated many times, it carries a much larger weight in the loss than other data points. The optimizer will prioritize minimizing the terms with the largest impact on the total loss, and the copies of $\ell(x_j,s)$ encourage the optimizer to reduce this specific term as much as possible. Such prioritization will make the duplicated points prone to memorization.

---

> > ### Author Response · Authors · 2025-11-22
> >
> > > **Q4:** Why is finetuning after pruning necessary?
> >
> > **A4:** Finetuning is considered because, in the original training, even the least important heads still play some role in the overall generalization performance, although a comparable model can also be trained from scratch without them. Thus, after pruning, the model still suffers a small performance drop. To mitigate this drop, we apply light finetuning so that the remaining heads can absorb the generalization role of the removed heads. In the finetuning stage, although the pruned model is exposed again to the training samples, it no longer has enough capacity to approximate the empirical score and is thus less likely to memorize them.
> >
> > > **Q5:** How are recall and precision calculated in Table 1?
> >
> > **A5:** Thank you for the clarification request, and we have added the following to *Appendix D.2* in our updated PDF. We compute recall and precision following [1]. Let the real and generated feature sets (feature embeddings extracted from the InceptionV3 network) be $\Phi\_r$ and $\Phi\_g$ with elements $\phi_r$ and $\phi_g$. Define the binary function $f(\phi,\Phi)=1$ if there exists $\phi' \in \Phi$ such that $||\phi-\phi'||\_2 \leq ||\phi' - \mathrm{NN}\_k(\phi',\Phi)||\_2$, and $0$ otherwise, where $\mathrm{NN}\_k(\phi',\Phi)$ returns the $k$-th nearest neighbor of $\phi'$ in $\Phi$. Precision is defined as $\text{precision}(\Phi\_r,\Phi\_g)=\frac{1}{|\Phi\_g|}\sum_{\phi\_g \in \Phi\_g} f(\phi\_g,\Phi\_r)$, and recall as $\text{recall}(\Phi\_r,\Phi\_g)=\frac{1}{|\Phi\_r|}\sum_{\phi\_r \in \Phi\_r} f(\phi\_r,\Phi\_g)$. Intuitively, each set forms a manifold via local $k$-NN neighborhoods; precision measures the fraction of generated samples lying within the real-data manifold, and recall measures the fraction of real-data lying within the generated-samples’ manifold.
> >
> > > **Q6:** Does the analysis only hold for unconditioned text-to-image diffusion models? Does it also hold for other noise schedulers?
> >
> > **A6:** Our theoretical analysis can adapt to different unconditional models because it operates in a general regime that only assumes the data distribution follows a sub-Gaussian mixture and that the denoising network is a deep ReLU architecture. The analysis can also be extended to conditional models, where the objective simply shifts from the unconditional distribution to the corresponding conditional distributions. Furthermore, our techniques apply to a wide range of noise schedulers, since $\sigma_t$ appears as a general time-dependent quantity in both the gap term of Theorem 4.3 and the approximation error in Theorem 5.1. Changing the noise scheduler therefore results only in minor adjustments to the range of the small-$t$ regime in Theorem 4.3.
> >
> >
> > [1] Kynkäänniemi, Tuomas, et al. "Improved precision and recall metric for assessing generative models." *Advances in neural information processing systems* 32 (2019).

---

### Author Response · Authors · 2025-11-22
**Global Comments**

We thank all the reviewers and chairs for your time and efforts. Your help and feedback have been very helpful in improving our work. We summarize our major revisions and responses to common questions raised by reviewers below.

---

> ### Author Response · Authors · 2025-11-22
> **Common Questions**
>
> >**Q1:** What is the major contribution of the paper?
>
>
> **A1:** Our main contribution is to rigorously establish separations between memorization and generalization from the dual lens of statistical learning and function approximation by neural networks. In particular, we formalize this through two theorems:
>
> 1. **Theorem 4.3 — Statistical Separation.** We establish non-asymptotic statistical bounds on denoising score matching, fundamentally reveal that perfectly optimizing the empirical denoising loss leads to memorization. Specifically, when the sample size satisfies $\log(n) = \mathcal{O}(d)$, the denoising score matching loss admits an inherent gap between the ground-truth score function and the empirical score function. This statistical gap leads to memorization.
> 2. **Theorem 5.1 — Network Architectural Separation.** We establish bounds for deep ReLU networks approximating the ground-truth and empirical score functions, showing that more expressive networks are needed to represent the empirical score than the true score. This characterizes the network-size requirement for memorization.
>
>
> Our work provides new **non-asymptotic analysis** under **realistic deep-network settings** (deep ReLU architectures) and **general sub-Gaussian mixture distributions**, which cover many real-world scenarios. For instance, image distributions across multiple classes often follow such mixture structures.
>
>
> We acknowledge that some prior works study related ideas, but they operate in more restrictive regimes, like under asymptotic analysis, data distribution as simple Gaussian mixtures, and assuming simple network architectures like 2-layer random feature networks or parametric Gaussian mixture denoisers. These assumptions limit the generality of their findings. In contrast, our work provides both statistical and network architectural guarantees by working with general sub-Gaussian mixture distributions and deep ReLU networks.
>
>
> > **Q2:** The inclusion of the pruning-based mitigation method seems to blur the paper’s focus between theoretical analysis and engineering heuristics, and the current experimental scale may be insufficient to assess its practical effectiveness.
>
> **A2:** As emphasized in Q1 & A1, we mainly focus on theory of provable separations between memorization and generalization. The pruning procedure is developed based on the theoretical insights (as discussed in Section 6.2) and intentionally kept simple. Its purpose is to **provide empirical validation of the theoretical findings** rather than to introduce a new SOTA mitigation technique. We believe the current experimental scale is sufficient to support our theoretical claims. As for real-world applications, we acknowledge that such an intentionally simple algorithm would require additional engineering heuristics. We welcome further discussion and insights on how these ideas might be adapted in more applied settings.
>
>
> > **Q3:** How sample duplication affects memorization, from a theoretical perspective?
>
> **A3**: First, duplication can reduce the complexity of the network for representing the empirical score function, making memorization more likely.  As stated in Theorem 5.1, when the dataset contains $n$ i.i.d. samples (and under our sub-Gaussian mixture assumptions, they are well-separated with high probability), approximating the empirical score requires both the network width and the number of non-zero parameters to scale with $n$, while the corresponding quantities for approximating the true score do not depend on the sample size. However, if $m$ samples are duplicates of the remaining $n - m$ i.i.d. samples, then from the theoretical viewpoint, the dataset effectively has size $n - m < n$. This implies that duplication makes the empirical score become easier to represent, and it only requires a less expressive network. Consequently, duplication makes memorization more likely to appear.
>
>
> Second, duplicated samples are more likely to be memorized. During training, we minimize the empirical loss $\int_{t_0}^T \frac{1}{n}\ell_t(x_i,s) dt$. If we have $n$ unique points, each data point $x_i$ contributes $1/n$ to the total loss. However, if a point $x_j$ is duplicated many times, it carries a much larger weight in the loss than other data points. The optimizer will prioritize minimizing the terms with the largest impact on the total loss, and the copies of $\ell(x_j,s)$ encourage the optimizer to reduce this specific term as much as possible. Such prioritization will make the duplicated points prone to memorization.

---

> ### Author Response · Authors · 2025-11-22
> **Changes in the Updated Version**
>
> As advised by the reviewers, in the updated PDF we fix several typos and highlight all modifications below in _purple_.
> 1. **(Statistical separation under varying $t$ and sample size $n$)** We add Figure 1 in Section 4, which evaluates the statistical separation gap under a Gaussian mixture setting by varying both $t$ and the sample size $n$. The results align with our theory: the gap grows as $t$ decreases, and insufficient sample size also leads to a large gap.
> 2. **(Effect of sample duplication)** We include a discussion in Section 5 on the relationship between sample duplication, memorization, and network architectural separation.
> 3. **(Computation of precision and recall)** We provide the computation method for precision and recall in Appendix D.2.
> 4. **(Network configuration details)** We provide the network architectures and detailed configurations used in the experiments in Appendix D.3.
> 5. **(Additional references)** We include additional theoretical references in Section 2 to present a more comprehensive overview of related work in this area.
>
>
> We are also preparing further updates to incorporate additional insightful comments from the reviewers. We look forward to further discussion!

---

### Meta-Review · Area_Chair_dV1S · 2025-12-30

**Summary:**

The paper addresses the question of: „Can we disentangle memorization from generalization in practical regimes and mitigate it?“. The reviewers’ main concerns that remain after the rebuttal are about the presentation and emphasizing the novelty of the contribution.

**Reviewer Concerns:**

The rebuttal already does a good job in explaining these, however, the insights are not yet very well reflected in the writing of the paper (see comments for the second reviewer R2). Another major concern raised is the limited experimental evaluation on all ends, namely using small models, low-complexity data, and not comparing against the pruning-based baselines. Finally, the reviewers note that the pruning-based method does not yield substantial improvements. The authors argue that all experimentation is just to support the theory. Given that the main contributions of the paper indeed are of a theoretical nature, this might be a valid approach, therefore, I weakly suggest accepting the paper but encourage the authors strongly to improve their presentation based on the below-mentioned comments, and potentially think about running the baseline experiments for the pruning method and the memorization of fine-tuning data experiment, just to strengthen their claims.

**Detailed concerns of the respective reviewers and how it was addressed in the rebuttal**

R1:
- Experimental details:
    - Were provided in appendix
- No comparison to SOTA pruning
    - The authors say they don’t want SOTA results, they only want to validate the theory. However, comparing against Chavhan et al, 2024, and Hintersdorf et al, 2024 would have made it much stronger. How much does it help to know this theory vs. just using more empirically-based methods? This could be answered by adding the adequate experiments.
- Does fine-tuning not introduce additional memorized samples.
    - While the authors say that the remaining parameters do not have enough capacity to memorize, the AC could not find any theoretical or empirical validation for this claim. An experiment on measuring memorization of the fine-tuning data would mitigate this concern.


R2:
- Unclear theoretical hypothesis and motivation
    - The answer in the rebuttal was pretty clear, but this does not reflect in the updated version of the paper that the AC reviewed. Overall, what is described in the rebuttal might make a better entry (or at least a good addition) to the contributions currently listed in the paper’s introduction.
    - That being said, indeed, the observations are not particularly novel, see for example also NEURIPS.
- Weak evaluation
    - The authors argue that their evaluation is there to validate the theory. And this seems correct. As stated for R1, the AC would still suggest strengthening the evaluation by adding empirical comparison to Chavhan et al, 2024, and Hintersdorf et al, 2024, and measuring the fact that the new fine-tuning data is not memorized after fine-tuning.
- Limited contribution of pruning
    - AC agrees: looking at table 1, and seeing the standard deviation for memorization rate „ours“ vs. „random“, it seems that they lie still exactly at the border of each other’s standard deviation, indeed weakening the impact that the method has. The difference is in FID where random pruning harms significantly more. This finding is contrary to what was reported by Hintersdorf et al., 2025 where FID was rather stable among both, but slightly worse for removing memorizing neurons.

R3:
- Analysis only on small t
    - Has been addressed by adding the new Figure 1 which assesses multiple t, showing that they expose the same trend while larger t have a smaller improvement in the loss gap based on the sample size (given they are initially also much lower)
- Strong assumptions for the theory, namely sub-Gaussianity, which limit their applicability to real-world data
    - The authors argue that this is common in real-world data and acknowledge that the theory does not apply to heavy-tail data, which might be worth adding to the paper.
- Pruning tested on one dataset, hence, generalization is unclear
    - The authors argue that it is rather to validate the theory. While this makes sense, again, comparison would strengthen.


R4:

- Limited novelty of the core findings
    - Authors argue that their second core finding on the architectural separation is novel and provides insights into what models will memorize
- Low-resolution, low complexity datasets used in evaluation
    - The authors say that is not possible to run at Billion parameter models and complex data. However, it would be more convincing if they had explained what are exactly the bottlenecks. Why not just take a retrained model?
- Small improvement through pruning method
    - Emphasize again that they focus on the theory.

**Reviewer Scores:**

The reviewers were initially positive and remained mainly positive, so it would be expected that the work would have been suggested for acceptance.

---

### Decision · Program_Chairs · 2026-01-26

Accept (Poster)